# DeepRTAlign: toward accurate retention time alignment for large cohort mass spectrometry data analysis

Yi Liu[1,2,8], Yun Yang [3,4,8], Wendong Chen[3,4], Feng Shen[5], Linhai Xie[2,3,4], Yingying Zhang[2,6], Yuanjun Zhai[2], Fuchu He [2,3,7], Yunping Zhu [2] ✉ & Cheng Chang [2,7] ✉

Retention time (RT) alignment is a crucial step in liquid chromatography-mass spectrometry (LC-MS)-based proteomic and metabolomic experiments, especially for large cohort studies. The most popular alignment tools are based on warping function method and direct matching method. However, existing tools can hardly handle monotonic and non-monotonic RT shifts simultaneously. Here, we develop a deep learning-based RT alignment tool, DeepRTAlign, for large cohort LC-MS data analysis. DeepRTAlign has been demonstrated to have improved performances by benchmarking it against current state-of-the-art approaches on multiple real-world and simulated proteomic and metabolomic datasets. The results also show that DeepRTAlign can improve identification sensitivity without compromising quantitative accuracy. Furthermore, using the MS features aligned by DeepRTAlign, we trained and validated a robust classifier to predict the early recurrence of hepatocellular carcinoma. DeepRTAlign provides an advanced solution to RT alignment in large cohort LC-MS studies, which is currently a major bottleneck in proteomics and metabolomics research.

Liquid chromatography (LC) is usually coupled with mass spectrometry (MS) in proteomics experiments to separate complex samples. The retention time (RT) of each analyte in MS data usually have shifts for multiple reasons, including matrix effects and instrument performances[1]. Thus, in any experiment involving multiple samples, corresponding analytes must be mapped before quantitative, comparative, or statistical analysis. This process is called correspondence[2]. In other words, this problem can be defined as finding the "same compound" in multiple samples. Generally, in proteomics, correspondence can be done based on peptide identifications. However,

taking the test data in this study as an example, only 15%–25% of precursors have the corresponding identifications due to the data-dependent ion selection process in the data-dependent acquisition (DDA) mode. Even for the data-independent acquisition (DIA) data, a number of unidentified precursors (potential peptides) remain, which cannot be considered in the subsequent analysis due to the complex MS/MS spectra[3]. Most existing tools for DDA and DIA data analysis, such as MaxQuant[4], PANDA[5], MSFragger[6,7] and DIA-NN[8], perform RT alignment using the match between runs (MBR) function (also called the cross-align function) to transfer the identified sequences to the

[1]Faculty of Environment and Life, Beijing University of Technology, Beijing 100023, China. [2]State Key Laboratory of Proteomics, Beijing Proteome Research Center, National Center for Protein Sciences (Beijing), Beijing Institute of Lifeomics, Beijing 102206, China. [3]International Academy of Phronesis Medicine (Guang Dong), No. 96 Xindao Ring South Road, Guangzhou International Bio Island, Guangzhou 510000, China. [4]South China Institute of Biomedicine, No. 83 Ruihe Road, Guangzhou 510535, China. [5]Department of Hepatic Surgery IV, the Eastern Hepatobiliary Surgery Hospital, Naval Medical University, Shanghai 200433, China. [6]Chongqing Key Laboratory on Big Data for Bio Intelligence, Chongqing University of Posts and Telecommunications, Chongqing 400065, China. [7]Research Unit of Proteomics Driven Cancer Precision Medicine, Chinese Academy of Medical Sciences, Beijing 102206, China. [8]These authors contributed equally: Yi Liu, Yun Yang. ✉e-mail: zhuyunping@gmail.com; changchengbio@163.com

unidentified precursors between any two LC-MS runs. Although MBR can increase the total number of identifications to some extent, it is integrated into specific software tools and relies on the identified peptides, which limits its further application in clinical proteomics research to explore new biomarkers from unidentified precursors. In metabolomics, feature alignment is a prerequisite for identification and quantification. In theory, the accuracy of feature alignment depends on the m/z and RT information in MS data. Currently, high-resolution mass spectrometers can limit the m/z shift to less than 10 ppm. Thus, RT alignment is especially important for accurately analyzing large-scale proteomics and metabolomics research data.

There are two types of computational methods for RT alignment. One is called the warping method. Warping models first correct the RT shifts of analytes between runs by a linear or non-linear warping function[2, 9]. There were several existing popular alignment tools based on this method, such as XCMS[10], MZmine 2[11] and OpenMS[12]. However, this warping method is not able to correct the non-monotonic shift because the warping function is monotonic[9]. Another kind of method is the direct matching method, which attempts to perform correspondence solely based on the similarity between specific signals from run to run without a warping function. The representative tools include RTAlign[13], MassUntangler[14] and Peakmatch[15]. The performances of the existing direct matching tools are reported inferior to the tools using warping functions due to the uncertainty of MS signals[2]. Either way, these tools mentioned above can hardly handle both monotonic and non-monotonic RT shifts. Thus, machine learning or deep learning techniques are applied to solve this issue. Li et al. applied a Siamese network for accurate peak alignment in gas chromatography-MS data from complex metabolomic samples[16]. But there is no deep learning-based alignment algorithm for LC-MS data analysis currently.

Here, we present a deep learning-based RT alignment tool, named DeepRTAlign, for large cohort LC-MS data analysis. Combining a coarse alignment (pseudo warping function) and a deep learning-based model (direct matching), DeepRTAlign can deal with monotonic shifts as well as non-monotonic shifts. We have demonstrated its high accuracy and sensitivity in several proteomic and metabolomic datasets compared with existing popular tools. Further, DeepRTAlign allows us to apply MS features directly and accurately to downstream biological analysis, such as biomarker discovery or prognosis prediction, which can complement traditional identification (ID)-based methods.

## Results

### Workflow of DeepRTAlign

The whole workflow of DeepRTAlign is shown in Fig. 1, and can be divided into two parts, i.e., the training part and the application part. The training part contains the following steps.

(1) Precursor detection and feature extraction. Taking raw MS files as input, precursor detection and its feature extraction were performed using an in-house developed tool XICFinder, an MS feature extraction tool similar to Dinosaur[17]. The algorithm of XICFinder is based on our quantitative tool PANDA[5]. Using the Component Object Model (COM) of MSFileReader, it can handle Thermo raw files directly. XICFinder first detects isotope patterns in each spectrum. Then, the isotope patterns detected in several subsequent spectra are merged into a feature. A mass tolerance of 10 ppm was used in this step.

(2) Coarse alignment. First, the RT in all the samples will be linearly scaled to a certain range (e.g., 80 min in this study, as the RT range of training dataset HCC-T is 80 min). Second, for each m/z, the feature with the highest intensity is selected to build a new list for each sample. Then, all the samples except an anchor sample (we considered the first sample as the anchor in this study) will be divided into pieces by a user-defined RT window (we used 1 min in

this study). All the features in each piece are compared with the features in the anchor sample (mass tolerance: 0.01 Da). If the same feature does not exist in the anchor sample, this feature is ignored. Then, the RT shift is calculated for each feature pair. For all the features in each piece, the average RT shift is calculated. To align each piece with the anchor sample, the average RT shift between a piece and the anchor sample is directly added to each feature in this piece.

(3) Binning and filtering. All features will be grouped based on m/z, according to the parameters bin_width and bin_precision (0.03 and 2 by default). Bin_width is the m/z window size, and bin_precision is the number of decimal places used in this step. Only the features in the same m/z window will be aligned. After the binning step, there is also an optional filtering step. For each sample in each m/z window, only the feature with the highest intensity will be kept in the user-defined RT range. DeepRTAlign does not perform the filtering step by default.

(4) Input vector construction. Only the RT and m/z of each feature are considered when constructing the input vector. As shown in Fig. 1a, we consider two adjacent features (according to RT) before and after the target feature corresponding to a peptide. Part 1 and Part 4 are the original values of RT and m/z. Part 2 and Part 3 are the difference values between the two samples. Then, we used two base vectors (base_1 is [5, 0.03] and base_2 is [80, 1500]) to normalize the features (the difference values will be divided by base_1, the original value will be divided by base_2). Each feature-feature pair will be transferred to a $5 \times 8$ vector as the input of the deep neural network (Supplementary Fig. 1).

(5) Deep neural network (DNN). The DNN model in DeepRTAlign contains three hidden layers (each has 5000 neurons), which is used as a classifier that distinguishes between two types of feature Supplementary Table -feature pairs (i.e., the two features should be or not be aligned). Finally, a total of 400,000 feature-feature pairs were collected from the HCC-T dataset (1) based on the Mascot identification results (mass tolerance: ±10 ppm, restrict the RT of a peptide to be within the RT range of the corresponding precursor feature). 200,000 of them are collected from the same peptides, which should be aligned (labeled as positive). The other 200,000 are collected from different peptides with a m/z tolerance of 0.03 Da, which should not be aligned (labeled as negative). These 400,000 feature-feature pairs were used to train the DNN model. It should be noted that it is not necessary to know the peptide sequences corresponding to the features when performing feature alignment. The identification results of several popular search engines (such as Mascot, MaxQuant and MSFragger) are only used as ground truths when benchmarking DeepRTAlign.

(6) The hyperparameters in DNN. BCELoss function in Pytorch is used as the loss function. The sigmoid function was used as the activation function. We used the default initialization method of pytorch (kaiming_uniform). We used Adam (betas = (0.9, 0.999), eps = 1e-08, weight_decay = 0, amsgrad = False, foreach = None, maximize = False, capturable = False, differentiable = False, fused = None) as the optimizer. The initial learning rate is set to 0.001, and is multiplied by 0.1 every 100 epochs. Batch size is set to 500. The number of epochs is chosen empirically. We conducted several trial runs and found that the loss tended to be stable at 100–300. So, we set the epoch number to 400 for final training. All other parameters are kept by default in Pytorch v1.8.0.

(7) Parameter evaluation. Network parameters used were examined on the training set (HCC-T) by 10-fold cross validation and the best parameters were selected based on the cross-validation results (Supplementary Table 2). The trained model was evaluated on several independent test sets (Supplementary Table 3). These results demonstrated that there is no overfitting in the DNN model.

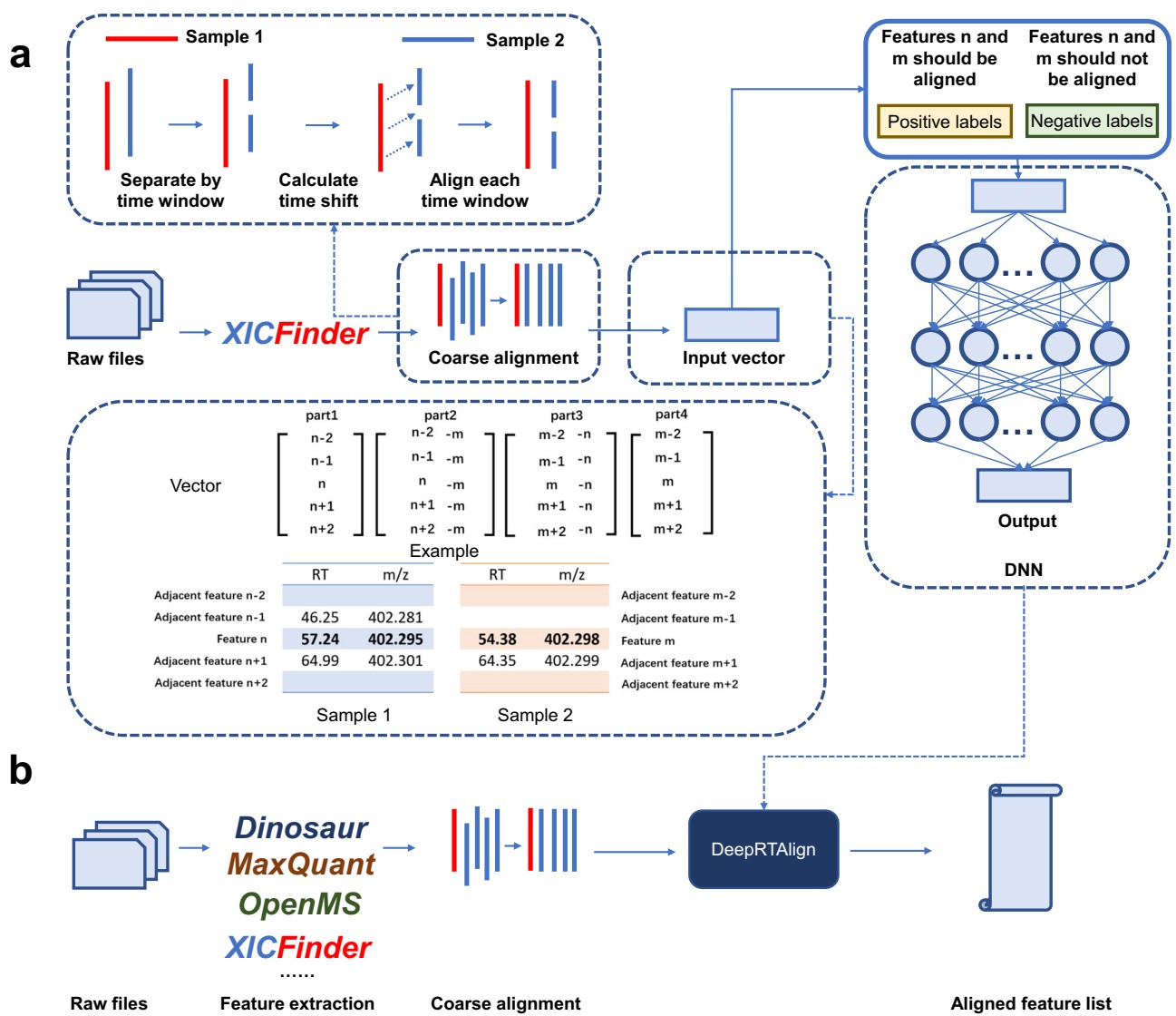

**Fig. 1 | The illustration of DeepRTAlign algorithm. a** The training procedures of DeepRTAlign. The n is the feature rank number (sorted by RT) in one m/z window of sample 1. The m is the feature rank number (sorted by RT) in one m/z window of sample 2. Please note that "n-1", "n-2", "n + 1" and "n + 2" represent four adjacent features of the feature n in one m/z window. **b** The workflow for RT alignment using DeepRTAlign. DeepRTAlign supports four feature extraction methods (Dinosaur, MaxQuant, OpenMS and XICFinder). The features extracted will be aligned using the trained model shown in (**a**), with the aligned feature list as the output.

(8)   Quality control (QC) of the output. A QC module was implemented in DeepRTAlign to perform QC for the output of DeepRTAlign. As shown in Supplementary Fig. 2, in each m/z window, DeepRTAlign will randomly select a sample as a target and build its decoy. In theory, all features in the decoy sample should not be aligned. Based on this, the final false discovery rate (FDR) of the alignment results can be calculated.

In the application part (Fig. 1b), DeepRTAlign directly supports the results of four MS feature extraction tools, i.e., Dinosaur[17], MaxQuant, OpenMS and XICFinder, as input. Feature lists from other tools (such as MZmine 2) can be used after converting format to txt or csv files. In this part, feature lists will first go through the coarse alignment and input vector construction steps, as shown in the training part. Then, the constructed input vectors will be fed into the trained DNN model. According to the classification results of the DNN model, DeepRTAlign will output an aligned feature list for further analysis.

## Model evaluation on the training set and the test sets

To optimize the network parameters in the DNN model, the best parameters were selected based on the 10-fold cross-validation results on the training set (HCC-T) (Supplementary Table 2). Then, to benchmark the DNN model, we additionally trained four models using several popular machine learning methods (RF, KNN, SVM and LR) on the same training set (HCC-T). The test results of our DNN model and all the other machine-learning models are shown in Supplementary Table 3. We found that our DNN model owned the highest AUC compared with other models. Although the DNN model is trained on the HCC-T dataset, it achieved good generalizability and can be applied to other datasets with different sample types or species.

Then, we randomly collected 2000 negative pairs correctly predicted by both DNN and RF (BOTH-right pairs), and 2000 negative pairs correctly predicted by DNN but incorrectly predicted by RF (DNN-right pairs) in a test set (HCC-N, RT range: 1–80 min, m/z tolerance: 0.03 Da, the same way as we collected the negative pairs in the training set HCC-T, see "Workflow of DeepRTAlign"). The average RT differences of the 2000 BOTH-right pairs and the 2000 DNN-right pairs were similar (5.03 min and 4.89 min, respectively). However, the average m/z difference of the 2000 BOTH-right pairs was 0.011 Da, which was five times the average m/z differences of the 2000 DNN-

right pairs (0.002 Da). These results indicated that the advantages of DNN may mainly lie in recognizing the negative pairs with close m/z values. And we found that those negative pairs with similar m/z values (with differences less than 0.002 Da) were rare cases in the whole training data (10% only). Traditional machine learning methods usually are difficult to discriminate and predict correctly in such a situation. Although the bootstrapping-based RF is supposed to alleviate the problem such as imbalanced data in some degree, its classification power will drastically drop when the minority is extremely rare[18]. The minor pairs might be scarcely sampled during the bootstrapping and their effects on trees are overwhelmed by the majority of the data. In contrast, the DNN model is trained with mini-batches iteratively through gradient descent, which guarantees each pair, including the rare cases, is learned through a shuffled order. Particularly, those incorrectly predicted rare cases lead to greater loss values and thus greater gradients to update the model, which may explain why DNN can predict them better.

## Ablation analysis

We performed the ablation analysis to evaluate the DNN model in DeepRTAlign on different test sets (HCC-N, HCC-R, UPS2-M, UPS2-Y, EC-H, AT and SC). For the coarse alignment step, we have shown that there are no obvious differences when using different samples as the anchor sample (Supplementary Table 4). We also tested the performance of DeepRTAlign with or without the coarse alignment step (Supplementary Table 5). We found that our DNN model with the coarse alignment step owned a higher AUC compared with the same DNN model without coarse alignment.

Then, we calculated the importance of each feature in the DNN model, the RF model and the LR model to show which feature is more important when training the DNN, RF and LR models on the same training set (HCC-T). For DNN, the importance of each feature in DNN was calculated as the AUC decrease tested in the HCC-N dataset when

each feature in DNN was replaced with a random value, compared to the DNN model without random features. For RF, the feature importance is the attribute "feature_importances_" in sklearn.ensemble.RandomForestClassifier (scikit-learn framework v0.21.3). For LR, the feature importance is the attribute "coef_" in sklearn.linear_model.LogisticRegression (scikit-learn framework v0.21.3). As shown in Supplementary Table 6, we found that feature 6 ($mz_n$-$mz_m$) and feature 16 ($mz_m$-$mz_n$) were the top 2 important features, indicating that DNN, RF and LR models relied mostly on the features about m/z for alignment prediction. We also found the importances of feature 5 ($RT_n$-$RT_m$) and feature 15 ($RT_m$-$RT_n$) in DNN were about half of feature 6 ($mz_n$-$mz_m$) and feature 16 ($mz_m$-$mz_n$), while the ratio (importance of feature 5/importance of feature 6) decreased to only about 32.7% in RF and 0.72% in LR. This difference indicates that the importance of RT-related features in the DNN model is relatively higher than that in the RF and LR models.

## Comparison with existing alignment tools

According to the information required, all the alignment tools can be divided into three types (Supplementary Table 7). DeepRTAlign, MZmine 2 and OpenMS only need MS information. Quandenser requires both MS and MS/MS information. In addition, MaxQuant, MSFragger and DIA-NN with MBR functions require identification results for alignment.

## Comparison with MZmine 2, OpenMS

First, DeepRTAlign was compared with two other popular MS-based only alignment tools, MZmine 2 and OpenMS, on proteomic datasets. As shown in Fig. 2, we found that DeepRTAlign showed an improvement in both precision and recall compared with OpenMS and MZmine 2. The noise threshold in MZmine 2 is the key parameter associated with the number of extracted features. But it is difficult for users to choose. Lowering the noise threshold will further

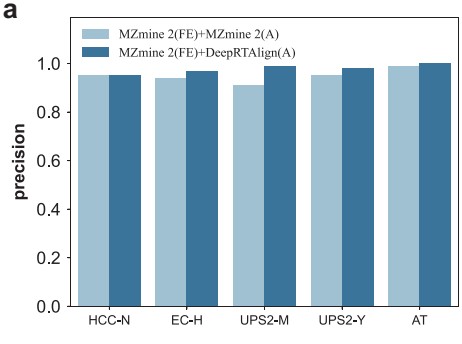

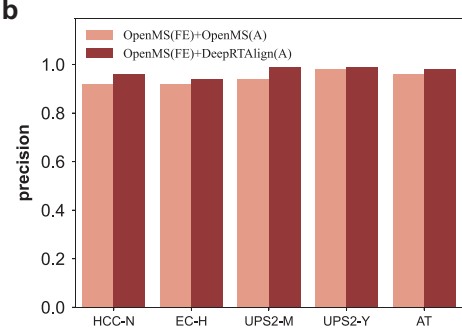

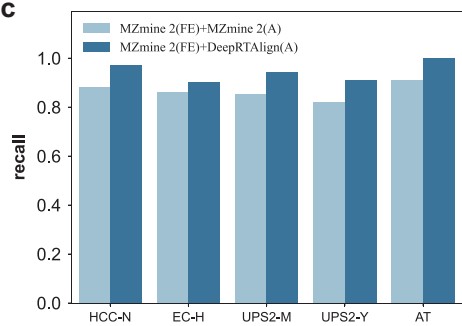

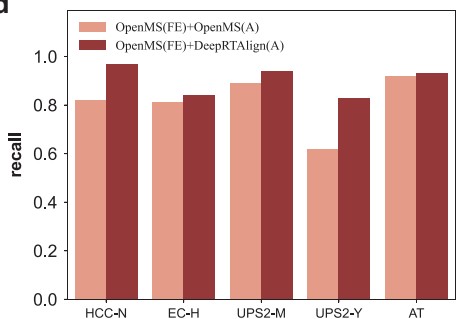

**Fig. 2 | Performance evaluation of DeepRTAlign compared with MZmine 2 and OpenMS. a, c** The precisions and recalls of MZmine 2 and DeepRTAlign on different test sets. **b, d** The precisions and recalls of OpenMS and DeepRTAlign on different test datasets. "FE" means the feature extraction method. "A" means the RT alignment method. We took the Mascot identification results with FDR < 1% as the ground truth in datasets HCC-N, UPS2-M and UPS2-Y and took the MaxQuant identification results with FDR < 1% as the ground truth in datasets EC-H and AT. In dataset EC-H, we only considered the E. coli peptides for evaluation. In datasets UPS2-M and UPS2-Y, we only considered the UPS2 peptides for evaluation. Source data are provided as a Source Data file.

increase its extraction time. The focus of this work is not to compare the pros and cons of different feature extraction tools, so we chose relatively high noise thresholds (1.0E6 on proteomic data, 1.0E5 on metabolomic data) to ensure the feature quality and control of the running time.

Then, we also compared DeepRTAlign with MZmine 2 and OpenMS on a public real-world metabolomic dataset SM1100[19]. This dataset was generated from standard mixtures consisting of 1100 compounds with specified concentration ratios. It contains two groups (SA and SB), and each group has 5 replicates. We used DeepRTAlign (combined with 3 different feature extraction tools: MZmine 2, OpenMS and Dinosaur), MZmine 2 and OpenMS to align features across runs. Thus, it resulted in five different algorithm combinations (Supplementary Table 8). Here, to demonstrate DeepRTAlign's capacity to deal with metabolomic data, every adjacent sample pair in each group (i.e., SA1-SA2, SA2-SA3, SA3-SA4, SA4-SA5 and SB1-SB2, SB2-SB3, SB3-SB4, and SB4-SB5) was aligned using the five algorithm combinations.

For feature extraction, the default parameters in OpenMS and Dinosaur were used. In MZmine 2, the parameter "Noise Level" was set to 1.0E5 to make the extracted feature number similar to those of OpenMS and Dinosaur. Then, the extracted features in each sample were annotated according to the 1100 standard compounds with strict standards (mass tolerance: ±5 ppm, RT tolerance: ±0.5 min as suggested in ref. [19]). Based on the annotation results, the precision and recall values for each combination were calculated and showed in Supplementary Table 8. We can see that all five combinations performed well due to the simpler composition of this metabolomic standard dataset (compared with the proteomic datasets).

## Comparison with quandenser

DeepRTAlign was then compared with another popular tool Quandenser[20] which used both MS and MS/MS information. Quandenser applies unsupervised clustering on both MS1 and MS2 levels to summarize all analytes of interest without assigning identities. Using Dinosaur as the feature extraction method, we compared DeepRTAlign with Quandenser on the Benchmark-FC dataset. We mapped all the extracted features to the identification results and only considered the Escherichia coli (E. coli) peptides without missing values in all the replicates. As shown in Supplementary Fig. 3, we found that DeepR-TAlign was comparable to Quandenser in terms of the number of aligned peptides and the quantification accuracy. But DeepRTAlign could align many more features (no matter if there were corresponding identification results) than Quandenser since it was an ID-free alignment approach.

## Comparison with MaxQuant, MSFragger and DIA-NN with or without MBR

We compared DeepRTAlign with the alignment methods based on the identification results, which was currently the most used alignment strategy. MBR is an updated version of these kinds of alignment methods, which can transfer identification results to the un-identification features[21]. We further compared DeepRTAlign with MaxQuant's MBR and MSFragger's MBR on the Benchmark-FC dataset (Fig. 3). MaxQuant or MSFragger was run twice with and without the MBR function while keeping the other parameters unchanged.

As shown in Fig. 3, by combining MaxQuant to extract features, DeepRTAlign to align features and MSFragger to identify their peptide sequences, we could obtain 150% more peptides compared to the

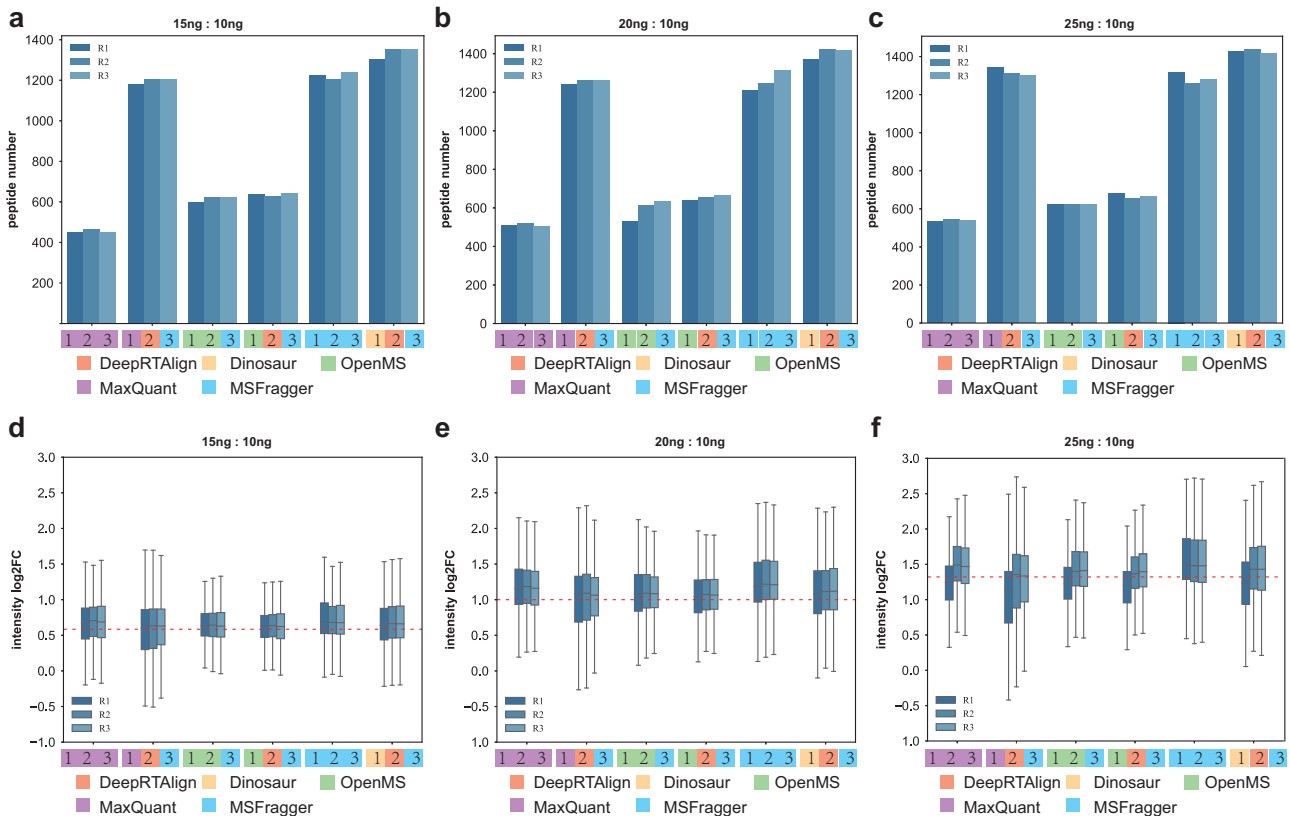

**Fig. 3 | The number and ratio distributions of all E. coli peptides between specific samples (15 ng/10 ng, 20 ng/10 ng, and 25 ng/10 ng) in each replicate (R1, R2 and R3) after alignment. a–c** The E. coli peptide number in each sample after alignment. **d–f** The ratio distributions of all E. coli peptides between specific samples (15 ng/10 ng, 20 ng/10 ng, and 25 ng/10 ng). The numbers 1, 2 and 3 in the legends indicate feature extraction method, alignment method and identification method, respectively. In all boxplots, the center red line is the median of log2 of the peptide fold changes. The box limits are the upper and lower quartiles. The whiskers extend to 1.5 times the size of the interquartile range. Source data are provided as a Source Data file.

original MaxQuant workflow (using MaxQuant to extract features, align them and identify their peptide sequences) without compromising the quantification accuracy. DeepRTAlign also aligned 7% and 14.5% more peptides than OpenMS and MSFragger's MBR, respectively, while its quantification accuracy was comparable to other methods. Benchmarked on dataset Benchmark-MV against MSFragger's MBR, DeepRTAlign also aligned more peptides with a comparable quantification accuracy (Supplementary Fig. 4). These results demonstrated DeepRTAlign had a better performance for alignment than MBR-applied MaxQuant and MSFragger.

It should be noted that since MSFragger did not provide its extracted feature list, we used the features extracted by Dinosaur, OpenMS and MaxQuant for a fair comparison. DeepRTAlign is compatible with multiple feature extraction methods (Dinosaur, OpenMS, MaxQuant and XICFinder), which is convenient for users to choose the feature extraction method suitable for their experiments.

Furthermore, DeepRTAlign also performed better than MBR in DIA-NN on single-cell proteomics DIA data (Supplementary Fig. 5). Considering the aligned peptides present in at least two cells, DeepRTAlign can align 39 (6.33%) more peptides on average than the existing popular tool DIA-NN in each cell. Moreover, using DeepRTAlign, the average number of the aligned features present in at least two cells is approximately 42.3 times the average number of the aligned peptides, providing the possibility to identify different cell types using the aligned MS features besides the aligned peptides in the future.

### Generalizability evaluation on simulated datasets
Based on the 14 real-world datasets (9 proteomic datasets: EC-H, HCC-N, UPS2-M, UPS2-Y, HCC-R, AT, MI, SC and CD; 5 metabolomic datasets: NCC19, SM1100, MM, SO and GUS), we generated a total of 336 (14*24) simulated datasets with different RT shifts to evaluate the generalizability boundary of DeepRTAlign.

As shown in Fig. 4, it can be found that in most cases, DeepRTAlign owns higher precision and recall values than OpenMS on the proteomic datasets. Meanwhile, we also find DeepRTAlign and OpenMS perform worse when the standard deviation of the RT shift increases (from 0.1 to 5). Thus, we recommend that the RT shift distribution be controlled to different levels in practical applications for proteomic and metabolomic studies. In most proteomic datasets, when the standard deviation of RT shift is larger than 1 min, the precision and recall drop significantly (Fig. 4). In most metabolomic datasets, a similar phenomenon occurs when the standard deviation is larger than 0.3 min, especially in recall (Supplementary Figs. 6, 7). Under the FDR < 1%, DeepRTAlign outperformed OpenMS on two metabolomic datasets (SM1100 and MM) but not on the other datasets (NCC19, SO and GUS). We explored the extracted features in datasets NCC19, SO and GUS and found that they contained a large number of features with very close m/z. One possible reason is this type of data is not compatible with our current decoy design method. Thus, we further tested DeepRTAlign on these three datasets by setting FDR threshold to 100%. As shown in Supplementary Fig. 7, we found that the performance of DeepRTAlign became comparable with that of OpenMS on these three datasets. However, it should be noted that OpenMS does not perform well on these three datasets either, indicating a potential limitation for MS1-only alignment method. Meanwhile, we found that both proteomic and metabolomic datasets were unaffected when only changing the mean of RT shifts.

### MS features aligned by DeepRTAlign enable accurate prediction of HCC early recurrence
The accurate detection of HCC early recurrence after surgery is one of the main challenges in liver cancer research[22]. In our previous work[23], MS features were proved to have a good discriminating power to classify tumor and non-tumor samples.

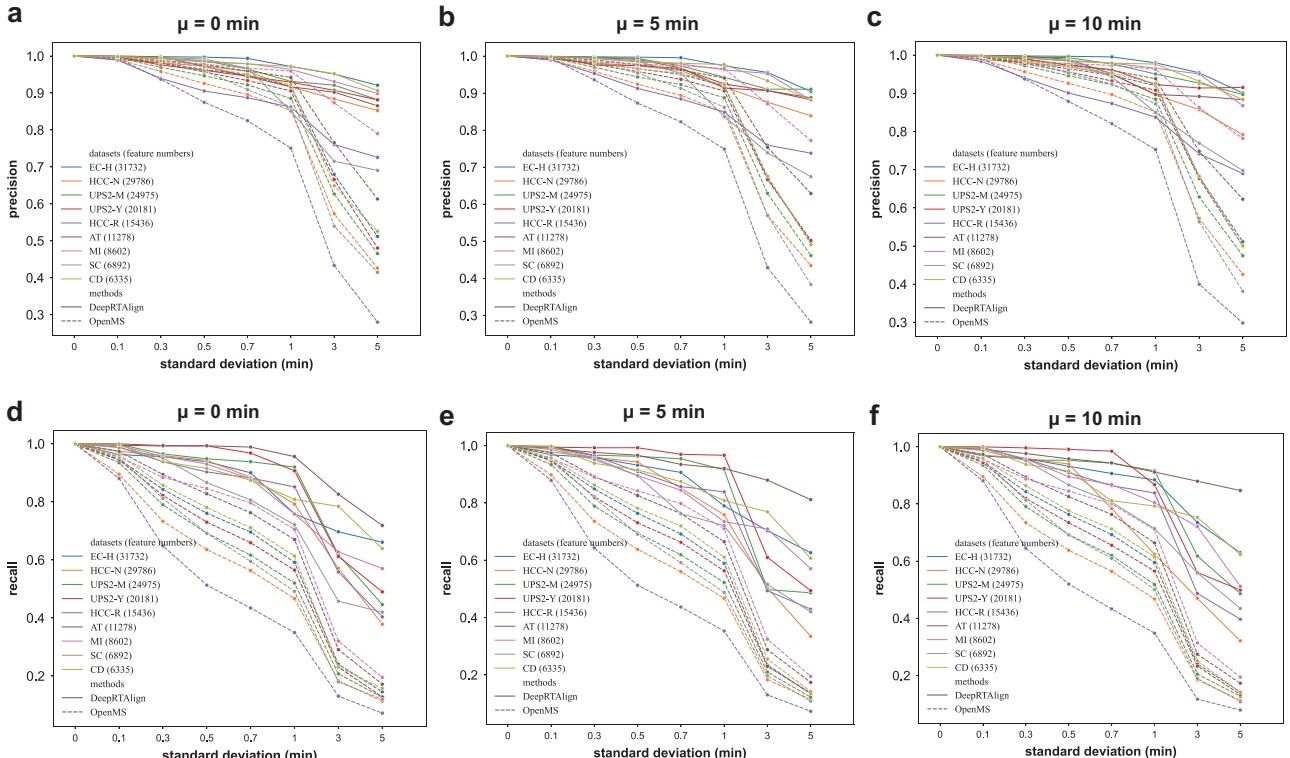

**Fig. 4 | Comparison of DeepRTAlign and OpenMS on multiple simulated datasets generated from 9 real-world proteomic datasets.** The simulated datasets were constructed by adding normally distributed RT shifts to the corresponding real-world dataset. (**a**, **d**) μ = 0 min. **b**, **e** μ = 5 min. **c**, **f** μ = 10 min. The normal distribution has an increasing σ, i.e., σ = 0, 0.1, 0.3, 0.5, 0.7, 1, 3, 5 for different μ (0, 5 and 10 min), respectively. Source data are provided as a Source Data file.

Here, we applied DeepRTAlign to re-analyze the proteomics data of the tumor samples from 101 HCC patients in Jiang et al.'s paper[24] and used the aligned MS features to train a classifier for predicting HCC early recurrence (Fig. 5a). For all the 101 patients in Jiang et al.'s paper, we define the recurrent patients within 24 months after surgery as the early recurrences ($N = 30$), and the non-recurrent patients longer than 60 months after surgery as late recurrences ($N = 22$). Two reference classifiers for recurrence prediction were also built based on the quantified peptides or proteins. We used the same workflow for a fair evaluation of the three classifiers based on MS features, peptides and proteins, respectively: (1) Perform minimum Redundancy Maximum Relevance (mRMR) algorithm[25] between the early recurrence group and late recurrence group. (2) Choose the top 200 MS features, peptides, and proteins with the highest mRMR scores as input to train an SVM classifier, respectively. All the parameters in SVM were set by default using the scikit-learn package (kernel='rbf', gamma='auto', C = 1). As shown in Fig. 5b, the five-fold cross validation of the top 200 aligned features-based classifier on the training set (C1, $N = 52$) shows

the highest AUC (0.998) compared with those classifiers built on top 200 peptides (AUC: 0.931) and top 200 proteins (AUC: 0.757). Please note that in the 101 patients from dataset HCC-T, the cases of early recurrence ($N = 30$), and the cases of late recurrence ($N = 22$) were considered as the training set (C1) for HCC early recurrence prediction.

Next, we mapped the top 200 MS features, peptides and proteins selected by mRMR to a test set HCC-R (C2, $N = 11$), respectively. There were 15 features, 55 peptides and 56 proteins successfully mapped in HCC-R. Then, the mapped 15 features were used to train a SVM classifier on the HCC-T dataset. To keep the same criteria, the top 15 of the mapped peptides and proteins were used to train the corresponding SVM classifiers in the same way, respectively. Notably, the feature-based classifier achieved better results than peptide- and protein-based classifiers on HCC-R (Fig. 5c). We tried to map all the 15 MS features in the feature-based early recurrence classifier to the corresponding identification results. There were eight features successfully mapped to the identification results of the HCC-T dataset. Seven features left remained unknown (Supplementary Data 1).

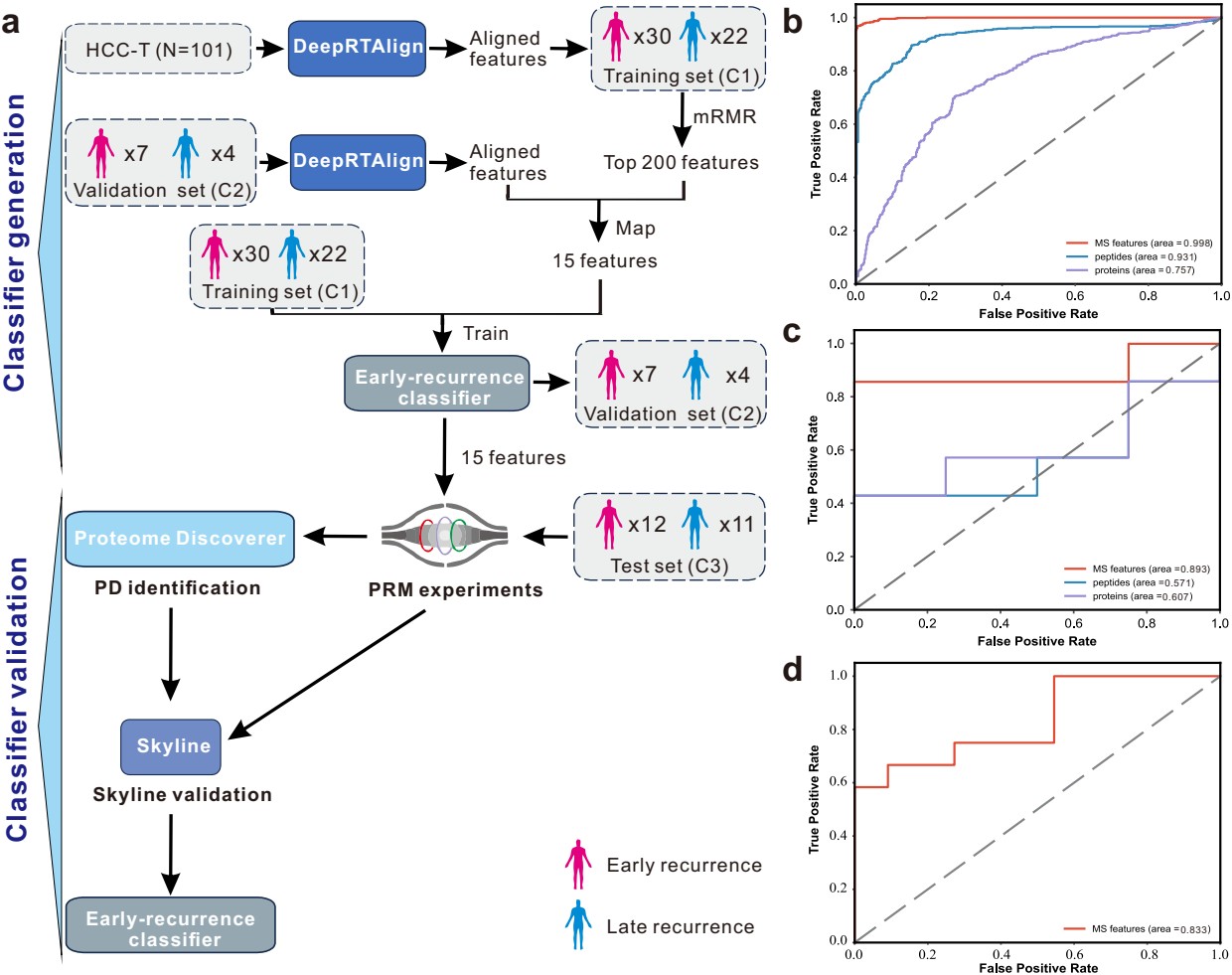

**Fig. 5 | Construction and validation of an HCC early recurrence classifier based on the MS features aligned by DeepRTAlign. a** Study design for construction and validation of this 15-feature-based classifier. First, all the features in dataset HCC-T ($N = 101$) were aligned by DeepRTAlign. Top 200 features were selected by mMRM. We mapped the top 200 MS features to an independent test set HCC-R (C2). There were 15 features successfully mapped to HCC-R. In the 101 patients from dataset HCC-T, the cases of early recurrence ($N = 30$), and the cases of late recurrence ($N = 22$) were considered as the training set (C1) for HCC early recurrence prediction. Then, the mapped 15 features were used to train a SVM classifier based on the C1 dataset, and tested on the HCC-R dataset (C2, $N = 11$). Peptide- and protein-based classifiers were generated in the same way, but they were not marked in Fig. 5a for

clarity. These 15 features in the early recurrence classifier were then targeted in PRM experiments and analyzed by PD and Skyline based on an independent dataset HCC-R2 (C3, $N = 23$). The 15-feature-based classifier was further tested on dataset HCC-R2 (C3) using the Skyline quantification results. **b** The five-fold cross validation AUCs on C1 of the SVM models for HCC recurrence prediction trained by the top 200 aligned MS features, top 200 peptides, and top 200 proteins, respectively. Top 200 peptides and top 200 proteins were also selected by mRMR. **c** The test AUCs on C2 of the SVM models trained on HCC-T dataset by the aligned 15 MS features, top 15 peptides, and top 15 proteins, respectively. **d** The test AUC on C3 of the 15-feature-based SVM model. Source data are provided as a Source Data file.

To further validate the 15 MS features in the feature-based early recurrence classifier, we enrolled a new HCC cohort HCC-R2 (C3, $N = 23$, Supplementary Data 2) and performed scheduled PRM (sPRM)[26] experiments for validation. Those 23 HCC patient samples were not fractionated, allowing us to detect all targeted features in a single run. The 15 features and the other 34 features from the top 200 features identified from our training set (HCC-T) were tentatively included in the target list of our sPRM method (Supplementary Data 3). Remarkably, all the 15 features were reliably identified in Proteome Discoverer (PD) results and verified in Skyline results (Supplementary Data 4 and Supplementary Data 5), including the seven previously unknown features in the HCC-T dataset (Supplementary Data 1).

Then, we performed an independent test using the sPRM results of the 15 features on the HCC-R2 cohort ($N = 23$, 12 cases of early recurrence and 11 cases of late recurrence). Peak areas of precursors and transitions were exported from Skyline (Supplementary Data 6) and summed as peptide abundance for the 15 features. As shown in Fig. 5d, the classifier to predict HCC early recurrence achieved an AUC value of 0.833 in this independent cohort (HCC-R2), indicating the generalizability of the 15-feature-based early recurrence classifier. These results indicate that the aligned MS features may contain more hidden information than peptides and proteins and thus enable more accurate stratification of the patients, showing the advantage of the ID-free alignment.

## Discussion

We present a deep learning-based tool DeepRTAlign for RT alignment in large cohort proteomic and metabolomic data analysis. DeepRTAlign is based on the basic information of MS spectra (m/z, RT and intensity), which can be applied to all the precursor ions in MS data before identification. We have demonstrated that DeepRTAlign outperformed other existing alignment tools by aligning more corresponding features without compromising the quantification precision and determined its generalizability boundary on multiple proteomic and metabolomic datasets. DeepRTAlign is flexible and robust with different feature extraction tools, which can help researchers achieve accurate and reproducible research data. While, since alignment has less influence on quantification than feature extraction in theory, we will try to improve the quantification accuracy by optimizing feature extraction and feature alignment simultaneously in the future work.

Finally, we applied DeepRTAlign to HCC early recurrence prediction as a real-world example. The results showed that aligned MS features have more effective information compared with peptides and proteins. DeepRTAlign is expected to be useful in finding low-abundant biomarkers, which usually only have low-quality MS/MS spectra, and may play a key role in proteomics-driven precision medicine.

## Methods
### Ethical statement
HCC tissues in HCC-R2 were obtained from patients underwent a curative-intent liver resection at the Eastern Hepatobiliary Surgery Hospital in Shanghai between 2012 and 2014. Written informed consent was obtained from all patients. The study was approved by the ethics committee of the Eastern Hepatobiliary Surgery Hospital in Shanghai.

### Datasets
As shown in Supplementary Table 1, all the datasets used in training and testing the deep learning model in DeepRTAlign (i.e., one training set and seven test sets) were collected from the following papers: (1) The training set HCC-T was from the tumor samples of 101 early-stage hepatocellular carcinoma (HCC) patients in Jiang et al.'s paper[24]. The corresponding non-tumor dataset (HCC-N) was considered as a test set

in this study. (2) Dataset HCC-R was from the tumor samples of 11 HCC patients who had undergone liver transplantations in ref. 27. (3) Datasets UPS2-M and UPS2-Y were from ref. 28, and were constructed by spiking 48 UPS2 standard proteins (Proteomics Dynamic Range Standard, Sigma-Aldrich) into mouse cell digestion mixture (UPS2-M) and yeast digestion mixture (UPS2-Y), respectively. The detailed experimental design is shown in Supplementary Fig. 8. (4) Dataset EC-H was from ref. 29, which was constructed by spiking E. coli cell digestion mixture into human cell digestion mixture. (5) Dataset AT was based on the *Arabidopsis thaliana* seeds (AT) from ref. 30. (6) Dataset SC was a single-cell (SC) proteomic dataset including 18 HT22 cells without the treatment of nocodazole from ref. 31.

Further, we tested the generalizability boundary of DeepRTAlign on the seven test sets and seven other datasets (five metabolomic datasets and two proteomic datasets). The five metabolomic datasets include (1) Dataset NCC19, a large-scale plasma analysis of SARS-CoV-2 from ref. 32; (2) Dataset SM1100, comprised standard mixtures consisting of 1100 compounds, from ref. 19; (3) Dataset MM, which was from the *Mus musculus* samples in ref. 33; (4) Dataset SO, which was from soil samples in ref. 34; (5) Dataset GUS, which was about global untargeted serum metabolomic samples from ref. 35. The two proteomic datasets are (1) Dataset MI, which was about mouse intestinal proteomes, from ref. 36; (2) Dataset CD, which was obtained from the gut microbiota of patients with Crohn's disease in ref. 37.

In this work, we generated six additional datasets (Supplementary Table 1): Dataset HCC-R2 was generated from the tumor samples of 23 HCC patients using parallel reaction monitoring (PRM). It was used to validate the 15-feature-based classifier for predicting HCC early recurrence (Methods). Dataset Benchmark-FC was a mixture of Human embryonic kidney (HEK) 293 T samples and E. coli samples with known fold changes. In this dataset, 10 ng, 15 ng, 20 ng, and 25 ng of E. coli samples were spiked into 200 ng of HEK 293 T samples, respectively. Each group contained 3 replicates. It was used for quantification accuracy evaluation. Dataset Benchmark-QC-H consisted of pure HEK 293 T with three replicates, and dataset Benchmark-QC-E consisted of pure E. coli with three replicates. Both datasets were used to evaluate the QC module in DeepRTAlign. Dataset Benchmark-RT contained two HEK 293 T samples with different RT gradients (60 min and 120 min), which were used for evaluating the performance of DeepRTAlign on different LC conditions. Benchmark-MV was a benchmark dataset containing different proportions of HEK 293 T and E. coli samples from six Orbitrap Exploris 480 instruments.

### Proteomic sample preparation
HCC tissues in HCC-R2 were obtained from patients underwent a curative-intent liver resection at the Eastern Hepatobiliary Surgery Hospital in Shanghai between 2012 and 2014. Written informed consent was obtained from all patients. The study was approved by the ethics committee of the Eastern Hepatobiliary Surgery Hospital in Shanghai. Filter-aided sample preparation was employed for tissue sample pretreatment. A 500 µg aliquot of liver proteins was diluted to 500 µL with a UA solution containing 8 M urea in 100 mM Tris-HCl (pH 8.5). Following centrifugation on a 30-kDa filter for 20 min, 200 µL of UA solution containing 10 mM DTT was added and the reaction was allowed to proceed at 37 °C for 4 h. After removal of the solution, a UA solution containing 50 mM iodoacetamide (IAA) was introduced and let react in the dark at room temperature for 30 min. The ultra-fraction tubes were then washed thoroughly with 200 µL of UA three times and 200 µL of 50 mM ammonium bicarbonate (ABC) three times. Next, 100 µL of ABC containing 0.1 µg/µL trypsin was added and let incubate at 37 °C for 12 h. The filter tubes were then washed twice with 100 µL of ABC via centrifugation, and flow-through fractions were pooled. Peptide concentration was determined using a NanoDrop One at 280 nm. Peptide mixtures were dried in a SpeedVac and stored at −80 °C before use.

HEK 293 T cells (National Infrastructure Cell Line Resource) were cultured and harvested according to established protocols[38]: HEK 293 T cells were cultured in Dulbecco's Modified Eagle Medium medium supplemented with 10% (v/v) fetal bovine serum and Penicillin/ Streptomycin (1:1000) under conditions of 37 °C and 5% CO_2. HEK 293 T cells were harvested post-treatment with trypsin and subjected to multiple washing procedures with 1X phosphate buffered saline. E. coli (DH5a) cell lysate was purchased from MCLAB (ECCL-100). Both HEK 293 T cells and E. coli cell lysate were then treated by the same in-solution digestion method to obtain peptides. First, lysis buffer containing 10% sodium deoxycholate (DOC), 1 M tris HCl (pH = 8.8), 100 mM tris (2-carboxyethyl) phosphine (TCEP) and 400 mM 2-Chloroacetamide (CAA) was added to the solutions. Solutions were heated at 95 °C for 5 min on a ThermoMixer (Eppendorf) and cooled down on ice for 5 min. Next, solutions were sonicated (Ningbo scientz, 25% energy, 30 s) on ice-water. Subsequently, proteins were digested overnight with 1:25 (w/w) MS-grade trypsin (Promega, V5280) at 37 °C. Finally, digested peptides were desalted using C18 solid-phase extraction column (Waters), dried in SpeedVac, and stored at −20 °C before nanoLC-MS analysis. The concentration of the peptides was determined using a NanoDrop One at a wavelength of 280 nm.

### Targeted LC-MS/MS analysis (dataset HCC-R2)

NanoLC-MS experiments were performed using Vanquish Neo system interfaced with an Orbitrap Exploris 480 mass spectrometer (Thermo Fisher Scientific, USA) operated in scheduled parallel-reaction-monitoring (sPRM). Peptide samples were separated on homemade capillary columns (100 μm i.d. x 30 cm) with integrated spray tips. The columns were packed with 1.9 μm/120 Å ReproSil-Pur C18 resins (Dr. Maisch GmbH, Germany), and all columns were heated at 55 °C. In sPRM mode, 500 ng of HCC peptide sample was injected each time, and a bit 1x iRT kit (Biognosys) was spiked in each sample for retention time calibration in Skyline analysis.

Mobile phases A and B were water and 80%/20% acetonitrile (ACN)/water (v/v) with 0.1% FA (v/v), respectively. The flow rate was 300 nL/min across the total gradients. The segmented 190-min gradient used in PRM mode is the following: 3%–8% (v/v) buffer B in 4.5 min, 8%–30% (v/v) buffer B for 168 min, 30%–40% (v/v) buffer B for 7.5 min, followed by a 2 min wash from 40% to 95% (v/v) buffer B. In the end, 95% (v/v) buffer B was kept for 8 min. After the gradient, a 20 min wash from 3% to 95% (v/v) buffer B was performed to minimize potential carryover. Scheduled PRM parameters were as follows: Full scan (MS1) from m/z 335 to 1 120 was acquired at the resolution of 120, 000. The automatic gain control (AGC) target was 1E6, and the maximum injection time (maxIT) was set to 200 ms. A subset of the top 200 MS features, the 15 features and other 34 features (a total of 49 features), were selected as targeted precursors in the mass list table (Supplementary Data 3) for further MS/MS scans. Targeted precursors were isolated through a window of 1.2 Th. The MS2 scans were acquired at a resolution of 120, 000 with an AGC setting of 4E5, a maxIT of 200 ms, and a normalized collision energy (NCE) of 28%.

### Nano-flow LC-MS/MS analysis of HEK 293 T and E. coli mixtures (dataset Benchmark-FC)

The Vanquish Neo system, Orbitrap Exploris 480 mass spectrometer, mobile phases, and capillary columns were the same as described above for targeted LC-MS/MS analysis. The flow rate was 300 nL/min, except that 500 nL/min was applied in the first 1.5 min of the total gradient. The gradient used in nano-flow LC-MS/MS analysis of HEK 293 T and E.coli mixtures is: 3%–8% (v/v) buffer B in 1.5 min, 8%–30% (v/v) buffer B for 55 min, 30%–40% (v/v) buffer B for 3.5 min, followed by a 1.5 min wash from 40% to 95% (v/v) buffer B, and 95% (v/v) buffer B was kept for 18.5 min at the end. DDA parameters were as follows: Full MS scans over the m/z range of 350–1500 were performed at a resolution of 60,000. The AGC target was set to 3E6, and the maximum injection

time was 45 ms. MS/MS acquisition was performed in top speed mode with a 1.5-s cycle time. The resolved fragments were scanned at a mass resolution of 15,000 and an AGC target value of 3E5. The threshold to trigger MS2 scans was 5E3, and the maxIT was 22 ms. Ions with charge states of 2–6 were sequentially fragmented by higher-energy collision dissociation (HCD) with an NCE of 28%. The isolation window was 1.6 Th, and the dynamic exclusion time was set to 40 s. 200 ng of HEK 293 T peptides spiked with 10, 15, 20, or 25 ng of E. coli peptides was loaded in each run, and 0.5 μL of 2.5x iRT kit was spiked in each sample. Each group contains three technical replicates.

### Micro-flow LC-MS/MS analysis of pure HEK 293 T or E. coli (datasets Benchmark-QC-H and Benchmark-QC-E)

A Dionex UltiMate 3000 Rapid Separation LC (RSLC) system was online coupled to an Orbitrap Exploris 480 mass spectrometer (Thermo Fisher Scientific) with an OptaMax NG Atmospheric Pressure Ionization Source (H-ESI mode). Commercially available Thermo Fisher Scientific Acclaim PepMap 100 C18 LC columns (1 mm ID × 150 mm, 2 μm particle size, catalog number 164711) were used for peptide separations. The column temperature was maintained at 55 °C using the integrated column oven in the LC system. Flow rate of 50 μL/min was used to deliver the segmented gradient, and mobile phases A and B were water and ACN with 0.1% FA (v/v), respectively. The segmented gradient is: 0.5%–5% (v/v) buffer B in 0.5 min, 5%–32% (v/v) buffer B for 60 min, 32%–95% (v/v) buffer B for 0.2 min. In the end, 95% (v/v) buffer B was kept for 2.5 min. After the gradient, columns were equilibrated 0.5% B for 1.8 min before the next injection. DDA data were acquired using the following parameters: Full scans were acquired in Orbitrap at a resolution of 60,000 (m/z 200) and AGC value of 3E6. The isolation window was 1.3 Th and m/z range was 350–1550 in full scans. The top 30 precursors found in full scans were selected for fragmentation. MS/ MS scans were acquired with 28% normalized collision energy in HCD mode at a resolution of 15,000 (m/z 200), charge states of 2–6 and minimum intensity of 1000. AGC target value for fragment spectra was set to 1.5E5. For MS2 spectra, the maxIT was set to 30 ms, and dynamic exclusion time was set to 30 s. 10 μg of HEK 293 T or E. coli tryptic peptides were loaded each time and the DDA data were collected from three Orbitrap Exploris 480 mass spectrometers with the same settings.

### Micro-flow LC-MS/MS analysis of HEK 293 T and E. coli mixtures (dataset Benchmark-MV)

The LC and MS parameters were the same as described in the above section "Micro-flow LC-MS/MS analysis of pure HEK 293 T or E. coli". Four samples were prepared: HEK 293 T peptides (73%, 91%, 97% and 99%) offset by varied proportions (27%, 9%, 3% and 1%) of E. coli peptides. 10 μg of HEK 293 T and E. coli peptide mixtures were loaded each time, and the DDA data were collected for each of the four samples from six Orbitrap Exploris 480 mass spectrometers with the same settings.

### Nano-flow LC-MS/MS analysis of HEK 293 T digest (dataset Benchmark-RT)

The Dionex UltiMate 3000 system, Orbitrap Exploris 480 mass spectrometer and mobile phases, were the same as described above for micro-flow LC-MS/MS analysis but operated in nano-flow mode. DDA parameters were the same as described in Nano-flow LC-MS/MS analysis of HEK 293 T and E. coli mixtures except that NCE was set to 30% instead of 28%. Homemade capillary columns (100 μm i.d. x 20 cm) with integrated spray tips were applied for LC separation. Two different gradients (60 and 120 min active gradients) were used with flow rate of 250 nL/min. Samples were loaded into capillary columns in 15 min at 700 nL/min, followed by flow rate decreased to 250 nL/min in 4.5 min. The active 60 min gradient is: 0.5%–6.4% (v/v) buffer B in 2 min, 6.4%–24% (v/v) buffer B for 55 min, 24%–32% (v/v) buffer B for 3.5 min. In the end, 32% (v/v) buffer B was increased to 95% in

1.5 min and kept for 5 min. The active 120 min gradient is the same as the active 60 min gradient, except that 6.4% buffer B increased to 24% (v/v) in 115 min. 1 μg of HEK 293 T digest was injected in each run.

## PRM data analysis

PRM raw data were firstly searched against the human UniProt FASTA database (downloaded on September 27, 2018) using PD 2.5 software with the Sequest HT search engine to identify peptide sequences. Trypsin was selected as the proteolytic enzyme, and missed cleavage sites were allowed up to two. Cysteine carbamidomethylation was set as the static modification. The oxidation of M and acetylation of the protein N-terminal were set as the dynamic modifications. The precursor mass tolerance was set to 10 ppm, and the fragment mass tolerance was 0.02 Da. The FDRs of the peptide-spectrum matches (PSMs) and proteins were set to less than 1%.

PRM data files were further analyzed using Skyline v22.2. A merged spectral library was generated in Skyline from MSF files of PD results and used as the reference library. An iRT database was generated along with the merged library for RT prediction, and peptide peaks were filtered to within 5 min of the predicted RT. The digestion enzyme was set to trypsin. Precursor charges 2 to 7 and ion charges 1 to 3 were allowed. Ion match tolerance was set to 0.02 Th for the selection of fragment ions from the spectral library. Raw data files were imported into Skyline for automated peak detection using "targeted" MS/MS filtering mode with the mass analyzer set to Orbitrap. For those identified 52 precursors from 48 MS features, auto-picked peaks in Skyline were further filtered manually to fulfill the following criteria: dotp values > 0.7, mass error within 5 ppm, detection of at least 5 fragments, and consistent RT across all 23 HCC samples. Peptides in a few samples that did not meet these criteria were considered undetected, and their peak areas in those samples were assigned a value of 0. Peptide abundance values of the 15 features were exported from Skyline. Peptide abundance was obtained by summing the peak area of both precursor ions and selected fragment ions.

## DDA data analysis

MSFragger v3.7 (in FragPipe v19.1), Mascot v2.8.1, MaxQuant v1.6, and PD 2.5 were used for DDA data analysis. HCC-T, HCC-N and HCC-R datasets were searched against the human UniProt FASTA database (downloaded on September 27, 2018). UPS2-M dataset was searched against the mouse UniProt FASTA database (downloaded on November 4, 2015). UPS2-Y dataset was searched against the yeast UniProt FASTA database (downloaded on June 15, 2021). EC-H dataset was searched against the FASTA database provided in PRIDE (https://www.ebi.ac.uk/pride/archive/projects/PXD003881). AT dataset was searched against the mouse-ear cress UniProt FASTA database (downloaded on August 2, 2021).

In MSFragger, the digestion enzyme was set to trypsin. Precursor mass tolerance and fragment mass tolerance were 20 ppm, peptide length was limited to 7–50. In MaxQuant, precursor mass tolerance and fragment mass tolerance are 20 ppm, min peptide length is 7. In Mascot, precursor mass tolerance is 20 ppm, and fragment mass tolerance is 0.05 Da. For database searching of HEK 293 T DDA data collected in two gradients, PD 2.5 settings are the same as used for sPRM data. For all the tools, fixed and variable modifications, digestion enzyme, and FDR rate were the same as described above for PD 2.5 settings of sPRM data. The other parameters were set to default in these tools.

## DIA data analysis

DIA-NN v1.8 was used for DIA data analysis. Dataset SC were searched against the FASTA database provided in PRIDE (https://www.ebi.ac.uk/pride/archive/projects/PXD025634). The digestion enzyme was set to trypsin, missed cleavage was 1, peptide length range was 7–30, and precursor charge range was 1–4. Cysteine carbamidomethylation was set as the fixed modification, while the protein N-terminal were set as

the variable modification. The other parameters were the default settings in DIA-NN.

## Machine learning models for evaluation

To make a systematical evaluation of our DNN model's performance, we compared it with several popular machine learning methods, i.e., random forests (RF), k-nearest neighbors (KNN), support vector machine (SVM) and logistic regression (LR). When compared DNN with these machine learning methods, the inputs were exactly the same, i.e., the features after coarse alignment step. The parameters for each machine learning method were optimized based on the 10-fold cross validation results of the training set HCC-T (Supplementary Tables 9, 10 and Supplementary Data 7, 8). The parameters of the RF model are number of estimators 50, and max depth 50. For KNN, the k is set to 5. For SVM, the kernel function is set to 'poly' (gamma = 'auto', C = 1, and degree = 3). For LR, the penalty is L2 and the solver is 'lbfgs'. All the other parameters are default values in scikit-learn v0.21.3. In total, we trained four machine learning models (named as RF, KNN, SVM and LR) as references in this study.

## Tools for alignment comparison

As shown in Supplementary Table 7, the existing alignment tools can be classified into three types based on the input information required. The representative tools in each type were compared with DeepRTAlign in this study.

First, two existing popular alignment tools (MZmine 2 and OpenMS) were used for alignment comparison because these two tools showed the best precision and recall in refs. [11], [39]. The recommended parameters in the official user manuals of MZmine 2 and OpenMS were used. For MZmine 2, the ADAP-LC workflow was used. For OpenMS, we used FeatureFinderCentroided for feature extraction and MapAlignerPoseClustering for feature alignment. After alignment, we used FeatureLinkerUnlabeled and TextExporter to obtain the results. When comparing with MZmine 2 or OpenMS, we considered the Mascot identification results corresponding to MZmine 2 or OpenMS features (mass tolerance: ±10 ppm, restrict the RT of a peptide to be within the RT range of the corresponding precursor feature) as the ground truth, respectively.

The precision formula is:

$$\text{Precision} = \frac{1}{N} \sum_{k=1}^{N} \frac{|A_k \cap G_k|}{|A_k|} \tag{1}$$

The recall formula is:

$$\text{Recall} = \frac{1}{N} \sum_{k=1}^{N} \frac{|A_k \cap G_k|}{|G_k|} \tag{2}$$

N is the sample pair number. $A_k$ is a set containing all the aligned feature pairs in the $k_{th}$ sample pair. $G_k$ is a set containing all the ground truth of the $k_{th}$ sample pair. $|X|$ indicates the number of elements in the set X.

Second, Quandenser[20], an alignment method that requires MS/MS information, was also compared with DeepRTAlign on dataset Benchmark-FC. The peptide identification results (MSFragger) were considered as our ground truth.

Third, we compared DeepRTAlign with MaxQuant, MSFragger and DIA-NN for DDA and DIA data analysis, respectively. For DDA data (Benchmark-FC), the peptide identification results (MaxQuant or MSFragger) were considered as the ground truth. For DIA data, we used a single-cell proteomic dataset obtained from 18 HT22 cells without nocodazole treated in ref. [31]. All the parameters were kept the same as described in Li et al. We used Dinosaur to extract MS features in each cell and DeepRTAlign to align the features in all 18 cells, compared with the aligned results of DIA-NN with and without MBR function.

## Dataset simulation for generalizability evaluation

We further generated 24 simulated datasets with different RT shifts (considered as noise) for each real-world dataset. Here is the dataset simulation procedure. (1) All the features are extracted in each dataset using OpenMS to form an original feature list. Features with charges 2–6 are considered for proteomic data, and features with charges 1–6 are considered for metabolomic data. (2) An RT shift based on a normal distribution with increasing standard deviations ($\sigma$ = 0, 0.1, 0.3, 0.5, 0.7, 1, 3, 5) for each mean value ($\mu$ = 0, 5 and 10 min) are added in each feature to form a new feature list by modifying the featureXML file generated by OpenMS. (3) The new feature list with artificial RT shifts is aligned to the original feature list by DeepRTAlign and OpenMS. In theory, each feature with an RT shift in the new feature list should be aligned with the same feature in the original list. (4) The precision and recall values were calculated to evaluate the generalizability boundary of DeepRTAlign.

## Reporting summary

Further information on research design is available in the Nature Portfolio Reporting Summary linked to this article.

## Data availability

Datasets HCC-T and HCC-N can be downloaded from the iProX database[40] under accession number IPX0000937000 or PXD006512. Datasets UPS2-M and UPS2-Y can be downloaded from the iProX database under the accession number IPX00075500 or PXD008428. Datasets HCC-R, EC-H, AT, SC, MI and CD can be downloaded from the PRIDE database[41] under the accession numbers PXD022881, PXD003881, PXD027546, PXD025634, PXD002838 and PXD002882, respectively. Datasets NCC19, SM1100, MM, SO and GUS can be downloaded from the MetaboLights database[42] under the accession numbers MTBLS1866, MTBLS733, MTBLS5430, MTBLS492 and MTBLS650, respectively. Datasets HCC-R2, Benchmark-FC, Benchmark-QC-H, Benchmark-QC-E, Benchmark-RT and Benchmark-MV can be downloaded from the iProX database under the accession numbers IPX0006622000, IPX0006638000, IPX0006819000, IPX0006819000, IPX0006820000 and IPX0007319000, respectively. All other relevant data supporting the key findings of this study are available within the article or the Supplementary Information files. Source data are provided with this paper.

## Code availability

All the codes are programmed in Python v3.7.4. Numpy v1.16.4 is used to preprocess data. SciPy v1.6.2 is used for statistical methods. Pytorch framework v1.8.0 is used to implement the DNN model, and scikit-learn framework v0.21.3 is used to implement other machine learning algorithms. Mrmr-selection v0.2.3 is used for mRMR algorithm. All the source codes of DeepRTAlign (including the feature extraction tool XICFinder) are freely available from GitHub (https://github.com/PHOENIXcenter/deeprtalign) under GNU General Public License version 3.0. Other tools used in the study are MSFragger v3.7 (in FragPipe v19.1), Mascot v2.8.1, MaxQuant v1.6, MZmine 2 v2.53, OpenMS v2.6.0, Quandenser v0.03. DeepRTAlign is freely available on Zenodo (https://zenodo.org/record/10140300)[43].

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

## Acknowledgements

We thank Dr. Jing Yang, Dr. Mingchao Wang, Dr. Chaoying Li and Dr. Chao Liu for the helpful suggestions and comments. Y.L., Y.Y., F.H., Y.Z. and C.C. are financially supported by the National Key Research and Development Program of China (2021YFA1301603 and 2020YFE0202200) and the National Natural Science Foundation of China (32088101). F.H. and C.C. also acknowledge funding support from the CAMS Innovation Fund for Medical Sciences (CIFMS) (2019-I2M-5-063).

## Author contributions

Y.Z. and C.C. designed the study and co-supervised the project. Y.L. designed the architecture of DeepRTAlign, performed the experiments, analyzed the results. Y.Y. designed and performed the PRM experiments and analyzed the PRM data. Y.L., Y.Y. and C.C. wrote the manuscript. F.S. enrolled the new HCC cohort ($N = 23$) for PRM validation and provided the necessary clinical information. W.C. generated the benchmark dataset with known fold changes. Yingying.Z. and Yuanjun.Z. collected the public datasets. L.X. helped to revise the manuscript. F.H. coordinated the study. All authors helped to revise the manuscript and approved the final manuscript.

## Competing interests

The authors declare no competing interests.
