## [Peer Review File · Nature Communications]

Reviewers' Comments:

Reviewer #1:

Remarks to the Author:

The authors presented an "ion-centric" alignment method based on neural networks. It got potential, but there are several issues to be addressed.

Overall, I think this paper focused too much on software comparisons with numbers, whereas its own features were rarely discussed.

Major:

1. The key selling point of this work is the "ion-centric" or ID-free alignment. However, authors did not discuss much on why we need "ion-centric" methods. I only found one in Supp. Note, but this is not convincing enough. As a suggestion, this method may be able to detect more regulated but mis-identified features in addition to the original HCC studies. To verify this, authors can use PRM-like methods to check whether the ion-centric finding makes sense or not. If this is proven to work well, it will be a great complementary method to existing MBR-based quantification workflow.

2. Regrading software comparisons, as the quality control module is missing in DeepRTAlign, I think the comparisons are somewhat unfair to other tools that have such modules.

3. For the DNN model, the authors only used (RT, intensity) pairs as the input model features. However, for robust signal matching, peak shapes should be considered as well. Otherwise, if two co-eluted features share the same m/z, it will be hard for the model to recognize which is the correct one. And according to the input vector design (part 2 and part 3), I guess the signal with closer intensity will "win". If so, this will lose regulated MS features. Therefore, I highly suggest authors to test the robustness of this method against co-elution.

4. Will mass errors between MS feature pairs as a DNN feature be helpful in the model? In principle, mass errors are very useful, as in MS1-only methods, mass is the key information we can use to distinguish different signal species. And maybe mass recalibration is also required for different runs.

5. Line 235-248: The RT shift simulation is not sufficient. What I also expected is if adding some (n=5?) pseudo features to the left RT or right RT of the target features with minor intensity differences, is the algorithm robust enough to get the correct features back?

6. Line 386 and Figure S5: In single-cell data, it is not surprising that MS1-based methods get more peptides than MS2-based ones, as orbitrap is not sensitive enough to detect very low-abundant fragment signals at single-cell level. What concerns me is what peptides were aligned by DeepRTAlign but not DiaNN? Please check both of peptides' MS1 and MS2 signals. As there are only 298 peptides, I suggested to check them all on MS1 and MS2 matches, and show the consistency of chromatographical profiles of both the MS1 and MS2 peaks.

7. Figure 1: Triangle and square representation does not make it easier to understand the Input Vector, especially for adjacent features. A better Illustration is required.

8. Line 158-159: More explanation is needed about why using two adjacent features as the input vector. What is the idea behind it?

9. In Figure 2 to 4, it is better to show some PSM or XIC examples illustrating how DeepRTAlign can get MS1 features back by alignment, in addition to number comparisons.

10. It is better to test this tool on quantification benchmark datasets such as LFQ-bench with known fold changes.

Minor:

1. I suggest to rename "ion-centric" as it is not specific, identification-based methods also use ions.
2. Line 143-144: "The precursor ions with more than one ..." is hard to understand. What does it mean "the precursor ions"? As a MS2-free method, why it needs "more than one MS2 spectrum"?
3. Line 154-155: What algorithm did you use for alignment? It should be mentioned here.
4. Table S1: As this is only supplementary info, please put more information in the table legend, such as full dataset description and so on. Otherwise, it is hard to understand the abbreviations such as SC, AT, etc.
5. It is better to test the alignment on samples with different LC conditions, such as 21 min gradient to 120 min gradient.

Reviewer #2:

Remarks to the Author:

Liu Yi and colleagues have presented a manuscript in which they use a deep neural network (DNN) for retention time alignment. Compared to commonly used methods, the authors claim to have generated superior peak-correspondence with MS1 features alone. They have compared the performance of DNN against four machine-learning methods, and other software packages such as MaxQuant-MBR, FragPipe-MBR, OpenMS, MZmine2. While the results suggest that the deep neural network performs better than the other methods, it should be noted that a standard quantitative approach for analyzing these tools is currently lacking. Further discussion on the strengths and limitations of the manuscript is provided below:

1. Comparison to other DNN: Authors have not clarified as to why the Siamese network by Li et al which has three encoders - the Mass Spectrum Encoder, Peak Encoder, and Chromatograph Encoder in addition to ΔRT , would fail. It is natural to assume that incorporating more information, such as chromatogram and MS/MS encoding, would lead to better performance. However, the authors did not clarify whether the nature of the data, in this case, GC-MS and LC-MS, could be a factor in this regard. Although the authors have provided an explanation in the Supplementary Notes, it is not entirely convincing.
2. Standard benchmarking is needed: The model presented by the authors explicitly uses intensity for peak-matching, which can introduce a conservative bias towards selecting features with similar intensity. Thus, the use of the coefficient of variation (CV) as a measure to assess the superiority of a tool is not entirely justifiable. It would be more appropriate for the authors to compare the performance of the different tools using manually annotated heterogeneous data or test them on dilution-series or synthetic-proteomics data to demonstrate that the intensity of aligned features is correlated with the spiked-in pattern. Currently, it appears that the method is vulnerable to producing good CV but may not accurately capture differential expression.
3. Have not elaborated upon the technical details about DNN: In the manuscripts authors have presented the results of various software tools, but have not critically analyzed them. Specifically, the authors have not explored in detail the specific properties of the deep neural network (DNN) that make it superior to other tools, such as MZmine2, OpenMS, or MBR. It is essential to investigate what kind of nonlinearities the DNN captures that cannot be identified by other methods. Further exploration of the results is necessary to draw definitive conclusions.
4. Quantitative analysis is lacking. Retention time alignment is typically carried out to enhance the quantitiveness of LC-MS/MS data, which should, in turn, enable the identification of novel biological insights. While the authors have presented some analysis for early recurrence prediction, the analysis is currently quite crude and lacks convincing evidence and is not convincing at all. They should provide an explanation and interpretation of differential features. For example, how many belong to identified peptides, how many are low-abundant and present how many have

reproducible MS/MS spectra across multiple samples.

5. In Table S4, authors have compared the DNN with other non-linear methods for alignment. The performance with logistic regression is very poor. I wonder what could be the reason for that? Additionally, EC-H has only 20 runs; for such small-datasets non-linear methods typically perform at over 90% accuracy, which is not observed in this case. The authors should provide a more detailed technical assessment of the specific objectives achieved by the DNN that are missed by other tools, particularly in the case of smaller datasets like EC-H. The specific details of what the DNN brings to the table that logistic regression, KNN, and SVM cannot achieve should also be elaborated upon.

6. Authors have compared their tool to XCMS, OpenMS which employ a monotonic warping function. They have also reported other direct-matching methods: RTAlign, MassUntangler and PeakMatch which they claim are not suitable, but their alignment philosophy is similar to that of DeepRTAlign. The authors should describe specifically what drawbacks of the direct-matching approaches are addressed and fixed in their approach. In addition, it would be beneficial for the authors to benchmark their method against other similar approaches, such as LWBmatch.

7. What is the output vector of DNN? Is it binary Aligned/not-aligned? What would it choose if the feature n is aligned to $m-1$, m and $m+1$, I am assuming that you slide over five features while aligning them?

8. The claim that MS1-based alignment can outperform approaches that use both MS1 and MS2 information is counterintuitive. For example, Quandenser uses decoys to control false mappings while utilizing MS/MS anchor peptides for aligning MS1 chromatograms. However, the manuscript lacks rigorous analysis to support this claim.

Line 120: Add a supplementary table like S1 for metabolomics and proteomics data.

Line 214: The RT tolerance for identification is too big (5 mins). It should be less than a minute. Please report precision and recall with updated RT tolerance.

Figure 3: Report median and standard deviation for each boxplot. Are the intensities normalized across runs before calculating CV?

Figure 4a. Why are the last two bars different for MSFragger no-aligned?

Figure 4: Explain what is unique peptides and shared peptides? Is common means identified in all runs 100%?

Reviewer #3:

Remarks to the Author:

General comment on the paper

Overall, the paper is well-written and structured. The authors thoroughly tested their proposed method, DeepRTAlign, and generously provided code for using the tool. However, there are some areas that could be further improved. At present, the selection of datasets does not make it clear if this method is a significant contribution to improving alignment (see M1 and M4), and some aspects and details of the approach are not clearly explained (see M2). Finally, code for the experiments is not available, which hinders reproducibility (see M3). Below, please find my (constructive) points for the authors to consider to further improve the paper.

Major comments

M1. Impact of the method

Although the authors performed an extensive benchmark of their method, IMHO it is still unclear if DeepRTAlign is a significant contribution to improve alignment. Currently, there is a bloat of very similar methods for alignment with essentially the same performance of the algorithms they pretend to improve. Part of the problem is that the authors of each method select their own evaluation data, which is not ideal since researchers may select (probably unconsciously) datasets where their methods perform better. To tackle this issue, the use of common benchmarks should be promoted. Therefore, I suggest including the datasets from [1] in the comparison. These new set of tests can also be used to include further information as suggested in later commentaries.

[1] Eva Lange, Ralf Tautenhahn, Steffen Neumann and Clemens Gröpl. "Critical assessment of alignment procedures for LC-MS proteomics and metabolomics measurements". BMC Bioinformatics 2008, 9:375

M2. DeepRTAlign description

Code inspection reveals that the current description of DeepRTAlign, although nicely written, can be further improved.

Training Part

* Some details are missing in the description of the DNN and its hyperparameters:

- The text does not mention the sigmoid activation.
- It should be specified which initialization scheme the authors used for the weights and biases.
- The authors should mention the optimizer used and, if it makes use of hyperparameters, these should be described.
- The authors should explain how they selected the number of epochs.

Application Part

* The code implementation loads a matrix from an npy file to normalize the

features feeding the neural network. However, this is not clearly described in the text.

* Starting at line 188, the manuscript states: "In this part, feature lists will first go through the coarse alignment step and the input vector construction step as same as those in the training part. Then, the constructed input vectors will be fed into the trained DNN model". However, the code reveals that between the coarse alignment and the predictions of the DNN there is a binning process based on m/z, which is not obvious from the text. I would suggest to explicitly describe this process on the manuscript to enhance clarity.

* Furthermore, the "bin_width" and "bin_precision" parameters involved in this step seem to have a large impact on the alignment process (based on my own experiments). Although this is somewhat expected by the meaning of the parameters, a brief discussion on how they should be chosen is suggested, due to the large impact of the results. If different values were used during tests, it should be reported on the manuscript.

M3. Code availability

The Github repository is well-structured and can be easily read. However, the authors only provide code for the "application part" of the tool. That is, the validation experiments can't be reproduced without substantial work by other researchers, which hinders reproducibility and comparison with future alignment methods. It would be nice if the authors share the whole codebase for reproducing all the tests conducted in the study. The code for the tests suggested in M1 should be shared.

M4. DeepRTAlign computational performance

The code provided by the author does a lot of disk operations, which may harm the computational running time of the algorithm. It is crucial that any new alignment method seeking to make a substantial contribution should be highly performant, as it must compete with the current state-of-the-art tools. Wall-clock runtime for the DeepRTAlign method should be provided for, at least, the benchmarks suggested in M1.

Minor comments (organized by sections)

Workflow of DeepRTAlign

* "All other parameters are kept by default in Pytorch". Default parameters can change (or even disappear!) as software evolves. I would prefer to read the details explained in a software-agnostic way. If not possible, the pytorch version should be stated here for the reader's convenience (although it is true that it appears later in the text).

Machine learning models for evaluation

* It should be clearly stated if the only difference between DeepRTAlign and the other methods is the substitution of the neuronal network by the proper classifier, or if there is any other further difference.

* A few words describing the parameter tuning process should be included.

* What is the difference between the number of estimators (100) and the tree number in the

forest (10)?

- * The authors should specify which non-linear kernel was used for the SVM. Furthermore, they should state whether the SVM's hyperparameters were tuned. If not, I would suggest optimizing them since SVMs are quite sensitive to the hyperparameters configuration.

Tools for alignment comparison

- * The software tools MZmine and OpenMS offer numerous alignment methods. However, to ensure transparency and reproducibility, the authors should clearly specify which methods they employed in their analysis.

- * The precision and recall formulas should be reviewed and better explained. It seems that A_k and G_k are sets, but this is not obvious and, in that case, the cardinality of the sets is missing from the formulas.

Model evaluation on the training set and the test sets

- * Line 281. "We think this is because the RT shift density distribution of UPS2-Y is similar to HCC-T (Figure S2)". Does "RT shift" refer to the coarse alignment step? If yes, why this explains the better performance of RF? Isn't the coarse alignment step shared between the DNN (DeepRTAlign) and the RF version?

- * Line 282. "In general, DNN has a better generalization performance and RF can be an alternative solution when computing resources are limited" Can the authors further explain why this occurs? Intuitively, in the application part, the prediction times for the DNN and the RF shouldn't be so different.

- * In Figure 3, I have trouble understanding "DeepRTAlign Shared" (SH_D) and "Quandenser Shared" (SH_Q). I thought "shared" likely referred to the peptides that were identified by both DeepRTAlign and Quandenser algorithms, but the boxplot shows different CVs for SH_D and SH_Q and hence my interpretation must be wrong.

Some typos in the text:

- * Use of two periods after citations. For example, at lines 109 or 118.

- * Line 182: " and the trained model was evaluated..." -> The trained model was also evaluated

- * Lines 187-188: "...can be used after format conversion refer to the formats.." -> ... can be used after format conversion. Refer to the formats ...

- * Line 354: "the CV values of DeepRTAlign in all the group are 47.6% smaller in UPS2-M and 58.3% smaller..." -> "the CV values of DeepRTAlign are 47.6% smaller in UPS2-M and 58.3% smaller..."

- * Line 361: "indentification" -> identification

- * Line 640: "the histograms of ...> -> the barplots of (I don't think they should be referred to as histograms). There are a few more occurrences of "histogram" after this first one.

Supplementary material

- * Figure S2. It is hard to compare the distributions of the datasets. Maybe a log scale could be used in the y-axis to enhance the visualization.

- * Table S10. How are the standard errors computed in Table S10? In all other tables, I suppose the standard errors are computed using 10-fold cross-validation but, in this real dataset, this does not seem to be the case. If 10-fold cross-validation is indeed used with real datasets, then standard errors should be incorporated to other results (for example, error bars could be added to figure 2).

Point-by-point response to reviewers' comments

We appreciate the reviewers' insightful and constrictive comments as they have helped us improve the manuscript. Additional experiments, data analysis and manuscript revision have been performed to address the reviewers' comments. In this response, the reviewers' comments are colored in black, and our replies to the comments are colored in blue.

Please note that to access the five newly generated datasets in this revision shown in **Supplementary Table 1**, we have uploaded all the corresponding raw data and identification results to iProX database (doi: 10.1093/nar/gky869). These datasets will be public once this manuscript is accepted. During peer review, reviewers can access these data by the following links with codes.

(1) dataset HCC-R2: IPX0006622000

URL: <https://www.iprox.cn/page/PSV023.html?url=1690814542091HKwZ>

Code: e31M

(2) dataset Benchmark-FC: IPX0006638000

URL: <https://www.iprox.cn/page/PSV023.html?url=1690815188443CVgA>

Code: j0Nz

(3) datasets Benchmark-QC-H and Benchmark-QC-E: IPX0006819000

URL: <https://www.iprox.cn/page/PSV023.html?url=1690814745537jgtH>

Code: ff3O

(4) dataset Benchmark-RT: IPX0006820000

URL: <https://www.iprox.cn/page/PSV023.html?url=1690814804447Woki>

Code: HP6c

Reviewer #1 (Remarks to the Author):

The authors presented an "ion-centric" alignment method based on neural networks. It got potential, but there are several issues to be addressed. Overall, I think this paper focused too much on software comparisons with numbers, whereas its own features were rarely discussed.

Response: We thank the reviewer for the valuable comments and for highlighting certain issues, which, once addressed, have further improved our manuscript.

Major:

Q1. The key selling point of this work is the "ion-centric" or ID-free alignment. However, authors did not discuss much on why we need "ion-centric" methods. I only found one in Supp. Note, but this is not convincing enough. As a suggestion, this method may be able to detect more regulated but mis-identified features in addition to the original HCC studies. To verify this, authors can use PRM-like methods to check whether the ion-centric finding makes sense or not. If this is proven to work well, it will be a great complementary method to existing MBR-based quantification workflow.

Response: We thank the reviewer for this constructive suggestion. We agree that the ID-free alignment algorithm is the key selling point of our work. Following Reviewer #1's suggestion in Q11, we changed "ion-centric" to "ID-free" to give a clearer description. We moved the detailed discussion on why we need ID-free methods from the **Supplementary Notes** to the main text. Furthermore, following the reviewer's suggestion, we enrolled a new HCC cohort (named HCC-R2, N=23) with necessary clinical information (**Supplementary Table 14**). We performed scheduled PRM (sPRM) experiments on this cohort to validate our original finding (the 15 features for predicting HCC early recurrence).

A subset of the top 200 MS features identified from our training set (HCC-T), including the 15 features used for early recurrence prediction and 34 other features were tentatively targeted in our sPRM experiments on an independent cohort. Remarkably, 48 out of the 49 target features (**Supplementary Table 15**) were identified by Proteome Discoverer (PD) results (**Supplementary Table 16**) and further verified in Skyline results (**Supplementary Table 17**) of the sPRM data. This confirms the reliability of our ID-free strategy.

Importantly, our sPRM experiments also verified the unidentified features in the 15 features discovered from the HCC-T dataset (**Fig. 5a**). Out of the previously identified eight features, seven unidentified features were verified in PD and Skyline results of the sPRM data (**Supplementary Table 13**). This is possibly because some low-quality peptide signals cannot be accurately identified in DDA data, whereas most of those features can be successfully identified in PRM data with high-quality spectra. Previously, the feature "1022.52_1022.55_1" from HCC-T results was assigned the sequence LSEDYGVLK, however, this sequence was not found in the PD or Skyline results of the sPRM dataset. Instead, TLLEDGTFK, which has a similar m/z and RT, was identified as the correct sequence for this feature according to sPRM data (**Supplementary Table 13**). Please refer to the Skyline annotated spectra for each feature in **Supplementary File 1** for details.

Then, we performed an independent test using the sPRM results of the 15-feature-based classifier on the new HCC cohort (N=23). As shown in **Fig. R1** (denoted as **Fig. 5d** in this revision), the classifier to predict HCC early recurrence achieved an AUC value of 0.833, indicating the generalizability of the 15-feature-based classifier for HCC early recurrence prediction. Therefore, our ID-free method may be able to detect more regulated but mis-identified features in addition to the original HCC studies and thus develop better classifiers.

Fig. 5a. Study design for construction and validation of this 15-feature-based classifier. First, all the features in dataset HCC-T (N=101) were aligned by DeepRTAlign. Top 200 features were selected by mMRM. We mapped the top 200 MS features to an independent test set HCC-R (C2). There were 15 features successfully mapped to HCC-R. In the 101 patients from dataset HCC-T, the cases of early recurrence (N=30), and the cases of late recurrence (N=22) were considered as the training set (C1) for HCC early recurrence prediction. Then, the mapped 15 features were used to train a SVM classifier based on the C1 dataset, and tested on the HCC-R dataset (C2, N=11).

Peptide- and protein-based classifiers were generated in the same way, but they were not marked in Fig.5a for clarity. The 15 features in the early recurrence classifier were then targeted in PRM experiments and analyzed by PD and Skyline based on an independent dataset HCC-R2 (C3, N=23). The 15-feature-based classifier was further tested on dataset HCC-R2 (C3) using the Skyline quantification results.

Fig. R1. The test AUC on PRM dataset HCC-R2 of the 15-feature-based classifier for HCC early recurrence prediction.

Overall, we concurred with the reviewer's opinion and validated our ID-free alignment algorithm by sPRM experiments. We believe that our strategy can serve as a valuable complementary method to the existing MBR-based quantification workflow.

Q2. Regarding software comparisons, as the quality control module is missing in DeepRTAlign, I think the comparisons are somewhat unfair to other tools that have such modules.

Response: We thank the reviewer for pointing out this important issue. Following this valuable suggestion, we designed and implemented a quality control (QC) module in DeepRTAlign. As shown in **Fig. R2** (denoted as **Supplementary Fig. 2** in this revision), in each m/z window, DeepRTAlign will randomly select a sample as a target and build its decoy. In theory, all features in the decoy sample should not be aligned. Based on this, the final FDR of the alignment results can be calculated. We have updated all the results in this revision using the new version of DeepRTAlign.

Fig. R2. Illustration of the QC module in DeepRTAlign. (a) The decoy design workflow. (b) The FDR calculation workflow.

Then, we tested the reliability of the QC module on two newly generated datasets consisting of the pure HEK 293T samples and pure *E. coli* samples, respectively (Benchmark-QC-H and Benchmark-QC-E). Both datasets have three technical replicates (See **Datasets** section and **Supplementary Table 1** for details). As shown in **Fig. R3**, we tried to align these two different datasets (HEK 293T and *E. coli*). In theory, there should be no features aligned for totally different species. We found that with an FDR cutoff of 0.01, no results were reported by DeepRTAlign for the three replicates in the two datasets (e.g., repeat 1 in 293T vs repeat 1 in *E. coli*), proving the reliability of our QC method.

Fig. R3. The experimental design to benchmark the QC module in DeepRTAlign. R1,

R2, and R3 indicate the three replicates in the dataset.

Q3. For the DNN model, the authors only used (RT, intensity) pairs as the input model features. However, for robust signal matching, peak shapes should be considered as well. Otherwise, if two co-eluted features share the same m/z, it will be hard for the model to recognize which is the correct one. And according to the input vector design (part 2 and part 3), I guess the signal with closer intensity will "win". If so, this will lose regulated MS features. Therefore, I highly suggest authors to test the robustness of this method against co-elution.

Response: We thank the reviewer for this valuable suggestion. We agree that if we only used RT and intensity pairs as input for training the alignment model shown in the original Fig. 1, we would miss the regulated MS features. Following Reviewer #1's Q3 and Q4 suggestions, we removed intensity from the model features but added m/z as a new model feature. Fig. 1 has been updated accordingly. The model of DeepRTAlign was re-trained using the new input model features.

Fig. 1. The illustration of DeepRTAlign algorithm. (a) The training procedures of DeepRTAlign. The n is the feature rank number (sorted by RT) in one m/z window of sample 1. The m is the feature rank number (sorted by RT) in one m/z window of sample

2. Please note that “n-1”, “n-2”, “n+1” and “n+2” represent four adjacent features of the feature n in one m/z window. (b) The workflow for RT alignment using DeepRTAlign. DeepRTAlign supports four feature extraction methods (Dinosaur, MaxQuant, OpenMS and XICFinder). The features extracted will be aligned using the trained model shown in (a), with the aligned feature list as the output.

Then, we performed a co-elution test on the AT dataset to benchmark DeepRTAlign with new input model features. First, we randomly selected a sample in the AT dataset and denoted it as Sample N. It contained 11,278 features. For each feature in Sample N, we added an RT shift of 5 minutes and thus generated a simulated sample named Sample N'. In theory, every feature in Sample N can be aligned to its corresponding feature in Sample N'. And as shown in **Fig. 4**, the precision and recall of DeepRTAlign are both 100% ($\mu = 5$ min, $\sigma = 0$). Second, for each feature in Sample N, its pseudo feature was generated as follows: a mass shift of 1, 3, and 5 ppm, respectively, was added to its m/z; a random RT shift based on a normal distribution (**Table R1**) was added to its RT. All the 11,278 pseudo features were added to Sample N and Sample N', respectively. Third, we used DeepRTAlign to align Sample N and Sample N' to test if similar pseudo features would influence DeepRTAlign's performance. As shown in **Table R1**, we found that the precision and recall of DeepRTAlign stayed unchanged (100%), indicating it was a robust alignment method against co-elution.

Table R1. Experiment design of the co-elution test and the corresponding precisions and recalls of DeepRTAlign.

m/z shift (ppm)	RT shift μ (min)	RT shift σ (min)	precision	recall
1	1	0.5	1	1
3	1	0.5	1	1
5	1	0.5	1	1
1	1	1	1	1
3	1	1	1	1
5	1	1	1	1
1	1	1.5	1	1
3	1	1.5	1	1
5	1	1.5	1	1

Finally, we performed a test for peak shape on dataset HCC-T. Since most feature extraction tools did not output the detailed information (intensity and RT) for each MS1

scan in a feature, we tried a simplified three-point method to represent the peak shape of a feature. For each feature, we used a triangle consisting of the RT and intensity from RT-Start, RT-Apex and RT-End to mimic its shape. We trained the DeepRTAlign model with the shape information using 400,000 feature pairs from HCC-T (200,000 positive and 200,000 negative pairs) mentioned in this revision's “**Workflow of DeepRTAlign**” section. This model is named DeepRTAlign-shape.

Then, we built a test set by randomly selecting 10,000 positive and 10,000 negative feature pairs from the HCC-N dataset and randomly generating 10,000 decoy feature pairs using the method shown in **Fig. R2**. It should be noted that the feature pairs collected from the same peptides are labeled as positive, and the feature pairs collected from different peptides (RT difference < 5 min) are labeled as negative. In theory, the output score of DeepRTAlign should have a discrimination power to separate the target features and the negative ones. Ideally, the decoy features should have a similar score distribution to the negative features.

We tested the training models with and without shape features (DeepRTAlign-shape and DeepRTAlign) on these 30,000 feature pairs from the HCC-N dataset. As shown in **Fig. R4**, we found that although both models can distinguish negative and positive sets, the alignment scores of the decoy pairs from DeepRTAlign (the model without shape features) are close to those of the negative pairs. DeepRTAlign-shape (the model with shape features) did not show the same trend. We think the reason may be that the information from the simplified three-point method is not enough to represent feature shape. However, considering existing tools seldom provide detailed information (intensity and RT) for each MS1 scan in a feature, we plan to consider the peak shape in the next version of DeepRTAlign.

Fig. R4. The alignment score distributions for positive, negative and decoy features output by DeepRTAlign on randomly selected feature pairs from dataset HCC-N. (a) the model with shape features. (b) the model without shape features.

Q4. Will mass errors between MS feature pairs as a DNN feature be helpful in the model? In principle, mass errors are very useful, as in MS1-only methods, mass is the key information we can use to distinguish different signal species. And maybe mass recalibration is also required for different runs.

Response: We thank the reviewer for this kind suggestion. We agree with the reviewer that, as an MS1-only method, the mass error between MS features is a very important parameter for training the alignment model in DeepRTAlign. According to **Reviewer #1's Q3 and Q4** suggestions, we removed intensity from the model features but added m/z as a new model feature. **Fig. 1** has been updated accordingly.

Fig. 1. The illustration of DeepRTAlign algorithm. (a) The training procedures of DeepRTAlign. The n is the feature rank number (sorted by RT) in one m/z window of sample 1. The m is the feature rank number (sorted by RT) in one m/z window of sample 2. Please note that “n-1”, “n-2”, “n+1” and “n+2” represent four adjacent features of the feature n in one m/z window. (b) The workflow for RT alignment using

DeepRTAlign. DeepRTAlign supports four feature extraction methods (Dinosaur, MaxQuant, OpenMS and XICFinder). The features extracted will be aligned using the trained model shown in (a), with the aligned feature list as the output.

Furthermore, we evaluated the performance of DeepRTAlign with QC (FDR<1%) on the previously generated simulated datasets. We found that DeepRTAlign still showed superior performance over OpenMS on all the proteomic datasets (**Fig. R5**, denoted as **Fig. 4** in this revision). While on the metabolomic datasets, the performance of DeepRTAlign is not stable (**Fig. R6**, denoted as **Supplementary Fig. 6** in this revision): it performed well on two datasets (SM1100 and MM) but showed poor performance on three datasets (NCC19, SO and GUS). One possible reason is the model in DeepRTAlign was trained on a proteomic dataset (HCC-T), rather than a metabolomic dataset. Meanwhile, we explored the extracted features in the three metabolomic datasets (NCC19, SO and GUS) and found that they all contained a large number of features with very close m/z. **Fig. R7** shows an example of these features with close m/z, which are from part of the OpenMS-extracted features in one of the GUS samples. Thus, we further tested DeepRTAlign on these three datasets without QC (setting FDR threshold to 100%). As shown in **Fig. R8** (denoted as **Supplementary Fig. 7** in this revision), we found that the performance of DeepRTAlign became comparable with that of OpenMS. However, it should be noted that OpenMS does not perform well on these three data sets either, indicating a potential limitation for MS1-only alignment method.

Fig. R5. Comparison of DeepRTAlign and OpenMS on multiple simulated datasets

generated from 9 real-world proteomic datasets. The simulated datasets were constructed by adding normally distributed RT shifts to the corresponding real-world dataset. (a) $\mu=0$ min. (b) $\mu=5$ min. (c) $\mu=10$ min. The normal distribution has an increasing σ , i.e., $\sigma=0, 0.1, 0.3, 0.5, 0.7, 1, 3, 5$ for different μ (0, 5 and 10 minutes), respectively.

Fig. R6. Comparison of DeepRTAlign and OpenMS on multiple simulated datasets generated from 5 real-world metabolomic datasets. The simulated datasets were constructed by adding normally distributed RT shifts to the corresponding real-world dataset. (a) $\mu=0$ min. (b) $\mu=5$ min. (c) $\mu=10$ min. The normal distribution has an increasing σ , i.e., $\sigma=0, 0.1, 0.3, 0.5, 0.7, 1, 3, 5$ for different μ (0, 5 and 10 minutes), respectively. The FDR of DeepRTAlign's results is set to 1%.

#FEATURE	rt	mz	intensity	charge	width	quality	rt_qualit	mz_qual	rt_start	rt_end
FEATURE	1576.521	436.308	1.85E+04	2	1.061985	0.557512	0	0	1575.792	1577.369
FEATURE	5.213633	436.308	4.12E+04	2	1.918418	0.731764	0	0	4.310362	6.674969
FEATURE	1820.557	436.308	1.59E+04	2	1.085908	0.748352	0	0	1820.141	1821.455
FEATURE	228.0939	436.308	1.40E+04	2	0.854004	0.605057	0	0	227.2972	228.8736
FEATURE	1728.272	436.308	1.79E+04	2	1.191783	0.760323	0	0	1727.919	1729.233
FEATURE	1904.461	436.308	1.44E+04	2	1.038304	0.699026	0	0	1903.693	1905.269
FEATURE	2093.775	436.308	2.62E+04	2	2.015822	0.711677	0	0	2093.391	2094.442
FEATURE	1594.084	436.308	2.10E+04	2	1.349056	0.742974	0	0	1593.659	1594.973
FEATURE	2211.795	436.308	1.89E+04	2	1.416161	0.769985	0	0	2210.574	2211.887
FEATURE	2388.183	436.3081	2.59E+04	2	2.468815	0.549669	0	0	2386.61	2388.974
FEATURE	285.5324	436.3081	2.70E+04	2	1.47624	0.750137	0	0	284.3118	286.4138
FEATURE	136.4081	436.3081	2.83E+04	2	1.258693	0.695483	0	0	136.206	137.2571
FEATURE	313.7667	436.3081	1.53E+04	2	0.889035	0.532343	0	0	313.2133	314.2642
FEATURE	1844.908	436.3081	2.47E+04	2	1.519761	0.746312	0	0	1844.313	1845.627
FEATURE	2288.986	436.3081	1.61E+04	2	1.360502	0.59854	0	0	2288.082	2289.658
FEATURE	1700.515	436.3081	2.60E+04	2	2.004957	0.694282	0	0	1700.240	1700.451

SC_20160411_QCSM_Pos_25ug_HILIC

Fig. R7. Screenshot of a part of the OpenMS-extracted features from one of the GUS samples.

Fig. R8. Comparison of DeepRTAlign and OpenMS on multiple simulated datasets generated from 3 real-world metabolomic datasets. The simulated datasets were constructed by adding normally distributed RT shifts to the corresponding real-world dataset. (a) $\mu=0$ min. (b) $\mu=5$ min. (c) $\mu=10$ min. The normal distribution has an increasing σ , i.e., $\sigma=0, 0.1, 0.3, 0.5, 0.7, 1, 3, 5$ for different μ (0, 5 and 10 minutes), respectively. The FDR of DeepRTAlign's results is set to 100%.

Meanwhile, since most mass recalibration methods are based on peptide identification results, we think there is no golden standard method to conduct this process in ID-free alignment. As an example, **Fig. R9** shows the distribution of the mass errors between 20,000 randomly selected aligned feature pairs from dataset HCC-T. We can see that the mass error is relatively low (μ : -0.61 ppm, σ : 2.85 ppm). Thus, we have not added a mass recalibration module in the current version of DeepRTAlign. But we plan to keep an eye on this issue in the future.

Fig. R9: The distribution of the mass errors between 20,000 randomly selected aligned feature pairs from dataset HCC-T according to Mascot search results.

Q5. Line 235-248: The RT shift simulation is not sufficient. What I also expected is if adding some ($n=5$?) pseudo features to the left RT or right RT of the target features with minor intensity differences, is the algorithm robust enough to get the correct features back?

Response: We thank the reviewer for this helpful suggestion. We performed the simulation experiments following the reviewer's suggestion.

First, we randomly selected a sample in the AT dataset and denoted it as Sample N. Here, it contained 11,278 features. For each feature in Sample N, we added an RT shift of 5 minutes and thus generated a simulated sample named Sample N'. In theory, every feature in Sample N can be aligned to its corresponding feature in Sample N'. As shown in **Fig. R5** (denoted as **Fig. 4** in this revision), the precision and recall of DeepRTAlign are both 100% ($\mu = 5$ min, $\sigma = 0$). Second, for each feature in Sample N, its pseudo feature was generated as follows: a mass shift of 1, 3, and 5 ppm, respectively, was added to its m/z; a random RT shift based on a normal distribution (**Table R1**) was added to its RT. **All 11,278 pseudo features were added to Sample N and Sample N', respectively.** Third, we used DeepRTAlign to align Sample N and Sample N' to test if these similar pseudo features would influence DeepRTAlign's performance. As shown in **Table R1**, we found that the precision and recall of DeepRTAlign stayed unchanged (100%), indicating it is a robust alignment method.

Fig. R5. Comparison of DeepRTAlign and OpenMS on multiple simulated datasets generated from 9 real-world proteomic datasets. The simulated datasets were constructed by adding normally distributed RT shifts to the corresponding real-world dataset. (a) $\mu=0$ min. (b) $\mu=5$ min. (c) $\mu=10$ min. The normal distribution has an increasing σ , i.e., $\sigma=0, 0.1, 0.3, 0.5, 0.7, 1, 3, 5$ for different μ (0, 5 and 10 minutes), respectively.

Table R1. Experiment design of the co-elution test and the corresponding precisions and recalls of DeepRTAlign.

m/z shift (ppm)	RT shift μ (min)	RT shift σ (min)	precision	recall
1	1	0.5	1	1
3	1	0.5	1	1
5	1	0.5	1	1
1	1	1	1	1
3	1	1	1	1
5	1	1	1	1
1	1	1.5	1	1
3	1	1.5	1	1
5	1	1.5	1	1

Q6. Line 386 and Figure S5: In single-cell data, it is not surprising that MS1-based methods get more peptides than MS2-based ones, as orbitrap is not sensitive enough to

detect very low-abundant fragment signals at single-cell level. What concerns me is what peptides were aligned by DeepRTAlign but not DiaNN? Please check both of peptides' MS1 and MS2 signals. As there are only 298 peptides, I suggested to check them all on MS1 and MS2 matches, and show the consistency of chromatographical profiles of both the MS1 and MS2 peaks.

Response: We apologize that our unclear descriptions may have misled the reviewer. As shown in the original **Fig. S5**, DeepRTAlign and DIA-NN's MBR were used to align each sample to all the other samples. In each sample, only the peptide that can be aligned in another sample were considered in the original **Fig. S5**. On average, DeepRTAlign aligned 298 more peptides than DIA-NN for each sample (N=18).

However, this result was obtained by using a 5-minute RT tolerance for matching peptide identifications to features. **Reviewer #2's Q10** suggested that the 5-minute RT tolerance is too big. Thus, in this revision, we **restricted the RT of an MS2 spectrum (peptide identification) to be within the RT range of the corresponding precursor feature when matching the peptide sequence to a feature**. As a result, DeepRTAlign aligned an average of 39 more peptides (6.33%) than DIA-NN for each sample (**Supplementary Fig. 5** in this revision).

However, since DIA-NN does not show the annotated spectra, it is hard for us to manually check the MS1 and MS2 signals for each feature in every sample. In this revision, we showed two example features from dataset SC that were successfully aligned by DeepRTAlign, rather than DIA-NN in **Fig. R10** and **Fig. R11**. The extracted ion currents (XIC) of the two features (corresponding to peptides SSVPGVR and EKAEGDAAALNR) were obtained from Thermo XcaliburTM software (v3.0).

The feature in sample N_12 was aligned to the features corresponding to SSVPGVR in other samples by DeepRTAlign, but was not aligned by DIA-NN with MBR.

The feature was identified by DIA-NN with MBR as peptide SSVPGVR in sample N_1.

The feature was identified by DIA-NN with MBR as peptide SSVPGVR in sample N_3.

The feature was identified by DIA-NN with MBR as peptide SSVPGVR in sample N_4.

Fig. R10. The extracted ion currents (XIC) of a feature (m/z : 351.20 Th and charge: 2) that was successfully aligned by DeepRTAlign, rather than DIA-NN's MBR in sample N_12 from dataset SC. The corresponding peptide sequence is SSVPGVR based DIA-NN's identification results.

The feature in sample N_12 was aligned to the feature corresponding to EKAEGDAAALNR in other sample by DeepRTAlign, but was not aligned by DIA-NN with MBR.

The feature was identified by DIA-NN with MBR as peptide EKAEGDAAALNR in sample N_3.

Fig. R11. The extracted ion currents (XIC) of a feature (m/z : 622.79 Th and charge: 2) that was successfully aligned by DeepRTAlign, rather than DIA-NN's MBR in sample N_12 from dataset SC. The corresponding peptide sequence is EKAEGDAAALNR based DIA-NN's identification results.

For a clearer description, we have modified the corresponding descriptions about the **Supplementary Fig. 5** as follows (“**Comparison with existing alignment tools**” section, **Page 15**):

Considering the aligned peptides present in at least two cells, DeepRTAlign can align 39 (6.33%) more peptides on average than the existing popular tool DIA-NN in each cell.

Supplementary Fig. 5. The peptide number and feature number of each HT22 cell. Features are extracted by Dinosaur. Only the features presented in at least two cells are considered. MBR: match between runs. Error bar indicates standard deviation. It should be noted that a group is defined as a set of aligned features in different runs.

Q7. Figure 1: Triangle and square representation does not make it easier to understand the Input Vector, especially for adjacent features. A better Illustration is required.

Response: We thank the reviewer for this helpful suggestion. We replaced the triangle and square with a text description. Following **Reviewer #1's Q3, Q4 and Q7** suggestions, we removed intensity from the model features but added m/z as a new model feature. The updated **Fig. 1** is shown below.

Fig. 1. The illustration of DeepRTAlign algorithm. (a) The training procedures of DeepRTAlign. The n is the feature rank number (sorted by RT) in one m/z window of sample 1. The m is the feature rank number (sorted by RT) in one m/z window of sample 2. Please note that “ $n-1$ ”, “ $n-2$ ”, “ $n+1$ ” and “ $n+2$ ” represent four adjacent features of the feature n in one m/z window. (b) The workflow for RT alignment using DeepRTAlign. DeepRTAlign supports four feature extraction methods (Dinosaur, MaxQuant, OpenMS and XICFinder). The features extracted will be aligned using the trained model shown in (a), with the aligned feature list as the output.

Q8. Line 158-159: More explanation is needed about why using two adjacent features as the input vector. What is the idea behind it?

Response: We found that if a feature (such as feature n in sample 1) is correctly aligned to another feature (such as feature m in sample 2), the similarity of feature n and the adjacent feature of feature m is more likely to be low, thus, considering the adjacent features can help us perform an accurate alignment. As shown in **Fig. 1**, we consider the two adjacent features before and after the feature to be aligned as **the input vector** to increase model performance in DeepRTAlign.

Q9. In Figure 2 to 4, it is better to show some PSM or XIC examples illustrating how DeepRTAlign can get MS1 features back by alignment, in addition to number comparisons.

Response: We thank the reviewer for this suggestion. According to **Reviewer #1's Q2, Q3 and Q4** suggestions, we updated the input model feature for training DeepRTAlign and implemented a QC module. Thus, we have updated all the results in revision using the new version of DeepRTAlign. When matching features with peptide identification results, we restricted the RT of an MS2 spectrum (peptide identification) to be within the RT range of the corresponding precursor feature. Taking the updated **Fig. 2** as an example, the new version of DeepRTAlign is superior to MZmine 2 and OpenMS in both precision and recall. Considering the search results of Mascot as ground truth, we show two examples aligned by DeepRTAlign but not by MZmine 2 or OpenMS in **Tables R2 and R3**. In both cases, DeepRTAlign is able to align the correct ions despite the interference ions with similar m/z and RT. We think this is because MZmine 2 and OpenMS are likely to align to the interference ions with similar RT when using a warping function. While, DeepRTAlign can consider the m/z and RT of the target feature and its adjacent features for alignment in its deep learning-based model to achieve an accurate alignment (**Fig. 1**).

Fig. 2. Performance evaluation of DeepRTAlign compared with MZmine 2 and OpenMS. Panel a shows the precisions and recalls of MZmine 2 and DeepRTAlign on

different test sets. Panel b shows the precisions and recalls of OpenMS and DeepRTAlign on different test datasets. “FE” means the feature extraction method. “A” means the RT alignment method. We took the Mascot identification results with FDR<1% as the ground truth in datasets HCC-N, UPS2-M and UPS2-Y and took the MaxQuant identification results with FDR<1% as the ground truth in datasets EC-H and AT. In dataset EC-H, we only considered the *E. coli* peptides for evaluation. In datasets UPS2-M and UPS2-Y, we only considered the UPS2 peptides for evaluation.

Table R2. One example of a feature pair successfully aligned by DeepRTAlign, but not by MZmine 2. The peptide sequence was identified from the Mascot search results.

Method	sample	m/z	charge	intensity	RT	peptide
DeepRTAlign	sample 1	419.2094	2	2299482	22.543	EESGFLR
DeepRTAlign	sample 2	419.2088	2	4346550	21.57131	EESGFLR
MZmine 2	sample 1	419.2097	2	18040604	22.97301	EFLDGTR
MZmine 2	sample 2	419.2088	2	4346550	21.57131	EESGFLR

Table R3. One example of a feature pair successfully aligned by DeepRTAlign, but not by OpenMS. The peptide sequence was identified from the Mascot search results.

Method	sample	m/z	charge	intensity	RT	peptide
DeepRTAlign	sample 1	530.7687	2	1.54E+09	11.97347	QVDQLTNDK
DeepRTAlign	sample 2	530.7673	2	1.19E+09	11.89762	QVDQLTNDK
OpenMS	sample 1	530.7589	2	5.11E+08	11.93935	TCTPTPVER
OpenMS	sample 2	530.7673	2	1.19E+09	11.89762	QVDQLTNDK

Q10. It is better to test this tool on quantification benchmark datasets such as LFQ-bench with known fold changes.

Response: We thank the reviewer for the valuable suggestion. Following the reviewer’s suggestion, we designed and produced a benchmark dataset (dataset Benchmark-FC) with known fold changes in this revision. In this benchmark dataset, 10 ng, 15 ng, 20 ng, and 25 ng of *E. coli* samples were spiked into 200 ng of 293T samples, respectively. Each group contained 3 replicates.

We used the peptide intensities aligned by DeepRTAlign or the MBR function of OpenMS, MaxQuant and MSFragger to calculate the ratio of all *E. coli* peptides between specific samples (15 ng/10 ng, 20 ng/10 ng, and 25 ng/10 ng) in each replicate. As shown in **Fig. 3**, we found that the combination of using MaxQuant to extract

features, DeepRTAlign to align features and MSFragger to identify their peptide sequences could obtain 150% more peptides compared with the original MaxQuant workflow (using MaxQuant to extract features, align them and identify their peptide sequences) without compromising the quantification accuracy. DeepRTAlign also aligned 7% and 14.5% more peptides than OpenMS and MSFragger's MBR, respectively, while its quantification accuracy was comparable with other methods. Overall, we found that OpenMS extracted features represented a slightly better accuracy, regardless of the alignment methods employed. This may be because OpenMS has strict quality controls in feature extraction. DeepRTAlign can still increase the number of aligned peptides based on OpenMS extracted features, showing a higher sensitivity than OpenMS in terms of alignment.

It should be noted that DeepRTAlign is compatible with multiple feature extraction methods (Dinosaur, MaxQuant, OpenMS and XICFinder), which is convenient for users to choose the feature extraction method suitable for their experiments.

Fig. 3. The number and ratio distributions of all *E. coli* peptides between specific samples (15ng/10ng, 20ng/10ng, and 25ng/10ng) in each replicate (R1, R2 and R3) after alignment. In all boxplots, the center black line is the median of log₂ of the peptide fold changes. The box limits are the upper and lower quartiles. The whiskers extend to 1.5 times the size of the interquartile range.

Minor:

Q11. I suggest to rename "ion-centric" as it is not specific, identification-based methods also use ions.

Response: We thank the reviewer for this kind suggestion. "Ion-centric" was renamed "ID-free" in this revision.

Q12. Line 143-144: "The precursor ions with more than one ..." is hard to understand. What does it mean "the precursor ions"? As a MS2-free method, why it needs "more than one MS2 spectrum"?

Response: We apologize that our unclear descriptions may have misled the reviewer. We have deleted this sentence to eliminate the discrepancy.

Q13. Line 154-155: What algorithm did you use for alignment? It should be mentioned here.

Response: To align each piece with the anchor sample, the average RT shift between a piece and the anchor sample is directly added to each feature in this piece. We have added a detailed description in this revision ("**Workflow of DeepRTAlign**" section, **Page 7**).

Q14. Table S1: As this is only supplementary info, please put more information in the table legend, such as full dataset description and so on. Otherwise, it is hard to understand the abbreviations such as SC, AT, etc.

Response: We thank the reviewer for this kind suggestion. We have updated **Supplementary Table 1** accordingly. All the datasets, including the five newly generated datasets in this revision, have been listed in **Supplementary Table 1**.

Q15. It is better to test the alignment on samples with different LC conditions, such as 21 min gradient to 120 min gradient.

Response: Following the reviewer's suggestion, we tried to align one sample with 60 min gradient to another with 120 min gradient using the Benchmark-RT dataset. According to the identification results of Proteome Discoverer, 9,529 peptides were commonly identified in two samples. Then, the precision and recall of DeepRTAlign and OpenMS were calculated based on the 9529 peptides. **Table R4** shows that DeepRTAlign showed a much higher precision and recall than OpenMS, suggesting that DeepRTAlign is robust for alignment on samples with different RT gradients.

Table R4. The precision and recall of DeepRTAlign and OpenMS when aligning two samples with different RT gradients (60 min vs 120 min).

Method	precision	recall
DeepRTAlign	0.91	0.94
OpenMS	0.22	0.06

Reviewer #2 (Remarks to the Author):

Liu Yi and colleagues have presented a manuscript in which they use a deep neural network (DNN) for retention time alignment. Compared to commonly used methods, the authors claim to have generated superior peak-correspondence with MS1 features alone. They have compared the performance of DNN against four machine-learning methods, and other software packages such as MaxQuant-MBR, FragPipe-MBR, OpenMS, MZmine2. While the results suggest that the deep neural network performs better than the other methods, it should be noted that a standard quantitative approach for analyzing these tools is currently lacking. Further discussion on the strengths and limitations of the manuscript is provided below:

Response: Yes, we agree that a standard quantitative approach to evaluate these alignment tools is currently lacking. Following reviewers' valuable suggestions, a new benchmark dataset (Benchmark-FC) with known fold changes is generated and used for quantitative evaluation in this revision. We performed PRM experiments for further validation. Please see the detailed replies below. All the datasets used in this study (**Supplementary Table 1**) are freely available. We expect to provide a resource and a solution to alignment evaluation for the community.

Q1. Comparison to other DNN: Authors have not clarified as to why the Siamese network by Li et al which has three encoders - the Mass Spectrum Encoder, Peak Encoder, and Chromatograph Encoder in addition to Δ RT, would fail. It is natural to assume that incorporating more information, such as chromatogram and MS/MS encoding, would lead to better performance. However, the authors did not clarify whether the nature of the data, in this case, GC-MS and LC-MS, could be a factor in this regard. Although the authors have provided an explanation in the Supplementary Notes, it is not entirely convincing.

Response: We apologize that our unclear descriptions misled the reviewer. We agree that Li et al.'s Siamese network (ChromAlignNet) contains more information due to

the three encoders, but it was developed for GC-MS data alignment. We would like to clarify that the Siamese network we compared with DeepRTAlign in **Supplementary Notes** was built by ourselves rather than ChromAlignNet. In our original manuscript, the Siamese network we built was trained on the same training set (LC-MS data rather than GC-MS data) using the same input model features (RT and intensity). However, we have updated the input model features following **Reviewer #1's Q3 and Q4** suggestions (**Fig. 1**). We agree that the data type (GC-MS or LC-MS) should be considered for training an alignment model. Our model in DeepRTAlign was trained and tested on LC-MS data, while ChromAlignNet was trained and tested on GC-LC data. Thus, to make a clear description, we revised the **Introduction** section (**Page 3** in the main text) and **Supplementary Notes (Page 15** in the SI). The original Table S11 has been updated using the new input model features and denoted as **Supplementary Table 18** in this revision.

Q2. Standard benchmarking is needed: The model presented by the authors explicitly uses intensity for peak-matching, which can introduce a conservative bias towards selecting features with similar intensity. Thus, the use of the coefficient of variation (CV) as a measure to assess the superiority of a tool is not entirely justifiable. It would be more appropriate for the authors to compare the performance of the different tools using manually annotated heterogeneous data or test them on dilution-series or synthetic-proteomics data to demonstrate that the intensity of aligned features is correlated with the spiked-in pattern. Currently, it appears that the method is vulnerable to producing good CV but may not accurately capture differential expression.

Response: We thank the reviewer for the valuable suggestion. We agree that using intensity for alignment is not proper, and thus only using CV as a measure of alignment performance is not entirely justifiable. Following **Reviewer #2's Q2** and **Reviewer #1's Q10** suggestions, we have designed and generated a new benchmark dataset (dataset Benchmark-FC) with known fold changes for quantification evaluation. In this benchmark dataset, 10 ng, 15 ng, 20 ng, and 25 ng of *E. coli* samples were spiked into 200 ng of 293T samples, respectively. Each group contained 3 replicates. We used the peptide intensities aligned by DeepRTAlign or the MBR function of OpenMS, MaxQuant and MSFragger to calculate the ratio of all *E. coli* peptides between specific samples (15 ng/10 ng, 20 ng/10 ng, and 25 ng/10 ng) in each replicate.

As shown in **Fig. 3**, we found that the combination of using MaxQuant to extract features, DeepRTAlign to align features and MSFragger to identify their peptide

sequences could obtain 150% more peptides compared with the original MaxQuant workflow (using MaxQuant to extract features, align them and identify their peptide sequences) without compromising the quantification accuracy. DeepRTAlign also aligned 7% and 14.5% more peptides than OpenMS and MSFragger's MBR, while its quantification accuracy was comparable with other methods. Overall, we found that OpenMS extracted features represented a slightly better accuracy, regardless of the alignment methods employed. This may be because OpenMS has strict quality controls in feature extraction. DeepRTAlign can still increase the number of aligned peptides based on OpenMS extracted features, showing a higher sensitivity than OpenMS in terms of alignment.

It should be noted that DeepRTAlign is compatible with multiple feature extraction methods (Dinosaur, MaxQuant, OpenMS and XICFinder), which is convenient for users to choose the feature extraction method suitable for their experiments.

Fig. 3. The number and ratio distributions of all *E. coli* peptides between specific samples (15ng/10ng, 20ng/10ng, and 25ng/10ng) in each replicate (R1, R2 and R3) after alignment. In all boxplots, the center black line is the median of log₂ of the peptide fold changes. The box limits are the upper and lower quartiles. The whiskers extend to 1.5 times the size of the interquartile range.

Q3. Have not elaborated upon the technical details about DNN: In the manuscripts authors have presented the results of various software tools, but have not critically analyzed them. Specifically, the authors have not explored in detail the specific properties of the deep neural network (DNN) that make it superior to other tools, such as MZmine2, OpenMS, or MBR. It is essential to investigate what kind of nonlinearities the DNN captures that cannot be identified by other methods. Further exploration of the results is necessary to draw definitive conclusions.

Response: We thank the reviewer for this helpful suggestion. According to Smith et al.'s review of RT alignment¹, there are two types of computational methods for RT alignment: the warping method (such as MZmine 2 and OpenMS) and the direct matching method (such as MBR). Based on Mitra et al.'s paper², RT shifts can be classified into two parts: the monotonic and non-monotonic shifts. **Monotonic shifts** are defined as the differences of values in one of the RT and m/z dimensions or in the ion intensity readout of MS1 map pair of the same compounds (for RT and m/z dimensions) or the same compounds with the same quantity (ion intensity readout) that can be corrected using a monotonic function. **Non-monotonic shifts** are defined as the differences of values in one of the RT and m/z dimensions or in the ion intensity readout of MS1 map pair of the same compounds (for RT and m/z dimensions) or the same compounds with the same quantity (ion intensity readout), which remains after correction with monotonic shift. Here, we try to explain the superiority of DeepRTAlign over MZmine 2, OpenMS and MBR technically and theoretically following the reviewer's suggestion.

(1) Comparison with MZmine 2 and OpenMS. Since the warping functions used in MZmine 2 and OpenMS are monotonic, the features aligned by the two methods can be considered as the features with monotonic RT shift¹. And the rest of the features can be considered as those with non-monotonic shift. Taking the OpenMS-extracted features as an example, the features with monotonic shifts and non-monotonic shifts in one HCC-N pair (patient 314, fraction 1 vs. patient 355, fraction 1) were shown in **Fig. R12**. The features commonly identified by OpenMS and DeepRTAlign, as well as the features uniquely identified by OpenMS can be regarded as monotonic shift features. The features uniquely identified by DeepRTAlign as well as the "none" features, can be regarded as non-monotonic shift features (**Fig. R12**). The percentages of monotonic shift features and non-monotonic shift features in 24 randomly selected sample pairs from dataset HCC-N were shown in **Fig. R13**, where an average of **81.78%** of features have monotonic shifts and **18.22%** of features have non-monotonic shifts.

DeepRTAlign can align **96.51%** of all the ground truths on average.

(2) Comparison with MBR. Briefly, the MBR algorithm compares each identified peak in an MS1 spectrum from an LC–MS/MS run to the unidentified peaks in another run. An identification is transferred if an unidentified peak with the same properties (e.g., m/z and charge state) is found within a specified RT window. Thus, the RT window size is a key parameter. MaxQuant and MSFragger (FragPipe) usually use a static RT window size on all samples. But based on the DNN model of DeepRTAlign, an appropriate window size can be learned from the training data to distinguish whether two features should be aligned.

Here, considering the Mascot search results as ground truth, we show two examples aligned by DeepRTAlign but not by MZmine 2 or OpenMS in **Tables R2 and R3**. In both cases, DeepRTAlign is able to align the correct ions despite the interference ions with similar m/z and RT. We think this is because MZmine 2 and OpenMS are likely to align to the interference ions with similar m/z using a warping function. DeepRTAlign can consider the m/z and RT of the target feature and its adjacent features for alignment in its deep learning-based model (**Fig. 1**).

In summary, DeepRTAlign can deal with monotonic shifts as well as non-monotonic shifts by implementing a coarse alignment as a pseudo-warping function for monotonic RT shifts and a deep learning-based model for non-monotonic RT shifts.

Fig. R12. Comparison of monotonic and non-monotonic shift features in one HCC-N pair (patient X314, fraction 1 vs. patient X355, fraction 1). (a) The scatterplot shows the RT of the features that should be aligned according to the Mascot identification results in one HCC-N pair. Shared: commonly aligned by DeepRTAlign and OpenMS. DeepRTAlign Only: only aligned by DeepRTAlign. OpenMS Only: only aligned by

OpenMS. None: should be aligned based on Mascot results, but both DeepRTAlign and OpenMS failed to align the features. (b) The Venn diagram of all the features that should be aligned according to the Mascot identifications in one HCC-N pair.

Fig. R13. The proportions of the features with monotonic shifts (OpenMS), non-monotonic shifts and DeepRTAlign aligned features in 24 randomly selected sample pairs from dataset HCC-N.

Table R2. One example of a feature pair successfully aligned by DeepRTAlign, but not by MZmine 2. The peptide sequence was identified from the Mascot search results.

Method	sample	m/z	charge	intensity	RT	peptide
DeepRTAlign	sample 1	419.2094	2	2299482	22.543	EESGFLR
DeepRTAlign	sample 2	419.2088	2	4346550	21.57131	EESGFLR
MZmine 2	sample 1	419.2097	2	18040604	22.97301	EFLDGTR
MZmine 2	sample 2	419.2088	2	4346550	21.57131	EESGFLR

Table R3. One example of a feature pair successfully aligned by DeepRTAlign, but not by OpenMS. The peptide sequence was identified from the Mascot search results.

Method	sample	m/z	charge	intensity	RT	peptide
DeepRTAlign	sample 1	530.7687	2	1.54E+09	11.97347	QVDQLTNDK
DeepRTAlign	sample 2	530.7673	2	1.19E+09	11.89762	QVDQLTNDK

OpenMS	sample 1	530.7589	2	5.11E+08	11.93935	TCTPTPVER
OpenMS	sample 2	530.7673	2	1.19E+09	11.89762	QVDQLTNDK

References:

1. Smith, R.; Ventura, D.; Prince, J. T. LC-MS Alignment in Theory and Practice: A Comprehensive Algorithmic Review. *Briefings in Bioinformatics* 2013, 16 (1), 104–117. <https://doi.org/10.1093/bib/bbt080>.
2. Mitra, V.; Smilde, A. K.; Bischoff, R.; Horvatovich, P. Tutorial: Correction of Shifts in Single-Stage LC-MS(/MS) Data. *Analytica Chimica Acta* 2018, 999, 37–53. <https://doi.org/10.1016/j.aca.2017.09.039>.

Q4. Quantitative analysis is lacking. Retention time alignment is typically carried out to enhance the quantitateness of LC-MS/MS data, which should, in turn, enable the identification of novel biological insights. While the authors have presented some analysis for early recurrence prediction, the analysis is currently quite crude and lacks convincing evidence and is not convincing at all. They should provide an explanation and interpretation of differential features. For example, how many belong to identified peptides, how many are low-abundant and present how many have reproducible MS/MS spectra across multiple samples.

Response: We thank the reviewer for pointing out the lack of quantitative analysis and that the early recurrence prediction results were not convincing in our original manuscript. To address these issues, we enrolled a new HCC cohort (named HCC-R2, N=23) and performed scheduled PRM (sPRM) experiments on this cohort to validate a subset of the top 200 MS features identified from our training set (HCC-T), including the 15 features used for early recurrence prediction and 34 other features (**Fig. 5a**). The sPRM data were analyzed by Proteome Discoverer (PD) and Skyline. Remarkably, 48 out of the 49 target features (**Supplementary Table 15**) were identified by Proteome Discoverer (PD) results (**Supplementary Table 16**) and further verified in Skyline results (**Supplementary Table 17**) of the sPRM data. Only one feature failed to be identified, possibly due to low abundance. Importantly, all 48 identified features exhibited reproducible, high-quality MS/MS spectra in the majority of the HCC samples in the HCC-R2 cohort. Please see the Skyline annotated spectra for each feature in **Supplementary File 1**. Furthermore, we performed an independent test of the 15 features on the HCC-R2 cohort (N=23) using the Skyline results from sPRM data. As shown in **Fig. R1** (denoted as **Fig. 5d** in this revision), we found that the AUC

value of the classifier to predict HCC early recurrence is 0.833, indicating the good generalizability of our original findings, despite the fact that the HCC samples used in the sPRM experiments were unfractionated and collected from another medical center. Overall, these results provide convincing evidence for the reliability of our ID-free strategy and show the ability to discover novel biological insights. We have incorporated these new results into our revised manuscript (**Page 16**).

Fig. 5a. Study design for construction and validation of this 15-feature-based classifier. First, all the features in dataset HCC-T (N=101) were aligned by DeepRTAlign. Top 200 features were selected by mRMR. We mapped the top 200 MS features to an independent test set HCC-R (C2). There were 15 features successfully mapped to HCC-R. In the 101 patients from dataset HCC-T, the cases of early recurrence (N=30), and the cases of late recurrence (N=22) were considered as the training set (C1) for HCC early recurrence prediction. Then, the mapped 15 features were used to train a SVM classifier based on the C1 dataset, and tested on the HCC-R dataset (C2, N=11). Peptide- and protein-based classifiers were generated in the same way, but they were

not marked in Fig.5a for clarity. These 15 features in the early recurrence classifier were then targeted in PRM experiments and analyzed by PD and Skyline based on an independent dataset HCC-R2 (C3, N=23). The 15-feature-based classifier was further tested on dataset HCC-R2 (C3) using the Skyline quantification results.

Fig. R1. The test AUC on PRM dataset HCC-R2 of the 15-feature-based classifier for HCC early recurrence prediction.

Q5. In Table S4, authors have compared the DNN with other non-linear methods for alignment. The performance with logistic regression is very poor. I wonder what could be the reason for that? Additionally, EC-H has only 20 runs; for such small-datasets non-linear methods typically perform at over 90% accuracy, which is not observed in this case. The authors should provide a more detailed technical assessment of the specific objectives achieved by the DNN that are missed by other tools, particularly in the case of smaller datasets like EC-H. The specific details of what the DNN brings to the table that logistic regression, KNN, and SVM cannot achieve should also be elaborated upon.

Response: We thank the review for the suggestion. We agree that for small datasets, non-linear methods typically perform at over 90% accuracy. In this revision, according to **Reviewer #1's Q3 and Q4** suggestions, we removed intensity from the model features but added m/z as a new model feature (**Fig. 1**). All the results have been updated using the new version of DeepRTAlign. The original Table S4 is denoted as **Supplementary Table 3** in this revision. As shown in the **Supplementary Table 3**, we found DNN outperforms other machine learning models in all the test datasets. Please note that in this revision, all the results on test datasets were based on the model trained

on the whole training set (HCC-T) with tuned hyperparameters (hyperparameters were optimized based on 10-fold cross validation on the training set HCC-T), no longer from the 10-fold cross validation models on the training set HCC-T. Thus, we did not have to calculate the mean and standard deviation of the precisions and recalls in each test set. As shown in the **Supplementary Table 3**, SVM and LR performed much better than before, indicating that it may be inappropriate to use intensity as features for SVM and LR. But for KNN, its performance did not increase after using new features suggesting it may not be suitable for the current task.

Supplementary Table 3. The AUCs on different test sets. All the results are based on the model trained on the HCC-T dataset. In each test set, we randomly selected 10,000 positive and 10,000 negative feature pairs to perform this evaluation.

Dataset	DNN	RF	KNN	SVM	LR
HCC-N	0.925	0.916	0.656	0.865	0.894
HCC-R	0.933	0.905	0.668	0.901	0.899
UPS2-M	0.979	0.919	0.683	0.896	0.905
UPS2-Y	0.971	0.920	0.702	0.900	0.897
EC-H	0.972	0.938	0.733	0.912	0.944
AT	0.975	0.943	0.785	0.932	0.945
SC	0.917	0.901	0.752	0.842	0.898

Theoretically, a DNN with non-polynomial activation functions can approximate any function¹. This great property of DNN makes it less depend on an elaborately designed input features and largely save our efforts on feature engineering and hyper-parameter tuning. On the contrary, classical machine learning methods such as a KNN model, are more sensitive to and largely rely on the selected features and hyper-parameters since their approximation ability is not as strong as a DNN. Therefore, whether these classical methods perform well depends more on the complexity of the underlying mapping function between the given feature and the target variable that needs to be approximated, while the amount of data may be the secondary factor. And we believe this is the reason why the other models cannot achieve satisfactory performance in our task.

Reference:

1. M. Leshno, V. Ya. Lin, A. Pinkus and S. Schocken (1993), 'Multilayer feedforward networks with a non-polynomial activation function can approximate any function', *Neural Networks* 6, 861-867.

Q6. Authors have compared their tool to XCMS, OpenMS which employ a monotonic warping function. They have also reported other direct-matching methods: RTAlign, MassUntangler and PeakMatch which they claim are not suitable, but their alignment philosophy is similar to that of DeepRTAlign. The authors should describe specifically what drawbacks of the direct-matching approaches are addressed and fixed in their approach. In addition, it would be beneficial for the authors to benchmark their method against other similar approaches, such as LWBmatch.

Response: We thank the reviewer for this helpful suggestion. Direct matching methods attempt to establish a correspondence based only on the similarity between specific signals from run to run without applying a warping function. The core of this type of method is how to accurately calculate the similarity. As stated in Smith et al.'s review¹, direct matching methods are easily influenced by the data complexity, especially for alignment of large-scale samples.

Here, we would like to clarify that the philosophy of DeepRTAlign is combining both advantages of the warping function-based and direct-matching methods. Its coarse alignment step can be regarded as a simplified warping function, and its DNN can be regarded as a direct matching method. To further improve the alignment accuracy, DeepRTAlign considers not only the feature to be aligned but also its adjacent features.

In addition, we compared DeepRTAlign with LWBmatch and OpenMS on the newly generated dataset with known fold changes (dataset Benchmark-FC) using the OpenMS extracted features and MSFragger search results. As shown in **Fig. R14**, we can find that the quantification accuracy of LWBmatch is not as good as that of OpenMS and DeepRTAlign.

Fig. R14. The number and ratio distributions of all *E. coli* peptides between specific samples (15ng/10ng, 20ng/10ng, and 25ng/10ng) in each replicate (R1, R2 and R3) after alignment by LWBMatch, OpenMS and DeepRTAlign.

Reference:

1. Smith, R.; Ventura, D.; Prince, J. T. LC-MS Alignment in Theory and Practice: A Comprehensive Algorithmic Review. *Briefings in Bioinformatics* 2013, 16 (1), 104–117. <https://doi.org/10.1093/bib/bbt080>.

Q7. What is the output vector of DNN? Is it binary Aligned/not-aligned? What would it choose if the feature n is aligned to m-1, m and m+1, I am assuming that you slide over five features while aligning them?

Response: We are sorry for the unclear descriptions. The output is binary aligned/not-aligned. As shown in **Fig. 1**, when the feature n is aligned to feature m, **the DeepRTAlign model will consider the adjacent features before and after the feature n and feature m as the input vector**, rather than sliding over five features while aligning them. We found that if a feature (such as feature n in sample 1) is correctly aligned to another feature (such as feature m in sample 2), the similarity of feature n and the adjacent feature of m (such as m-1 or m+1) is more likely to be low. Thus, considering the adjacent features can help us perform an accurate alignment.

Fig. 1. The illustration of DeepRTAlign algorithm. (a) The training procedures of DeepRTAlign. The n is the feature rank number (sorted by RT) in one m/z window of sample 1. The m is the feature rank number (sorted by RT) in one m/z window of sample 2. Please note that “ $n-1$ ”, “ $n-2$ ”, “ $n+1$ ” and “ $n+2$ ” represent four adjacent features of the feature n in one m/z window. (b) The workflow for RT alignment using DeepRTAlign. DeepRTAlign supports four feature extraction methods (Dinosaur, MaxQuant, OpenMS and XICFinder). The features extracted will be aligned using the trained model shown in (a), with the aligned feature list as the output.

Q8. The claim that MS1-based alignment can outperform approaches that use both MS1 and MS2 information is counterintuitive. For example, Quandenser uses decoys to control false mappings while utilizing MS/MS anchor peptides for aligning MS1 chromatograms. However, the manuscript lacks rigorous analysis to support this claim. **Response:** We thank the reviewer for this helpful comment. Following **Reviewer #2’s Q2** suggestions, using intensity for alignment is improper, and thus only using CV as a measure of alignment performance is not entirely justifiable. Therefore, we compared DeepRTAlign with Quandenser on a new benchmark dataset (dataset Benchmark-FC) with known fold changes for performance evaluation. Please note that the

DeepRTAlign evaluated in this revision is an updated version with new input model features and a QC module. As shown in **Fig. R15a-b**, we can find that DeepRTAlign is comparable to Quandenser in terms of the number of aligned peptides and the quantification accuracy. DeepRTAlign can align more features (regardless of corresponding identification results) than Quandenser since it is ID-free (**Fig. R15c**). The original **Fig. 3** is removed and **Fig. R15** is denoted as **Supplementary Fig. 4** in this revision.

Fig. R15. The number and ratio distributions of all *E. coli* peptides and the group number of aligned features between specific samples (a: 15ng/10ng, b: 20ng/10ng, and c: 25ng/10ng) in each replicate (R1, R2 and R3) after alignment by Quandenser and DeepRTAlign. It should be noted that a group is defined as a set of aligned features in different runs.

Q9. Line 120: Add a supplementary table like S1 for metabolomics and proteomics data.
Response: We have updated **Supplementary Table 1** to contain all the datasets in this work.

Q10. Line 214: The RT tolerance for identification is too big (5 mins). It should be less than a minute. Please report precision and recall with updated RT tolerance.

Response: We thank the reviewer for this valuable suggestion. We agree that a 5-minute RT tolerance is too big. Thus, in this revision, we restricted the RT of an MS2 spectrum (peptide identification) to be within the RT range of the corresponding precursor feature when matching the peptide identification result to a feature. All the results have been updated in this revision.

Q11. Figure 3: Report median and standard deviation for each boxplot. Are the intensities normalized across runs before calculating CV?

Response: We thank the reviewer for this helpful comment. Feature intensities were not normalized across runs before calculating CV in our original manuscript. However, following **Reviewer #2's Q2** suggestions, using intensity for alignment is improper and only using CV as a measure of alignment performance is not entirely justifiable. Therefore, we removed this figure and further compared DeepRTAlign with Quandenser on a new benchmark dataset (dataset Benchmark-FC) with known fold changes for performance evaluation. Please note that the DeepRTAlign evaluated in this revision is a new version with new input model features and a QC module. Please see **Supplementary Fig. 4** and the reply to **Reviewer #2's Q8** for details.

Q12. Figure 4a. Why are the last two bars different for MSFragger no-aligned? Figure 4: Explain what is unique peptides and shared peptides? Is common means identified in all runs 100%?

Response: We are sorry for these unclear descriptions. The reason that the last two bars for MSFragger no-aligned were different is that MBR (F) no-aligned means features were extracted by MSFragger and then identified by MSFragger, while DeepRTAlign (MF) no-aligned means features were extracted by MaxQuant and then identified by MSFragger (because MSFragger did not provide its extracted features). Shared peptides were the peptides that existed in both no-aligned and aligned. Unique peptides only exist in no-aligned or aligned. Common means exist in both no-aligned and aligned, rather than "identified in all runs 100%". Please note that the peptides without missing values in all three replicates were considered in the **original Fig. 4**. However, following the suggestion in **Reviewer #2's Q2**, only using CV as a measure of alignment performance is not entirely justifiable. Thus, we compared DeepRTAlign with other alignment methods on a new benchmark dataset (dataset Benchmark-FC) with known

fold changes for performance evaluation (**Fig. 3** in this revision). The **original Fig. 4** has been removed in this revision.

Reviewer #3 (Remarks to the Author):

General comment on the paper

Overall, the paper is well-written and structured. The authors thoroughly tested their proposed method, DeepRTAlign, and generously provided code for using the tool. However, there are some areas that could be further improved. At present, the selection of datasets does not make it clear if this method is a significant contribution to improving alignment (see M1 and M4), and some aspects and details of the approach are not clearly explained (see M2). Finally, code for the experiments is not available, which hinders reproducibility (see M3). Below, please find my (constructive) points for the authors to consider to further improve the paper.

Response: We thank the reviewer for the positive remarks and valuable suggestions.

Major comments

M1. Impact of the method

Although the authors performed an extensive benchmark of their method, IMHO it is still unclear if DeepRTAlign is a significant contribution to improve alignment. Currently, there is a bloat of very similar methods for alignment with essentially the same performance of the algorithms they pretend to improve. Part of the problem is that the authors of each method select their own evaluation data, which is not ideal since researchers may select (probably unconsciously) datasets where their methods perform better. To tackle this issue, the use of common benchmarks should be promoted. Therefore, I suggest including the datasets from [1] in the comparison. These new set of tests can also be used to include further information as suggested in later commentaries.

[1] Eva Lange, Ralf Tautenhahn, Steffen Neumann and Clemens Gröpl. "Critical assessment of alignment procedures for LC-MS proteomics and metabolomics measurements". BMC Bioinformatics 2008, 9:375

Response: We thank the reviewer for this helpful suggestion. We agree that some researchers may select datasets where their methods perform better. Thus, we tried to include different types of datasets (proteomics or metabolomics, different species, MS instruments, MS parameters, LC columns etc.) to make a fair and comprehensive evaluation of DeepRTAlign and the other methods in this work. Lange et al.'s dataset

is a classical benchmark dataset that was published 15 years ago. Lange et al.'s dataset is a low-resolution dataset, generated by an ESI IT mass spectrometer (Thermo Finnigan Dexa XP Plus). However, DeepRTAlign is trained on a high-resolution proteomic dataset (HCC-T) generated by an Orbitrap Fusion mass spectrometer (Thermo Fisher) with the OT-IT mode. Thus, we admit that DeepRTAlign is only applicable to high-resolution datasets.

Following **Reviewer #1's Q10** and **Reviewer #2's Q2** suggestions, we designed and produced a benchmark dataset (dataset Benchmark-FC) with known fold changes in this revision. In this benchmark dataset, 10 ng, 15 ng, 20 ng, and 25 ng of *E. coli* samples were spiked into 200 ng of 293T samples, respectively. Each group contained 3 replicates. We used the peptide intensities aligned by DeepRTAlign or the MBR function of OpenMS, MaxQuant and MSFragger to calculate the ratio of all *E. coli* peptides between specific samples (15 ng/10 ng, 20 ng/10 ng, and 25 ng/10 ng) in each replicate.

As shown in **Fig. 3**, we found that the combination of using MaxQuant to extract features, DeepRTAlign to align features and MSFragger to identify their peptide sequences could obtain 150% more peptides compared with the original MaxQuant workflow (using MaxQuant to extract features, align them and identify their peptide sequences) without compromising the quantification accuracy. DeepRTAlign also aligned 7% and 14.5% more peptides than OpenMS and MSFragger's MBR, while its quantification accuracy was comparable with other methods. Overall, we found that OpenMS extracted features represented a slightly better accuracy despite the alignment methods. This may be because OpenMS has strict quality controls in feature extraction. However, DeepRTAlign can still increase the number of aligned peptides based on OpenMS extracted features, showing a higher sensitivity than OpenMS in terms of alignment.

It should be noted that DeepRTAlign is compatible with multiple feature extraction methods (Dinosaur, MaxQuant, OpenMS and XICFinder), which is convenient for users to choose the feature extraction method suitable for their own experiments.

Fig. 3. The number and ratio distributions of all *E. coli* peptides between specific samples (15ng/10ng, 20ng/10ng, and 25ng/10ng) in each replicate (R1, R2 and R3) after alignment. In all boxplots, the center black line is the median of log₂ of the peptide fold changes. The box limits are the upper and lower quartiles. The whiskers extend to 1.5 times the size of the interquartile range.

M2. DeepRTAlign description

Code inspection reveals that the current description of DeepRTAlign, although nicely written, can be further improved.

Training Part

* Some details are missing in the description of the DNN and its hyperparameters:

- The text does not mention the sigmoid activation.
- It should be specified which initialization scheme the authors used for the weights and biases.
- The authors should mention the optimizer used and, if it makes use of hyperparameters, these should be described.
- The authors should explain how they selected the number of epochs.

Response: We thank the reviewer for the helpful suggestions. The detailed descriptions have been added in the “**Workflow of DeepRTAlign**” section in this revision:

The sigmoid function was used as the activation function. We used the default initialization method of pytorch (kaiming_uniform). We used Adam (betas=(0.9, 0.999),

eps=1e-08, weight_decay=0, amsgrad=False, foreach=None, maximize=False, capturable=False, differentiable=False, fused=None) as the optimizer. The initial learning rate is set to 0.001, and is multiplied by 0.1 every 100 epochs. Batch size is set to 500. The number of epochs is chosen empirically. We conducted several trial runs and found that the loss tended to be stable at 100-300. So we set the epoch number to 400 for final training.

Application Part

* The code implementation loads a matrix from an npy file to normalize the features feeding the neural network. However, this is not clearly described in the text.

Response: We thank the reviewer for the helpful suggestions. We randomly select a feature from training data (HCC-T) to normalize all the features feeding to DNN. In the new version of DeepRTAlign, we used two base vectors (base_1 is [5,0.03], and base_2 is [80,1500]) to do the same things. So, there is no need to read the npy file in the new version anymore. The base vectors refer to the value range of features from HCC-T. The detailed descriptions have been added in the “**Workflow of DeepRTAlign**” section in this revision.

* Starting at line 188, the manuscript states: "In this part, feature lists will first go through the coarse alignment step and the input vector construction step as same as those in the training part. Then, the constructed input vectors will be fed into the trained DNN model". However, the code reveals that between the coarse alignment and the predictions of the DNN there is a binning process based on m/z, which is not obvious from the text. I would suggest to explicitly describe this process on the manuscript to enhance clarity.

Response: We apologize for missing these details. After coarse alignment, all features will be grouped based on m/z, according to the parameters bin_width and bin_precision (0.03 and 2 by default). Bin_width is the m/z window size, and bin_precision is the number of decimal places used in this step. Only the features in the same m/z window will be aligned. After the binning step, there is also an optional filtering step. For each sample in each m/z window, only the feature with the highest intensity will be kept in the user-defined RT range. Then the DNN model in DeepRTAlign will align the features in each m/z window. The detailed descriptions have been added in this revision's “**Workflow of DeepRTAlign**” section (**Page 7**).

* Furthermore, the "bin_width" and "bin_precision" parameters involved in this step seem to have a large impact on the alignment process (based on my own experiments). Although this is somewhat expected by the meaning of the parameters, a brief discussion on how they should be chosen is suggested, due to the large impact of the results. If different values were used during tests, it should be reported on the manuscript.

Response: We thank the reviewer for the kind suggestion. After coarse alignment, all features will be grouped based on m/z according to the parameters bin_width and bin_precision. Bin_width is the m/z window size, and bin_precision is the number of decimal places used in this step. Only the features in the same m/z window will be aligned. In this work, we used 0.03 Da for bin_width and 2 for bin_precision as the default values in all the tests. Our original manuscript used RT and intensity as features and bin_width and bin_precision represented mass accuracy. Thus, they were two important parameters. But in this revision, following **Reviewer #1's Q3 and Q4** suggestions, we removed intensity from the model features but added m/z as a new model feature (**Fig. 1**). Thus, the importance of the two parameters has decreased. Users can set the m/z window wider to ensure the completeness of the alignment, but we do not recommend exceeding 0.03 Da as it will introduce more interfering ions. On the contrary, the m/z window can be appropriately narrowed if we use high-quality MS data.

M3. Code availability

The Github repository is well-structured and can be easily read. However, the authors only provide code for the "application part" of the tool. That is, the validation experiments can't be reproduced without substantial work by other researchers, which hinders reproducibility and comparison with future alignment methods. It would be nice if the authors share the whole codebase for reproducing all the tests conducted in the study. The code for the tests suggested in M1 should be shared.

Response: We thank the reviewer for the positive remarks. We have uploaded all the codes for reproducing all the experiments in this work to GitHub (<https://github.com/PHOENIXcenter/deeptalign>).

M4. DeepRTAlign computational performance

The code provided by the author does a lot of disk operations, which may harm the computational running time of the algorithm. It is crucial that any new alignment

method seeking to make a substantial contribution should be highly performant, as it must compete with the current state-of-the-art tools. Wall-clock runtime for the DeepRTAlign method should be provided for, at least, the benchmarks suggested in M1.

Response: We thank the reviewer for this valuable suggestion. Indeed, the codes in our original manuscript contained many disk operations to write down the intermediate data, which was not efficient enough. In the latest version of DeepRTAlign, we directly stored all the intermediate data in the memory by default. We also kept the hard disk mode, which allows users to check intermediate results conveniently.

As shown in **Table R5**, we evaluated the wall-clock runtime of OpenMS, LWBMatch and DeepRTAlign on dataset Benchmark-FC using the same computer server. Since the direct-matching methods (a part of DeepRTAlign and LWBMatch) need to do a lot of pairwise comparisons between any two features within a specific m/z window and RT window, they are usually slower than the warping function-based methods (such as OpenMS), especially for large scale samples. Meanwhile, considering OpenMS and LWBMatch were coded using C++, a much more efficient programming language than Python (we used in DeepRTAlign), we think there is still room to improve the efficiency of DeepRTAlign. In this revision, we provide a multi-process mode (-pn parameter) in DeepRTAlign for large dataset. In the future, we plan to develop a new version of DeepRTAlign coded by C++.

Table R5. The wall-clock runtime (min) of three alignment tools on Benchmark-FC dataset using a server (Intel® Xeon™ Platinum 9242 CPU @ 2.30GHz). The input of all three tools are features extracted by OpenMS. DeepRTAlign uses 10 processes (-pn 10). The average and standard deviation of the runtime are calculated based on all the sample pairs in Benchmark-FC.

Method	OpenMS	LWBMatch	DeepRTAlign
Runtime (min)	4±1	217±24	156±28

Minor comments (organized by sections)

Workflow of DeepRTAlign

* "All other parameters are kept by default in Pytorch". Default parameters can change (or even disappear!) as software evolves. I would prefer to read the details explained in a software-agnostic way. If not possible, the pytorch version should be stated here for the reader's convenience (although it is true that it appears later in the text).

Response: We are so sorry for the careless descriptions. We have added the pytorch version and the detailed descriptions of the all the parameters in this revision (“**Workflow of DeepRTAlign**” section, **Page 7**):

(5) Deep neural network (DNN). The DNN model in DeepRTAlign contains three hidden layers (each has 5000 neurons), which is used as a classifier that distinguishes between two types of feature-feature pairs (i.e., the two features should be or not be aligned). Finally, a total of 400,000 feature-feature pairs were collected from the HCC-T dataset based on the Mascot identification results (mass tolerance: ± 10 ppm, restrict the RT of a peptide to be within the RT range of the corresponding precursor feature). 200,000 of them are collected from the same peptides, which should be aligned (labeled as positive). The other 200,000 are collected from different peptides, which should not be aligned (labeled as negative). These 400,000 feature-feature pairs were used to train the DNN model. It should be noted that it is not necessary to know the peptide sequences corresponding to the features when performing feature alignment. The identification results of several popular search engines (such as Mascot, MaxQuant and MSFragger) are only used as ground truths when benchmarking DeepRTAlign.

(6) The hyperparameters in DNN. BCELoss function in Pytorch is used as the loss function. The sigmoid function was used as the activation function. We used the default initialization method of pytorch (kaiming_uniform). We used Adam (betas=(0.9, 0.999), eps=1e-08, weight_decay=0, amsgrad=False, foreach=None, maximize=False, capturable=False, differentiable=False, fused=None) as the optimizer. The initial learning rate is set to 0.001, and is multiplied by 0.1 every 100 epochs. Batch size is set to 500. The number of epochs is chosen empirically. We conducted several trial runs and found that the loss tended to be stable at 100-300. So, we set the epoch number to 400 for final training. All other parameters are kept by default in Pytorch v1.8.0.

Machine learning models for evaluation

* It should be clearly stated if the only difference between DeepRTAlign and the other methods is the substitution of the neuronal network by the proper classifier, or if there is any other further difference.

Response: We would like to clarify that DeepRTAlign is different from the other alignment methods in three aspects: (1) DeepRTAlign includes a coarse alignment step (see “**Workflow of DeepRTAlign**” section for details). (2) The classifier of DeepRTAlign is a deep neuron network (DNN) model rather than another machine learning-based model. (3) DeepRTAlign combines both advantages of the warping function-based and direct-matching methods in theory. Its coarse alignment function can be regarded as a simplified warping function, and its DNN model can be used for direct matching alignment. (4) To further improve the alignment accuracy, DeepRTAlign considers the feature to be aligned and its adjacent features. We found that if a feature (such as feature n in sample 1) is correctly aligned to another feature (such as feature m in sample 2), the similarity of feature n and the adjacent feature of m (such as m-1 or m+1) is more likely to be low.

* A few words describing the parameter tuning process should be included.

Response: We thank the reviewer for this suggestion. Here is the tuning process: The AUCs of different DNN hidden layer numbers (each layer has 5000 neurons) in DeepRTAlign when performing 10-fold cross-validation on the training set HCC-T were shown in **Supplementary Table 2a**. Based on the results, we chose 3 hidden layers for the model in DeepRTAlign. The AUCs of different neuron numbers in DeepRTAlign based on 10-fold cross-validation on the training set HCC-T were shown in **Supplementary Table 2b**. All the models have 3 hidden layers. According to these results, we chose 5000 neurons in each layer for the model in DeepRTAlign.

* What is the difference between the number of estimators (100) and the tree number in the forest (10)?

Response: We are sorry that the tree number in the forest (10) is a typo here. Based on 10-fold cross validation on the training set HCC-T, we optimized the hyperparameters of RF, KNN, SVM and LR in this revision (**Supplementary Tables 4-7**). We used `max_depth=50`, `n_estimators=50` for the RF model. All the other parameters were kept default in scikit-learn v0.21.3. The corresponding descriptions have been modified in the “**Machine learning models for evaluation**” section.

*The parameters for each machine learning method were optimized based on the 10-fold cross validation results of the training set HCC-T (**Supplementary Tables 4-7**). The parameters of the RF model are number of estimators 50, and max depth 50. For KNN, the k is set to 5. For SVM, the kernel function is set to 'poly' (gamma='auto', C=1, and*

degree=3). For LR, the penalty is L2 and the solver is “lbfgs”. All the other parameters are default values in scikit-learn v0.21.3.

* The authors should specify which non-linear kernel was used for the SVM. Furthermore, they should state whether the SVM's hyperparameters were tuned. If not, I would suggest optimizing them since SVMs are quite sensitive to the hyperparameters configuration.

Response: We thank the reviewer for this suggestion. Based on 10-fold cross validation results of the training set HCC-T, we optimized the hyperparameters of RF, KNN, SVM and LR in this revision (**Supplementary Tables 4-7**). For SVM, parameters kernel='poly', gamma='auto', C=1, and degree=3 were used.

Tools for alignment comparison

* The software tools MZmine and OpenMS offer numerous alignment methods. However, to ensure transparency and reproducibility, the authors should clearly specify which methods they employed in their analysis.

Response: We thank the reviewer for the kind suggestion. For MZmine 2, the ADAP-LC workflow was used. For OpenMS, we used FeatureFinderCentroided for feature extraction and MapAlignerPoseClustering for feature alignment. After alignment, we used FeatureLinkerUnlabeled and TextExporter to get the results. We have added the detailed descriptions to the “**Tools for alignment comparison**” section (**Page 9**).

* The precision and recall formulas should be reviewed and better explained. It seems that A_k and G_k are sets, but this is not obvious and, in that case, the cardinality of the sets is missing from the formulas.

Response: We thank the reviewer for pointing out this mistake. We have corrected the formulas and added a clearer description into the “**Tools for alignment comparison**” section in this revision (**Page 9**).

The precision formula is:

$$\text{Precision} = \frac{1}{N} \sum_{k=1}^N \frac{|A_k \cap G_k|}{|A_k|}$$

The recall formula is:

$$\text{Recall} = \frac{1}{N} \sum_{k=1}^N \frac{|A_k \cap G_k|}{|G_k|}$$

N is the sample pair number. A_k is a set containing all the aligned feature pairs in the k_{th} sample pair. G_k is a set containing all the ground truth of the k_{th} sample pair. $|X|$ indicates the number of elements in the set X .

Model evaluation on the training set and the test sets

* Line 281. "We think this is because the RT shift density distribution of UPS2-Y is similar to HCC-T (Figure S2)". Does "RT shift" refer to the coarse alignment step? If yes, why this explains the better performance of RF? Isn't the coarse alignment step shared between the DNN (DeepRTAlign) and the RF version?

Response: We apologize for the unclear descriptions about the **original Fig. S2** and **Table S4**. The RT shifts in **Fig. S2** were the original RT difference of the same peptide in two samples, calculated from original features without coarse alignment. However, in the original **Table S4**, coarse alignment step is performed before DNN (DeepRTAlign) and machine learning models including RF. **We agree that Fig. S2 cannot explain why RF outperformed DNN on the UPS2-Y dataset in the original Table S4.**

In this revision, according to **Reviewer #2's Q5** suggestions, we have updated Table S4 (denoted as **Supplementary Table 3**) using the new version of DeepRTAlign (new input model features and a new QC module, suggested by **Reviewer #1's Q2, Q3 and Q4**). As shown in the **Supplementary Table 3**, we found DNN outperforms RF in all the test datasets. Please note that in this revision, all the results on test datasets were based on the model trained on the whole training set (HCC-T) with tuned hyperparameters (hyperparameters were optimized based on 10-fold cross validation on the training set HCC-T), no longer from the 10-fold cross validation models on the training set HCC-T. Thus, we did not have to calculate the mean and standard deviation of the precisions and recalls in each test set. Using the new version of DeepRTAlign, we think the original **Fig. S2** is unnecessary, thus delete it in this revision.

Supplementary Table 3. The AUCs on different test sets. All the results are based on the model trained on the HCC-T dataset. In each test set, we randomly selected 10,000 positive and 10,000 negative feature pairs to perform this evaluation.

Dataset	DNN	RF	KNN	SVM	LR
---------	-----	----	-----	-----	----

HCC-N	0.925	0.916	0.656	0.865	0.894
HCC-R	0.933	0.905	0.668	0.901	0.899
UPS2-M	0.979	0.919	0.683	0.896	0.905
UPS2-Y	0.971	0.920	0.702	0.900	0.897
EC-H	0.972	0.938	0.733	0.912	0.944
AT	0.975	0.943	0.785	0.932	0.945
SC	0.917	0.901	0.752	0.842	0.898

* Line 282. "In general, DNN has a better generalization performance and RF can be an alternative solution when computing resources are limited" Can the authors further explain why this occurs? Intuitively, in the application part, the prediction times for the DNN and the RF shouldn't be so different.

Response: We apologize that our unclear descriptions misled the reviewer. In our test, the prediction times of DNN and RF were close. But training the RF model is much faster than training the DNN model. This is why we wrote, "RF can be an alternative solution when computing resources are limited". In this revision, to avoid discrepancy, we have deleted this sentence.

* In Figure 3, I have trouble understanding "DeepRTAlign Shared" (SH_D) and "Quandenser Shared" (SH_Q). I thought "shared" likely referred to the peptides that were identified by both DeepRTAlign and Quandenser algorithms, but the boxplot shows different CVs for SH_D and SH_Q and hence my interpretation must be wrong.

Response: We are sorry for these unclear descriptions about the **original Fig. 3**. "Shared" did refer to the peptides commonly aligned by DeepRTAlign and Quandenser. The boxplots showed the CVs of SH_D and SH_Q using the Dinosaur extracted features. In theory, these CVs should be the same. However, in our original manuscript, we used a mass tolerance of 10 ppm and an RT tolerance of 5 min for matching identifications to features. As suggested by **Reviewer #2's Q10**, a 5-minute RT tolerance is too big. Some peptides may correspond to multiple features. We only kept the most intense features in the DeepRTAlign and Quandenser results. Thus, the chosen feature may be different for the same peptide. That's why the reviewer found the CV boxplots of SH_D and SH_Q were different. Following the suggestion in **Reviewer #2's Q2**, only using CV as a measure of alignment performance is not entirely justifiable. Thus, we compared DeepRTAlign with Quandenser on a new benchmark

dataset (dataset Benchmark-FC) with known fold changes for performance evaluation. Please note that the DeepRTAlign evaluated in this revision is a new version with new input model features and a QC module. As shown in **Fig. R15a-b**, we can find that DeepRTAlign is comparable to Quandenser in terms of the number of aligned peptides and the quantification accuracy. DeepRTAlign can align more features (regardless of corresponding identification results) than Quandenser since it is ID-free (**Fig. R15c**). The original **Fig. 3** is removed and **Fig. R15** is denoted as **Supplementary Fig. 4** in this revision.

Fig. R15. The number and ratio distributions of all *E. coli* peptides and the group number of aligned features between specific samples (a: 15ng/10ng, b: 20ng/10ng, and c: 25ng/10ng) in each replicate (R1, R2 and R3) after alignment by Quandenser and DeepRTAlign. It should be noted that after alignment, the same features in different runs will be put into a group.

Some typos in the text:

* Use of two periods after citations. For example, at lines 109 or 118.

* Line 182: " and the trained model was evaluated..." -> The trained model was also evaluated.

* Lines 187-188: "...can be used after format conversion refer to the formats.." -> ... can be used after format conversion. Refer to the formats ...

* Line 354: "the CV values of DeepRTAlign in all the group are 47.6% smaller in UPS2-M and 58.3% smaller..." -> "the CV values of DeepRTAlign are 47.6% smaller in UPS2-M and 58.3% smaller..."

* Line 361: "indentification" -> identification

* Line 640: "the histograms of ...> -> the barplots of (I don't think they should be referred to as histograms). There are a few more occurrences of "histogram" after this first one.

Response: All these typos have been corrected in this revision.

Supplementary material

* Figure S2. It is hard to compare the distributions of the datasets. Maybe a log scale could be used in the y-axis to enhance the visualization.

Response: Similar to the response to the minor issue “## Model evaluation on the training set and the test sets” of **Reviewer #3**, we think the original Fig. S2 is unnecessary using the new version of DeepRTAlign, thus delete it in this revision.

* Table S10. How are the standard errors computed in Table S10? In all other tables, I suppose the standard errors are computed using 10-fold cross-validation but, in this real dataset, this does not seem to be the case. If 10-fold cross-validation is indeed used with real datasets, then standard errors should be incorporated to other results (for example, error bars could be added to figure 2).

Response: We apologize that our unclear descriptions misled the reviewer. In **the original Table S10**, the average and the standard deviation of the precisions and recalls were calculated between different sample pairs (such as sample 1-sample 2 and sample 2-sample 3) in the test set (SM1100, N=10) rather than the 10-fold cross-validation results. We found that the standard deviations were misleading and thus were removed in this revision. And this table updated using the new version of DeepRTAlign and is denoted as **Supplementary Table 12** in this revision.

Supplementary Table 12. The different algorithm combinations for benchmarking DeepRTAlign against MZmine 2 and OpenMS on a public metabolomic test set SM1100.

Abbreviations	Feature extraction	Feature alignment	Precision	Recall
MM	MZmine 2	MZmine 2	1.000	1.000
MD	MZmine 2	DeepRTAlign	1.000	1.000

OO	OpenMS	OpenMS	1.000	0.980
OD	OpenMS	DeepRTAlign	0.997	0.985
DD	Dinosaur	DeepRTAlign	0.971	0.965

Reviewers' Comments:

Reviewer #1:

Remarks to the Author:

This reviewer appreciated the authors' improvement in both of the manuscript and the program based on reviewers' comments and suggestions. Most of my concerns have been addressed in the revised version, while there are still two key issues to be processed.

1. In the reply of Q3, authors tried to fix the issue by changing the intensity feature to m/z feature. However, m/z feature also has its own problem as false ms1 features may have the same m/z values as true ms1 features. Therefore, a proper comparison between m/z feature and intensity feature is needed.

2. In the reply of Q1, authors used a subset of top-200 MS1 features, I think this is biased as top-200 features are usually high-quality features which are easier to be identified. So it did not well validate the proposed method itself. I wondered how accurate it will be if evaluating all aligned MS1 features at 1% FDR. I believe it is better to randomly select aligned pairs for validation.

Reviewer #2:

Remarks to the Author:

Thank you for addressing the concerns. Liu Yi et al. have made a significant improvement by using m/z instead of intensity, removing alignment bias towards intense peaks. They tested their approach on a fold-change dataset, Benchmark-FC and clinical cohort, showing its usefulness in predicting clinical responses, such as recurrence of HCC. However, the technical explanation of their algorithms seems weak. Given the introduction of a new method, at least a standard evaluation is needed, if not exhaustive.

Their use of neighboring signals and deep neural networks is innovative. Yet, the results are comparable to existing tools, slightly better in some cases. As demonstrated in Supplementary Table 3, the performance is better than logistic regression (LR), however it follows a similar pattern as LR across datasets. In Figure 3 and Figure R15, a comparison with OpenMS and Dinosaur reveals a slight performance advantage in favor of the proposed method. This modest improvement, in my opinion, may be attributed to the authors' limited utilization of the neural network's capabilities, as they primarily rely on RT and m/z values. In contrast, other tools make use of readily available MS/MS and intensity information. Nonetheless, it's worth noting that software developers might argue that a mere 5% increase in performance could be achieved with some parameter adjustments. Nonetheless, using only two parameters, the authors have shown to achieve results comparable to standard tools.

I suggest performing a technical analysis with existing data to understand the alignment approach. Here are my comments:

1. In Figure 3, the lower panel shows that there is little discernible difference between OpenMS, Dinosaur, and DeepRTAlign. In cases of improved quantitation, one would anticipate a noticeable shift in the median towards the dotted line and a decrease in the lower whisker height. However, this expected change is not observed. It's worth noting that the claim of a 150% increase in peptides in MaxQuant appears to be due to the presence of potentially spurious peptides, as the whisker height increases with alignment. Therefore, Figure 3 suggests that DeepRTAlign performs comparably to existing alignment methods.

2. The authors should conduct a more in-depth examination of the peaks selected after alignment. One useful plot to include in Figure 3 would be the fold-change vs. intensity before and after alignment. This plot can help determine whether the alignment process primarily adds low-intensity peaks or corrects high-intensity peaks. Such FC vs. intensity plots are standard in LFQbench studies and would enhance the persuasiveness of the approach to readers. Additionally, the authors could explore a FC vs. m/z plot to check for any potential bias towards m/z values. While such bias is not expected, including this as a sanity check would be beneficial.

3. The y-axis of the lower panel should be adjusted to a range of -1 to 3 in order to better

visualize the medians.

4. In the manuscript, the discussion of Figure 3 primarily revolves around peptide identification, rather than delving into fold-change and other quantitative metrics. To provide a more comprehensive analysis, the authors are encouraged to explore and discuss the quantitative aspects of this technical dataset in detail in the manuscript.

5. Table R3 highlights a feature pair that OpenMS could not successfully align, but the current listing lacks sufficient information to comprehend the reasons behind OpenMS/MzMine2's failure. To provide a clearer understanding of this issue, the authors have the option to either present chromatographic traces for the specified m/z values or create an RT-MZ map with intensity represented as heatmaps. Such visualizations would elucidate how neighboring features contribute to alignment, a capability not exhibited by OpenMS. This is just my suggestion, any other way to visualize the input to DNN and alignment outcome is also welcome.

6. In the last Figure 5, authors have demonstrated that the tool is useful in fetching features that can be used for training models. However, since the paper is focused on technical advancements, one good comparison would be how does the model fare with features from unaligned? Currently, they use aligned data and select 200 features which were filtered further through validation cohort and selected 15 features. It would be good to know if the same process is done for unaligned, would we miss these features? or how many would we get out of 15 in top 200. This would illustrate the advantages of the proposed alignment method in the context of biological research. It is just a suggestion, authors may opt not to do it.

Reviewer #3:

Remarks to the Author:

The reviewer would like to thank the authors for performing additional studies on their proposed method to provide additional insights. The authors have put considerable efforts to address the comments of all reviewers, further improving the quality of the paper. While the authors have made significant progress in addressing my previous concerns, I still have a few minor questions, particularly regarding the comparison between the neural network and other machine learning methods.

#1. In response to my initial minor comment in the section titled "Machine Learning Models for Evaluation," the authors mentioned that other machine learning models do not incorporate the coarse alignment step or the use of adjacent features. My concern here is that if the different machine learning methods employ dissimilar sets of features, it becomes challenging to draw meaningful comparisons. It is crucial to establish whether observed differences in performance are attributable to (1) the choice of features or (2) the specific algorithm used. In my view, ensuring that all algorithms use the same inputs is essential for a fair comparison and the experiments should be rerun.

#2. Building on the previous point, it is essential that the experiments conclusively demonstrate whether the DNN offers any advantages over other algorithms. If such advantages exist, the paper should provide a clear explanation for them. Based on my experience and considering the features employed by the authors, it appears unlikely that a DNN would significantly outperform, for example, a Random Forest (RF). DNNs are typically advantageous when processing raw data like text or images, as they can autonomously extract relevant features. However, for tabular data, RF and boosting methods often outperform DNNs, and this is a common observation in machine learning competitions of this nature.

#3. In Supplementary Tables 4-7, it appears that hyperparameters were tuned sequentially. This approach may result in suboptimal parameter combinations, particularly for complex classifiers like RF and Support Vector Machines (SVMs). Building on the previous point (point #1), if the authors choose to re-tune the hyperparameters with the new features as suggested, I would recommend employing a grid search for both RF and SVMs. This approach is not necessary for LR and KNN since they only involve a single main parameter.

Point-by-point response to reviewers' comments

We appreciate the reviewers' insightful and constrictive comments as they have helped us improve the algorithm and the manuscript. Additional data analysis and manuscript revision have been performed to address the reviewers' comments. In this response, the reviewers' comments are colored in black, and our replies to the comments are colored in blue.

Reviewer #1 (Remarks to the Author):

This reviewer appreciated the authors' improvement in both of the manuscript and the program based on reviewers' comments and suggestions. Most of my concerns have been addressed in the revised version, while there are still two key issues to be processed.

1. In the reply of Q3, authors tried to fix the issue by changing the intensity feature to m/z feature. However, m/z feature also has its own problem as false ms1 features may have the same m/z values as true ms1 features. Therefore, a proper comparison between m/z feature and intensity feature is needed.

Response: We thank the reviewer for this constructive suggestion. It is known that MS data consists of three dimensions: RT, m/z and intensity (despite that MS data from timsTOF Pro additionally contain ion mobility as a fourth dimension). **In theory, the information in RT dimension and m/z dimension are the most relevant factors for accurate RT alignment.** The information in intensity dimension was used in the original version of DeepRTAlign to assist RT alignment. However, as the reviewer pointed out in the first round of review (Q3 and Q4), the intensity dimension is only useful when the intensities of two features are equal or close, making DeepRTAlign difficult to align the regulated MS features. We realized that even if the intensity feature performs well in some situations, we should avoid using it in theory. Thus, we changed the intensity feature to m/z feature in the last revision.

In this revision, following the reviewer's suggestion, we compared the feature combinations of intensity+RT and m/z+RT in seven test sets. As shown in **Table R1**, when using random feature to replace intensity or m/z, the performances of model

“Intensity+RT” and model “m/z+RT” were close and they did decrease much compared with the original model (none of the features were replaced by random values) in the “None (Original)” column. On the contrary, when using random value to replace RT, the performance of model “Intensity+RT” dropped more than that of model “m/z+RT” in all the test sets. These results showed that m/z is more suitable than intensity as a feature when combining with RT. In addition, we agree with the reviewer that some false positive results in DeepRTAlign (the features should not be aligned but falsely aligned by DeepRTAlign) may have similar or same m/z values. This is actually one of the main reasons for false positive alignments, which is a major limitation for all the MS1-based alignment methods. We could improve the accuracy of alignment by using additional features (such as ion mobility) if possible.

Table R1. The AUCs of DeepRTAlign when using random values to replace the original features in the test sets. All the results are based on the models trained on the HCC-T dataset. “Intensity+RT” indicates the model considering the information of intensity and RT dimensions, i.e., the previous version of DeepRTAlign model we used in the original submission. And “m/z+RT” indicates the model considering the information of m/z and RT dimensions, i.e., the latest version of DeepRTAlign model we used in this revision.

Model	Intensity+RT			m/z+RT		
	Intensity	RT	None (Original)	m/z	RT	None (Original)
HCC-N	0.901	0.825	0.985	0.899	0.841	0.924
HCC-R	0.897	0.774	0.944	0.895	0.855	0.930
UPS2-M	0.955	0.811	0.982	0.952	0.932	0.979
UPS2-Y	0.960	0.776	0.972	0.958	0.941	0.970
EC-H	0.954	0.660	0.957	0.959	0.942	0.970
AT	0.965	0.766	0.966	0.964	0.950	0.972
SC	0.897	0.741	0.935	0.901	0.831	0.917

2. In the reply of Q1, authors used a subset of top-200 MS1 features, I think this is

biased as top-200 features are usually high-quality features which are easier to be identified. So it did not well validate the proposed method itself. I wondered how accurate it will be if evaluating all aligned MS1 features at 1% FDR. I believe it is better to randomly select aligned pairs for validation.

Response: We apologize that our unclear descriptions may have misled the reviewer. We would like to clarify that top-200 MS1 features for classifying the HCC patients between the early recurrence group and late recurrence group were **selected from all the aligned features based on mRMR importance scores**, rather than top-200 abundant MS1 features (**Figure 5a**). The intensities of the top-200 MS1 features selected by mRMR were shown in **Figure R1**. **We can find that the 200 MS1 features were evenly distributed from high abundance to low abundance**. In this study, we would also like to clarify that following the reviewers' suggestion in the last revision, we used PRM to verify the candidate markers (the aligned MS1 features which have the potential ability to distinguish HCC early recurrence patients from late recurrence patients), rather than verify all the aligned pairs.

Figure R1. All the features aligned by DeepRTAlign are displayed with their average intensities in the training set (HCC-T dataset) against their intensity ranks. Top 200 features based on mRMR scores are highlighted in orange.

Reviewer #2 (Remarks to the Author):

Thank you for addressing the concerns. Liu Yi et al. have made a significant improvement by using m/z instead of intensity, removing alignment bias towards intense peaks. They tested their approach on a fold-change dataset, Benchmark-FC and clinical cohort, showing its usefulness in predicting clinical responses, such as recurrence of HCC. However, the technical explanation of their algorithms seems weak. Given the introduction of a new method, at least a standard evaluation is needed, if not exhaustive.

Their use of neighboring signals and deep neural networks is innovative. Yet, the results are comparable to existing tools, slightly better in some cases. As demonstrated in Supplementary Table 3, the performance is better than logistic regression (LR), however it follows a similar pattern as LR across datasets. In Figure 3 and Figure R15, a comparison with OpenMS and Dinosaur reveals a slight performance advantage in favor of the proposed method. This modest improvement, in my opinion, may be attributed to the authors' limited utilization of the neural network's capabilities, as they primarily rely on RT and m/z values. In contrast, other tools make use of readily available MS/MS and intensity information. Nonetheless, it's worth noting that software developers might argue that a mere 5% increase in performance could be achieved with some parameter adjustments. Nonetheless, using only two parameters, the authors have shown to achieve results comparable to standard tools. I suggest performing a technical analysis with existing data to understand the alignment approach.

Response: We thank the reviewer for the positive comment and valuable suggestion. We agree with reviewer that we should enhance the technical explanation of our algorithm in this revision. As shown in **Supplementary Table 3**, RF showed the best performance in the traditional machine learning models, except DNN. Here, we would like to discuss the advantages of DNN over RF in the aspects of data and theory.

First, we randomly collected 2000 negative pairs correctly predicted by both DNN and RF (BOTH-right pairs), and 2000 negative pairs correctly predicted by DNN but wrongly predicted by RF (DNN-right pairs) in a test set (HCC-N, RT range: 1-80 min, m/z tolerance: 0.03 Da, the same way as we collected the negative pairs in the training

set HCC-T, see “Workflow of DeepRTAlign” section in the manuscript for details). We found that the average RT differences of the 2000 BOTH-right pairs and the 2000 DNN-right pairs were similar (5.03 min and 4.89 min, respectively). However, the average m/z difference of the 2000 BOTH-right pairs was 0.011 Da, which was five times the average m/z differences of the 2000 DNN-right pairs (0.002 Da). **These results indicated that the advantages of DNN may mainly lie in recognizing the negative pairs with close m/z values.**

Second, we found that those negative pairs with similar m/z values (with differences less than 0.002 Da) were rare cases in the whole training data (**10% only**). **Traditional machine learning methods usually are difficult to discriminate and predict correctly in such a situation.** Although the bootstrapping-based RF is supposed to alleviate the problem such as imbalanced data in some degree, its classification power will drastically drop when the minority is extremely rare (Xu, Z., Shen, D., Nie, T., & Kou, Y. (2020). A hybrid sampling algorithm combining M-SMOTE and ENN based on Random forest for medical imbalanced data. *Journal of biomedical informatics*, 103465). The minor pairs might be scarcely sampled during the bootstrapping and their effects on trees are overwhelmed by the majority of the data. **In contrast, the DNN model is trained with mini-batches iteratively through gradient descent, which guarantees each pair, including the rare cases, is learned through a shuffled order. Particularly, those wrongly predicted rare cases lead to greater loss values and thus greater gradients to update the model, which may explain why DNN can predict them better.**

Third, for the 40 features (5×8 vector, **Supplementary Fig. 1**) we used in DeepRTAlign, we calculated the importance of each feature in the DNN model, the RF model and the LR model to show which feature is more important when training the DNN, RF and LR models on the same training set (HCC-T). Please note that for DNN, the importance of each feature in DNN is calculated as the AUC decrease tested in the HCC-N dataset when each feature in DNN was replaced with a random value, compared to the DNN model without random features. For RF, the feature importance is the attribute “feature_importances_” in `sklearn.ensemble.RandomForestClassifier`

(scikit-learn framework v0.21.3). For LR, the feature importance is the attribute “coef_” in sklearn.linear_model.LogisticRegression (scikit-learn framework v0.21.3).

We found that feature 6 (mz_n-mz_m) and feature 16 (mz_m-mz_n) were the top 2 important features, indicating that DNN, RF and LR models rely mostly on the features about m/z for alignment prediction. We also found the importances of feature 5 (RT_n-RT_m) and feature 15 (RT_m-RT_n) in DNN are about half those of feature 6 (mz_n-mz_m) and feature 16 (mz_m-mz_n), while the ratio (importance of feature 5/importance of feature 6) decreased to only about 32.7% in RF and 0.72% in LR. This difference indicates that the importance of RT-related features in the DNN model is relatively higher than that in the RF and LR models (**Table R2**, denoted as **Supplementary Table 11** in this revision).

In summary, we have added more detailed descriptions and discussions about the technical explanation and analysis of our algorithm in this revision (**Pages 12-13**). We thank the reviewer again for the helpful suggestion.

Table R2 (Supplementary Table 11 in this revision). The list of feature importance of the DNN model, the RF model and the LR model. The DNN model, the RF model and the LR model were trained on the same training set (HCC-T dataset). Please note that the feature importance of LR model is ranked by the absolute value of “coef_”.

DNN			RF			LR		
index	importance	Features	index	importance	Features	index	importance	Features
16	0.107	mz_m-mz_n	16	0.205	mz_m-mz_n	16	8.804	mz_m-mz_n
6	0.107	mz_n-mz_m	6	0.199	mz_n-mz_m	6	8.803	mz_n-mz_m
5	0.048	RT_n-RT_m	5	0.065	RT_n-RT_m	27	-2.435	RT_{n+1}
15	0.048	RT_m-RT_n	15	0.058	RT_m-RT_n	21	-2.294	RT_{n-2}
11	0.026	$RT_{m-2}-RT_n$	11	0.030	$RT_{m-2}-RT_n$	25	1.129	RT_n
13	0.024	$RT_{m-1}-RT_n$	35	0.025	RT_m	37	0.928	RT_{m+1}
1	0.023	$RT_{n-2}-RT_m$	25	0.023	RT_n	29	0.854	RT_{n+2}
17	0.017	$RT_{m+1}-RT_n$	13	0.018	$RT_{m-1}-RT_n$	31	0.679	RT_{m-2}
10	0.010	$mz_{n+2}-mz_m$	19	0.017	$RT_{m+2}-RT_n$	23	0.415	RT_{n-1}
4	0.010	$mz_{n-1}-mz_m$	17	0.016	$RT_{m+1}-RT_n$	35	0.397	RT_m
31	0.006	RT_{m-2}	37	0.016	RT_{m+1}	17	0.234	$RT_{m+1}-RT_n$
21	0.006	RT_{n-2}	27	0.015	RT_{n+1}	32	-0.216	mz_{m-2}
8	0.005	$mz_{n+1}-mz_m$	39	0.015	RT_{m+2}	30	-0.216	mz_{n+2}

14	0.005	mZ _{m-1} -mZ _n	7	0.014	RT _{n+1} -RT _m	34	-0.216	mZ _{m-1}
9	0.004	RT _{n+2} -RT _m	9	0.013	RT _{n+2} -RT _m	38	-0.216	mZ _{m+1}
3	0.002	RT _{n-1} -RT _m	36	0.013	mZ _m	24	-0.216	mZ _{n-1}
7	0.002	RT _{n+1} -RT _m	23	0.013	RT _{n-1}	40	-0.216	mZ _{m+2}
18	0.002	mZ _{m+1} -mZ _n	12	0.012	mZ _{m-2} -mZ _n	36	-0.216	mZ _m
34	0.002	mZ _{m-1}	1	0.012	RT _{n-2} -RT _m	26	-0.216	mZ _n
20	0.001	mZ _{m+2} -mZ _n	14	0.012	mZ _{m-1} -mZ _n	22	-0.216	mZ _{n-2}
36	0.001	mZ _m	3	0.012	RT _{n-1} -RT _m	28	-0.216	mZ _{n+1}
22	0.001	mZ _{n-2}	10	0.012	mZ _{n+2} -mZ _m	18	-0.210	mZ _{m+1} -mZ _n
33	0.001	RT _{m-1}	32	0.012	mZ _{m-2}	4	-0.185	mZ _{n-1} -mZ _m
38	0.001	mZ _{m+1}	34	0.012	mZ _{m-1}	11	-0.163	RT _{m-2} -RT _n
19	0.001	RT _{m+2} -RT _n	21	0.011	RT _{n-2}	19	-0.141	RT _{m+2} -RT _n
28	0.001	mZ _{n+1}	38	0.011	mZ _{m+1}	33	-0.101	RT _{m-1}
40	0.001	mZ _{m+2}	18	0.011	mZ _{m+1} -mZ _n	20	0.080	mZ _{m+2} -mZ _n
26	0.001	mZ _n	22	0.011	mZ _{n-2}	7	-0.080	RT _{n+1} -RT _m
30	0.000	mZ _{n+2}	29	0.011	RT _{n+2}	8	-0.074	mZ _{n+1} -mZ _m
32	0.000	mZ _{m-2}	30	0.010	mZ _{n+2}	14	-0.073	mZ _{m-1} -mZ _n
25	0.000	RT _n	2	0.010	mZ _{n-2} -mZ _m	5	0.063	RT _n -RT _m
23	0.000	RT _{n-1}	33	0.010	RT _{m-1}	15	0.063	RT _m -RT _n
24	0.000	mZ _{n-1}	20	0.010	mZ _{m+2} -mZ _n	39	0.058	RT _{m+2}
2	0.000	mZ _{n-2} -mZ _m	8	0.010	mZ _{n+1} -mZ _m	9	-0.045	RT _{n+2} -RT _m
12	0.000	mZ _{m-2} -mZ _n	40	0.010	mZ _{m+2}	2	-0.040	mZ _{n-2} -mZ _m
27	0.000	RT _{n+1}	24	0.010	mZ _{n-1}	1	0.036	RT _{n-2} -RT _m
29	0.000	RT _{n+2}	26	0.009	mZ _n	13	0.036	RT _{m-1} -RT _n
35	0.000	RT _m	28	0.009	mZ _{n+1}	10	-0.026	mZ _{n+2} -mZ _m
37	0.000	RT _{m+1}	31	0.009	RT _{m-2}	12	-0.023	mZ _{m-2} -mZ _n
39	0.000	RT _{m+2}	4	0.009	mZ _{n-1} -mZ _m	3	0.010	RT _{n-1} -RT _m

Here are my comments:

1. In Figure 3, the lower panel shows that there is little discernible difference between OpenMS, Dinosaur, and DeepRTAlign. In cases of improved quantitation, one would anticipate a noticeable shift in the median towards the dotted line and a decrease in the lower whisker height. However, this expected change is not observed. It's worth noting that the claim of a 150% increase in peptides in MaxQuant appears to be due to the presence of potentially spurious peptides, as the whisker height increases with

alignment. Therefore, Figure 3 suggests that DeepRTAlign performs comparably to existing alignment methods.

Response: We thank the reviewer for this comment. A complete identification and quantification workflow of MS data involves three major steps, i.e., feature extraction, feature alignment, and feature identification. (1) After feature extraction, the intensity, RT and m/z values of each MS1 feature (potential peptide) are obtained and the extracted ion chromatogram (XIC) for each MS1 feature is constructed. The area under the XIC curve is usually used as the quantification result (abundance) of an MS1 feature in a single sample. (2) Feature alignment between samples is a prerequisite for the subsequent analysis. Because unaligned features are incomparable between samples. Feature alignment can also reduce the missing values between samples especially for MS data analysis with multiple samples or replicates. (3) Feature identification means assigning the corresponding peptide sequence to each aligned feature. Peptide sequences are usually identified by database search engines (such as MSFragger and Mascot). **In theory, feature extraction has a major influence on quantification accuracy. While, our work focuses on the alignment step, whose main usage is to improve the identification sensitivity and reduce missing values for large-scale samples. We think the main contribution of our work is providing a universal and flexible alignment algorithm (DeepRTAlign) that can significantly reduce missing values by aligning more features without compromising the quantification accuracy** compared to existing alignment methods (which has been demonstrated by the benchmarking results).

Large-cohort proteomic data, especially DDA data, usually contain numerous missing values. In this revision, to demonstrate the power of DeepRTAlign in eliminating the missing values from search results, we benchmarked DeepRTAlign against MSFragger's MBR on a new dataset Benchmark-MV. This dataset is generated by analyzing the mixtures containing four different proportions of HEK 293T peptides (73%, 91%, 97% and 99%) and *E. coli* peptides (27%, 9%, 3% and 1%) using six Orbitrap Exploris 480 instruments, resulting a total of 24 (4*6) RAW data files. We calculated the percentage of missing values (missing rate) of all the identified 3928 *E.*

coli peptides from the 24 RAW data files. As shown in **Figure R2 (Supplementary Fig. 5** in this revision), DeepRTAlign significantly reduced the missing rate to 34%, in comparison to the 75% missing rate for MSFragger's MBR.

Figure R2 (Supplementary Fig. 5 in this revision). Features corresponding to *E. coli* peptides identified in dataset Benchmark-MV by MSFragger with MBR (a) and DeepRTAlign (b). For DeepRTAlign results, features were extracted by Dinosaur, and then aligned by DeepRTAlign. MSFragger's identification results were used to match these features (mass tolerance: ± 10 ppm, RT tolerance: restrict the RT of a peptide to be within the RT range of the corresponding precursor feature).

In terms of the 150% increase in peptide identification compared to MaxQuant with MBR function, we would like to claim that it should be mainly due to the use of MSFragger as the feature identification method. The identification results using MSFragger were much more than MaxQuant results (also shown in **Figure 3**). We agree that it is possible that there were false positives when matching MaxQuant's features to MSFragger's identification results (mass tolerance: ± 10 ppm, RT tolerance: restrict the RT of a peptide to be within the RT range of the corresponding precursor feature). Thus, we have added a QC module in the last revision (**Supplementary Fig. 2**) to ensure the accuracy of alignment. While, we admit that it is hard to achieve significant quantitative improvements only using a novel alignment algorithm. The improvement of quantification accuracy requires to optimize and harmonize all the

three steps (feature extraction, feature alignment, and feature identification), and this is our future goal.

Please note that to access the new dataset Benchmark-MV in this revision shown in **Supplementary Table 1**, we have uploaded all the corresponding RAW data files and identification results to iProX database with the identifier IPX0007319000. The dataset will be public once this manuscript is accepted. During peer review, reviewers can access it by the following link with code.

URL: <https://www.iprox.cn/page/PSV023.html?url=1697544593596n9QC>

Code: ixMK

2. The authors should conduct a more in-depth examination of the peaks selected after alignment. One useful plot to include in Figure 3 would be the fold-change vs. intensity before and after alignment. This plot can help determine whether the alignment process primarily adds low-intensity peaks or corrects high-intensity peaks. Such FC vs. intensity plots are standard in LFQbench studies and would enhance the persuasiveness of the approach to readers. Additionally, the authors could explore a FC vs. m/z plot to check for any potential bias towards m/z values. While such bias is not expected, including this as a sanity check would be beneficial.

Response: We thank the reviewer for this constructive suggestion. We have plotted the figures (FC vs. intensity and FC vs. m/z) suggested by the reviewer in three different groups (15 ng/10 ng, 20 ng/10 ng and 25 ng/10 ng) with three replicates (R1, R2 and R3). We found that no significant biases were shown for DeepRTAlign and other methods in FC vs. intensity plots and FC vs. m/z plots (**Figure R3**).

Figure R3a. The fold change (FC) vs. intensity (upper) and m/z (lower) plots of 15 ng/10 ng samples in replicate 1 (R1). The dotted lines indicate the theoretical FC values of HEK 293T (orange) and *E. coli* (blue).

Figure R3b. The fold change (FC) vs. intensity (upper) and m/z (lower) plots of 15 ng/10 ng samples in replicate 2 (R2). The dotted lines indicate the theoretical FC values of HEK 293T (orange) and E. coli (blue).

Figure R3c. The fold change (FC) vs. intensity (upper) and m/z (lower) plots of 15 ng/10 ng samples in replicate 3 (R3). The dotted lines indicate the theoretical FC values of HEK 293T (orange) and E. coli (blue).

Figure R3d. The fold change (FC) vs. intensity (upper) and m/z (lower) plots of 20 ng/10 ng samples in replicate 1 (R1). The dotted lines indicate the theoretical FC values of HEK 293T (orange) and *E. coli* (blue).

Figure R3e. The fold change (FC) vs. intensity (upper) and m/z (lower) plots of 20 ng/10 ng samples in replicate 2 (R2). The dotted lines indicate the theoretical FC values of HEK 293T (orange) and E. coli (blue).

Figure R3f. The fold change (FC) vs. intensity (upper) and m/z (lower) plots of 20 ng/10 ng samples in replicate 3 (R3). The dotted lines indicate the theoretical FC values of HEK 293T (orange) and E. coli (blue).

Figure R3g. The fold change (FC) vs. intensity (upper) and m/z (lower) plots of 25 ng/10 ng samples in replicate 1 (R1). The dotted lines indicate the theoretical FC values of HEK 293T (orange) and *E. coli* (blue).

Figure R3h. The fold change (FC) vs. intensity (upper) and m/z (lower) plots of 25 ng/10 ng samples in replicate 2 (R2). The dotted lines indicate the theoretical FC values of HEK 293T (orange) and E. coli (blue).

Figure R3i. The fold change (FC) vs. intensity (upper) and m/z (lower) plots of 25 ng/10 ng samples in replicate 3 (R3). The dotted lines indicate the theoretical FC values of HEK 293T (orange) and *E. coli* (blue).

3. The y-axis of the lower panel should be adjusted to a range of -1 to 3 in order to better visualize the medians.

Response: We thank the reviewer for this constructive suggestion. We have updated **Figure 3** as the reviewer suggested.

Figure 3. The number and ratio distributions of all *E. coli* peptides between specific samples (15 ng/10 ng, 20 ng/10 ng, and 25 ng/10 ng) in each replicate (R1, R2 and R3) after alignment. In all boxplots, the center black line is the median of log₂ of the peptide fold changes. The box limits are the upper and lower quartiles. The whiskers extend to 1.5 times the size of the interquartile range.

4. In the manuscript, the discussion of Figure 3 primarily revolves around peptide identification, rather than delving into fold-change and other quantitative metrics. To provide a more comprehensive analysis, the authors are encouraged to explore and discuss the quantitative aspects of this technical dataset in detail in the manuscript.

Response: We thank the reviewer for pointing out the quantification issue. As described in response to Q1, a complete identification and quantification workflow of MS data involves three major steps, i.e., feature extraction, feature alignment, and feature identification. (1) After feature extraction, the intensity, RT and m/z values of each

MS1 feature (potential peptide) are obtained and the extracted ion chromatogram (XIC) for each MS1 feature is constructed. The area under the XIC curve is usually used as the quantification result (abundance) of an MS1 feature in a single sample. (2) Feature alignment between samples is a prerequisite for the subsequent analysis. Because unaligned features are incomparable between samples. Feature alignment can also reduce the missing values between samples especially for MS data analysis with multiple samples or replicates. (3) Feature identification means assigning the corresponding peptide sequence to each aligned feature. Peptide sequences are usually identified by database search engines (such as MSFragger and Mascot). **We can see that feature extraction has a major influence on quantification accuracy. While, our work focuses on the alignment step, whose main usage is to improve the identification sensitivity and reduce missing values for large-scale samples. We think the main contribution of this work is providing a universal and flexible alignment algorithm (DeepRTAlign) that can align more features without compromising the quantification accuracy** compared to existing alignment methods (which has been demonstrated by the benchmarking results).

Large-cohort proteomic data, especially DDA data, usually contain numerous missing values. In this revision, to demonstrate the power of DeepRTAlign in eliminating the missing values from search results, we benchmarked DeepRTAlign against MSFragger's MBR on a new dataset Benchmark-MV. This dataset is generated by analyzing the mixtures containing four different proportions of HEK 293T peptides (73%, 91%, 97% and 99%) and *E. coli* peptides (27%, 9%, 3% and 1%) using six Orbitrap Exploris 480 instruments, resulting a total of 24 (4*6) RAW data files. We calculated the percentage of missing values (missing rate) of all the identified 3928 *E. coli* peptides from the 24 RAW data files. As shown in **Figure R2 (Supplementary Fig. 5** in this revision), DeepRTAlign significantly reduced the missing rate to 34%, in comparison to the 75% missing rate for MSFragger's MBR. Meanwhile, we found that no significant biases were shown in FC vs. intensity plots and FC vs. m/z plots for DeepRTAlign (**Figure R3**).

We admit that it is hard to achieve significant quantitative improvements only using a novel alignment algorithm. The improvement of quantification accuracy requires to optimize and harmonize all the three steps (feature extraction, feature alignment, and feature identification), and this is our future goal. Following the reviewer's kind suggestion, we have added the limitation of DeepRTAlign and our future work in the discussion part in this revision (**Page 20**).

5. Table R3 highlights a feature pair that OpenMS could not successfully align, but the current listing lacks sufficient information to comprehend the reasons behind OpenMS/MzMine2's failure. To provide a clearer understanding of this issue, the authors have the option to either present chromatographic traces for the specified m/z values or create an RT-MZ map with intensity represented as heatmaps. Such visualizations would elucidate how neighboring features contribute to alignment, a capability not exhibited by OpenMS. This is just my suggestion, any other way to visualize the input to DNN and alignment outcome is also welcome.

Response: We thank the reviewer for this constructive suggestion. In this revision, we added a schematic diagram (**Figure R4**) to explain the role of adjacent features. As shown in the left column of **Figure R4**, when a feature (feature n) has a high similarity on m/z and RT with another feature (feature m), but has a low similarity with the adjacent features of feature m, DeepRTAlign will output a high alignment score. On the contrary, DeepRTAlign will output a low alignment score (the right column of **Figure R4**).

Figure R4. A schematic diagram to explain how adjacent features (features $n-2$, $n-1$, $n+1$, $n+2$ and features $m-2$, $m-1$, $m+1$, $m+2$) assist center features (feature n and feature m) in alignment scoring. In the left column, red center features indicate the two features should be aligned. In the right column, the red center feature and the grey center feature indicate the two features should not be aligned.

Then, following the reviewer’s suggestion, we collected the chromatographic traces of the features shown in “**Table R2 and Table R3**” in the last revision by Xcalibur (**Figure R5 and Figure R6**).

“Table R2” in the last revision. One example of a feature pair successfully aligned by DeepRTAlign, but not by MZmine 2. The peptide sequence was identified from the Mascot search results.

Method	sample	m/z	charge	intensity	RT	peptide
DeepRTAlign	sample 1	419.2094	2	2299482	22.543	EESGFLR
DeepRTAlign	sample 2	419.2088	2	4346550	21.57131	EESGFLR
MZmine 2	sample 1	419.2097	2	18040604	22.97301	EFLDGTR
MZmine 2	sample 2	419.2088	2	4346550	21.57131	EESGFLR

Figure R5. The extracted ion chromatogram (XIC) of a feature pair successfully aligned by DeepRTAlign, rather than MZmine 2 in Table R2 in the last revision.

“Table R3” in the last revision. One example of a feature pair successfully aligned by DeepRTAlign, but not by OpenMS. The peptide sequence was identified from the Mascot search results.

Method	sample	m/z	charge	intensity	RT	peptide
DeepRTAlign	sample 1	530.7687	2	1.54E+09	11.97347	QVDQLTNDK
DeepRTAlign	sample 2	530.7673	2	1.19E+09	11.89762	QVDQLTNDK
OpenMS	sample 1	530.7589	2	5.11E+08	11.93935	TCTPTVER
OpenMS	sample 2	530.7673	2	1.19E+09	11.89762	QVDQLTNDK

Figure R6. The extracted ion chromatogram (XIC) of a feature pair successfully aligned by DeepRTAlign, rather than OpenMS shown in Table R3 in the last revision.

6. In the last Figure 5, authors have demonstrated that the tool is useful in fetching features that can be used for training models. However, since the paper is focused on technical advancements, one good comparison would be how does the model fare with features from unaligned? Currently, they use aligned data and select 200 features which were filtered further through validation cohort and selected 15 features. It would be good to know if the same process is done for unaligned, would we miss these features? or how many would we get out of 15 in top 200. This would illustrate the advantages of the proposed alignment method in the context of biological research. It is just a suggestion, authors may opt not to do it.

Response: We thank the reviewer for this suggestion. We apologize that our unclear descriptions may have misled the reviewer. **We would like to clarify that feature alignment between samples is a prerequisite for MS data analysis with multiple samples or replicates.** Unaligned features are incomparable between samples. The commonly used MBR (match between runs) workflow can also be viewed as an identification-based alignment (features in different samples are aligned based on

peptide identifications). The unaligned features could not be used to build this recurrence prediction model. In this work, to illustrate the advantages of the proposed alignment method in the context of biological research, as shown in **Figure 5**, we compared the performances of different models trained on top 200 aligned features, top 200 peptides and top 200 proteins selected by mRMR scores in the same training set. The results showed that the classifier built on aligned features owned the highest AUC (**Figure 5b**).

Reviewer #3 (Remarks to the Author):

The reviewer would like to thank the authors for performing additional studies on their proposed method to provide additional insights. The authors have put considerable efforts to address the comments of all reviewers, further improving the quality of the paper. While the authors have made significant progress in addressing my previous concerns, I still have a few minor questions, particularly regarding the comparison between the neural network and other machine learning methods.

#1. In response to my initial minor comment in the section titled "Machine Learning Models for Evaluation," the authors mentioned that other machine learning models do not incorporate the coarse alignment step or the use of adjacent features. My concern here is that if the different machine learning methods employ dissimilar sets of features, it becomes challenging to draw meaningful comparisons. It is crucial to establish whether observed differences in performance are attributable to (1) the choice of features or (2) the specific algorithm used. In my view, ensuring that all algorithms use the same inputs is essential for a fair comparison and the experiments should be rerun.

Response: We apologize that our unclear descriptions may have misled the reviewer. **When compared DNN with other machine learning methods, the inputs are exactly the same, i.e., the features after coarse alignment step.** The claim of "do not incorporate the coarse alignment step or the use of adjacent features" indicates when we compared DeepRTAlign with existing alignment methods such as OpenMS and MZmine 2, the existing alignment methods did not contain the coarse alignment step

and they did not use the adjacent features. We have modified the corresponding descriptions to make a clear statement (**Page 9**).

#2. Building on the previous point, it is essential that the experiments conclusively demonstrate whether the DNN offers any advantages over other algorithms. If such advantages exist, the paper should provide a clear explanation for them. Based on my experience and considering the features employed by the authors, it appears unlikely that a DNN would significantly outperform, for example, a Random Forest (RF). DNNs are typically advantageous when processing raw data like text or images, as they can autonomously extract relevant features. However, for tabular data, RF and boosting methods often outperform DNNs, and this is a common observation in machine learning competitions of this nature.

Response: We thank the reviewer for the valuable suggestion. In this response, we would like to enhance the technical explanations of our algorithm. As shown in **Supplementary Table 3**, RF showed the best performance in the traditional machine learning models, except DNN. Here, we would like to discuss the advantages of DNN over RF in the aspects of data and theory.

First, we randomly collected 2000 negative pairs correctly predicted by both DNN and RF (BOTH-right pairs), and 2000 negative pairs correctly predicted by DNN but wrongly predicted by RF (DNN-right pairs) in a test set (HCC-N, RT range: 1-80 min, m/z tolerance: 0.03 Da, the same way as we collected the negative pairs in the training set HCC-T, see “Workflow of DeepRTAlign” section in the manuscript for details). We found that the average RT differences of the 2000 BOTH-right pairs and the 2000 DNN-right pairs were similar (5.03 min and 4.89 min, respectively). However, the average m/z difference of the 2000 BOTH-right pairs was 0.011 Da, which was five times the average m/z differences of the 2000 DNN-right pairs (0.002 Da). **These results indicated that the advantages of DNN may mainly lie in recognizing the negative pairs with close m/z values.**

Second, we found that those negative pairs with similar m/z values (with differences less than 0.002 Da) were rare cases in the whole training data (**10% only**).

Traditional machine learning methods usually are difficult to discriminate and predict correctly in such a situation. Although the bootstrapping-based RF is supposed to alleviate the problem such as imbalanced data in some degree, its classification power will drastically drop when the minority is extremely rare (Xu, Z., Shen, D., Nie, T., & Kou, Y. (2020). A hybrid sampling algorithm combining M-SMOTE and ENN based on Random forest for medical imbalanced data. *Journal of biomedical informatics*, 103465). The minor pairs might be scarcely sampled during the bootstrapping and their effects on trees are overwhelmed by the majority of the data. **In contrast, the DNN model is trained with mini-batches iteratively through gradient descent, which guarantees each pair, including the rare cases, is learned through a shuffled order. Particularly, those wrongly predicted rare cases lead to greater loss values and thus greater gradients to update the model, which may explain why DNN can predict them better.**

Third, for the 40 features (5×8 vector, **Supplementary Fig. 1**) we used in DeepRTAlign, we calculated the importance of each feature in the DNN model, the RF model and the LR model to show which feature is more important when training the DNN, RF and LR models on the same training set (HCC-T). Please note that for DNN, the importance of each feature in DNN is calculated as the AUC decrease tested in the HCC-N dataset when each feature in DNN was replaced with a random value, compared to the DNN model without random features. For RF, the feature importance is the attribute “feature_importances_” in `sklearn.ensemble.RandomForestClassifier` (scikit-learn framework v0.21.3). For LR, the feature importance is the attribute “coef_” in `sklearn.linear_model.LogisticRegression` (scikit-learn framework v0.21.3).

We found that feature 6 ($mz_n - mz_m$) and feature 16 ($mz_m - mz_n$) were the top 2 important features, indicating that DNN, RF and LR models rely mostly on the features about m/z for alignment prediction. We also found the importances of feature 5 ($RT_n - RT_m$) and feature 15 ($RT_m - RT_n$) in DNN are about half those of feature 6 ($mz_n - mz_m$) and feature 16 ($mz_m - mz_n$), while the ratio (importance of feature 5/importance of feature 6) decreased to only about 32.7% in RF and 0.72% in LR. This difference indicates that the importance of RT-related features in the DNN model is relatively

higher than that in the RF and LR models (**Table R2**, denoted as **Supplementary Table 11** in this revision).

In summary, we have added more detailed descriptions and discussions about the technical explanation and analysis of our algorithm in this revision (**Pages 12-13**). We thank the reviewer again for the helpful suggestion.

Table R2 (Supplementary Table 11 in this revision). The list of feature importance of the DNN model, the RF model and the LR model. The DNN model, the RF model and the LR model were trained on the same training set (HCC-T dataset). Please note that the feature importance of LR model is ranked by the absolute value of “coef_”.

DNN			RF			LR		
index	importance	Features	index	importance	Features	index	importance	Features
16	0.107	mZ _m -mZ _n	16	0.205	mZ _m -mZ _n	16	8.804	mZ _m -mZ _n
6	0.107	mZ _n -mZ _m	6	0.199	mZ _n -mZ _m	6	8.803	mZ _n -mZ _m
5	0.048	RT _n -RT _m	5	0.065	RT _n -RT _m	27	-2.435	RT _{n+1}
15	0.048	RT _m -RT _n	15	0.058	RT _m -RT _n	21	-2.294	RT _{n-2}
11	0.026	RT _{m-2} -RT _n	11	0.030	RT _{m-2} -RT _n	25	1.129	RT _n
13	0.024	RT _{m-1} -RT _n	35	0.025	RT _m	37	0.928	RT _{m+1}
1	0.023	RT _{n-2} -RT _m	25	0.023	RT _n	29	0.854	RT _{n+2}
17	0.017	RT _{m+1} -RT _n	13	0.018	RT _{m-1} -RT _n	31	0.679	RT _{m-2}
10	0.010	mZ _{n+2} -mZ _m	19	0.017	RT _{m+2} -RT _n	23	0.415	RT _{n-1}
4	0.010	mZ _{n-1} -mZ _m	17	0.016	RT _{m+1} -RT _n	35	0.397	RT _m
31	0.006	RT _{m-2}	37	0.016	RT _{m+1}	17	0.234	RT _{m+1} -RT _n
21	0.006	RT _{n-2}	27	0.015	RT _{n+1}	32	-0.216	mZ _{m-2}
8	0.005	mZ _{n+1} -mZ _m	39	0.015	RT _{m+2}	30	-0.216	mZ _{n+2}
14	0.005	mZ _{m-1} -mZ _n	7	0.014	RT _{n+1} -RT _m	34	-0.216	mZ _{m-1}
9	0.004	RT _{n+2} -RT _m	9	0.013	RT _{n+2} -RT _m	38	-0.216	mZ _{m+1}
3	0.002	RT _{n-1} -RT _m	36	0.013	mZ _m	24	-0.216	mZ _{n-1}
7	0.002	RT _{n+1} -RT _m	23	0.013	RT _{n-1}	40	-0.216	mZ _{m+2}
18	0.002	mZ _{m+1} -mZ _n	12	0.012	mZ _{m-2} -mZ _n	36	-0.216	mZ _m
34	0.002	mZ _{m-1}	1	0.012	RT _{n-2} -RT _m	26	-0.216	mZ _n
20	0.001	mZ _{m+2} -mZ _n	14	0.012	mZ _{m-1} -mZ _n	22	-0.216	mZ _{n-2}
36	0.001	mZ _m	3	0.012	RT _{n-1} -RT _m	28	-0.216	mZ _{n+1}
22	0.001	mZ _{n-2}	10	0.012	mZ _{n+2} -mZ _m	18	-0.210	mZ _{m+1} -mZ _n
33	0.001	RT _{m-1}	32	0.012	mZ _{m-2}	4	-0.185	mZ _{n-1} -mZ _m
38	0.001	mZ _{m+1}	34	0.012	mZ _{m-1}	11	-0.163	RT _{m-2} -RT _n
19	0.001	RT _{m+2} -RT _n	21	0.011	RT _{n-2}	19	-0.141	RT _{m+2} -RT _n

28	0.001	mZ _{n+1}	38	0.011	mZ _{m+1}	33	-0.101	RT _{m-1}
40	0.001	mZ _{m+2}	18	0.011	mZ _{m+1} -mZ _n	20	0.080	mZ _{m+2} -mZ _n
26	0.001	mZ _n	22	0.011	mZ _{n-2}	7	-0.080	RT _{n+1} -RT _m
30	0.000	mZ _{n+2}	29	0.011	RT _{n+2}	8	-0.074	mZ _{n+1} -mZ _m
32	0.000	mZ _{m-2}	30	0.010	mZ _{n+2}	14	-0.073	mZ _{m-1} -mZ _n
25	0.000	RT _n	2	0.010	mZ _{n-2} -mZ _m	5	0.063	RT _n -RT _m
23	0.000	RT _{n-1}	33	0.010	RT _{m-1}	15	0.063	RT _m -RT _n
24	0.000	mZ _{n-1}	20	0.010	mZ _{m+2} -mZ _n	39	0.058	RT _{m+2}
2	0.000	mZ _{n-2} -mZ _m	8	0.010	mZ _{n+1} -mZ _m	9	-0.045	RT _{n+2} -RT _m
12	0.000	mZ _{m-2} -mZ _n	40	0.010	mZ _{m+2}	2	-0.040	mZ _{n-2} -mZ _m
27	0.000	RT _{n+1}	24	0.010	mZ _{n-1}	1	0.036	RT _{n-2} -RT _m
29	0.000	RT _{n+2}	26	0.009	mZ _n	13	0.036	RT _{m-1} -RT _n
35	0.000	RT _m	28	0.009	mZ _{n+1}	10	-0.026	mZ _{n+2} -mZ _m
37	0.000	RT _{m+1}	31	0.009	RT _{m-2}	12	-0.023	mZ _{m-2} -mZ _n
39	0.000	RT _{m+2}	4	0.009	mZ _{n-1} -mZ _m	3	0.010	RT _{n-1} -RT _m

#3. In Supplementary Tables 4-7, it appears that hyperparameters were tuned sequentially. This approach may result in suboptimal parameter combinations, particularly for complex classifiers like RF and Support Vector Machines (SVMs). Building on the previous point (point #1), if the authors choose to re-tune the hyperparameters with the new features as suggested, I would recommend employing a grid search for both RF and SVMs. This approach is not necessary for LR and KNN since they only involve a single main parameter.

Response: We thank the reviewer for this constructive suggestion. As we claimed in the response for point #1, the input features of DNN and other machine learning methods are exactly the same. Following the reviewer's suggestion, we employed grid searches on RF (n_estimators=[10, 50, 100, 150], max_depth=[10, 50, 100, none]) and SVM (kernel=["linear", "poly", "rbf", "sigmoid"], gamma=["scale", "auto", 0.01, 0.1, 1], C=[0.1, 1, 5], degree=[1, 3, 5]), and found the hyperparameters we used still achieved the best results. The grid search results have been updated to **Supplementary Table 4 (RF) and Supplementary Table 6 (SVM)** in this revision.

Reviewers' Comments:

Reviewer #1:

Remarks to the Author:

My concerns have been addressed. Thank you.

Reviewer #2:

Remarks to the Author:

Thanks for addressing the concerns and explaining the model as it is clear that DNN uses both m/z differences and RT differences more effectively than traditional logistic regression models and Random Forest models.

If you are including Figure R2, You must show that alignment doesn't affect quantification negatively. One can reduce the missing values from 75% tpo 34% but the quantification should be accurate. This is reflected in the quantitation ratio which is missing.

In Figure 3, you mention that OpenMS has better features that is why deep learning is not able to perform better than OpenMS.

In Figure R3, you should include the median line and box plots for both Samples. Otherwise, it is difficult to see the differences by just scatter plots.

Reviewer #3:

Remarks to the Author:

I would like to acknowledge the authors for their work on this article. After this review, all my concerns have been satisfactorily addressed, and I recommend that the work be admitted for publication.

Reviewer #1 (Remarks to the Author):

My concerns have been addressed. Thank you.

Response: We appreciate the reviewer's constructive comments in the previous reviews, which helped us improve the algorithm and the manuscript.

Reviewer #2 (Remarks to the Author):

Thanks for addressing the concerns and explaining the model as it is clear that DNN uses both m/z differences and RT differences more effectively than traditional logistic regression models and Random Forest models.

If you are including Figure R2, you must show that alignment doesn't affect quantification negatively. One can reduce the missing values from 75% to 34% but the quantification should be accurate. This is reflected in the quantitation ratio which is missing.

Response: We thank the reviewer for this valuable suggestion. Following this suggestion, we first checked the previous results, and found that these results were obtained from the earlier version of DeepRTAlign which used "mass tolerance: ± 10 ppm, RT tolerance: ± 5 min" when matching aligned features to peptide sequences. While according to the suggestions of Reviewer #2 in the first round of review, we have updated the matching method as "mass tolerance: ± 10 ppm, RT tolerance: restrict the RT of a peptide to be within the RT range of the corresponding precursor feature" in the current version of DeepRTAlign. Thus, the missing rates in dataset Benchmark-MV have been updated to 68% for MSFragger and 41% for DeepRTAlign.

Importantly, following the reviewer's suggestion, we calculated the aligned feature number and the quantitation ratios for different groups of the six Orbitrap Exploris 480 mass spectrometers in dataset Benchmark-MV. As shown in the figure below (denoted as Supplementary Fig. 4 in this revision, replacing the original Supplementary Fig. 4 in last revision since it didn't provide quantitative information), the identified peptides aligned by DeepRTAlign are more than those aligned by MSFragger with MBR. Meanwhile, the boxplots of quantitation ratios of both E. coli peptides and HEK 293T peptides for different groups on six MS instruments (2700

ng/900 ng, 900 ng/300 ng, and 300 ng/100 ng for *E. coli* peptides; 9100 ng/7200 ng, 9700 ng/9100 ng, and 9900 ng/9700 ng for HEK 293T peptides) show that DeepRTAlign does not affect quantification accuracy negatively.

Supplementary Fig. 4. Comparison of DeepRTAlign and MSFragger on Benchmark-MV dataset. **a, e** Feature numbers corresponding to the *E. coli* peptides and the HEK 293T peptides identified in dataset Benchmark-MV by MSFragger with match between runs (MBR) and DeepRTAlign, respectively. **b-d** and **f-h** Ratio boxplots for the features corresponding to the *E. coli* peptides or the HEK 293T peptides identified in dataset Benchmark-MV by MSFragger with MBR and DeepRTAlign, respectively. The orange dashed line indicates the theoretical ratio. For DeepRTAlign results, features were extracted by Dinosaur, and then aligned by DeepRTAlign. MSFragger's identification results were used to match these features (mass tolerance: ± 10 ppm, RT tolerance: restrict the RT of a peptide to be within the RT range of the corresponding precursor feature). Source data are provided as a Source Data file.

In Figure 3, you mention that OpenMS has better features that is why deep learning is not able to perform better than OpenMS. In Figure R3, you should include the median line and box plots for both Samples. Otherwise, it is difficult to see the differences by just scatter plots.

Response: We thank the reviewer for this valuable suggestion. We have added the median line and boxplots for both samples in Figure R3. Here is the updated Figure R3.

Figure R3a. The fold change (FC) vs. intensity (upper) and m/z (lower) plots of 15 ng/10 ng samples in replicate 1 (R1). The dotted lines indicate the theoretical FC values of HEK 293T (orange) and *E. coli* (blue).

Figure R3b. The fold change (FC) vs. intensity (upper) and m/z (lower) plots of 15 ng/10 ng samples in replicate 2 (R2). The dotted lines indicate the theoretical FC values of HEK 293T (orange) and E. coli (blue).

Figure R3c. The fold change (FC) vs. intensity (upper) and m/z (lower) plots of 15 ng/10 ng samples in replicate 3 (R3). The dotted lines indicate the theoretical FC values of HEK 293T (orange) and *E. coli* (blue).

Figure R3d. The fold change (FC) vs. intensity (upper) and m/z (lower) plots of 20 ng/10 ng samples in replicate 1 (R1). The dotted lines indicate the theoretical FC values of HEK 293T (orange) and *E. coli* (blue).

Figure R3e. The fold change (FC) vs. intensity (upper) and m/z (lower) plots of 20 ng/10 ng samples in replicate 2 (R2). The dotted lines indicate the theoretical FC values of HEK 293T (orange) and *E. coli* (blue).

Figure R3f. The fold change (FC) vs. intensity (upper) and m/z (lower) plots of 20 ng/10 ng samples in replicate 3 (R3). The dotted lines indicate the theoretical FC values of HEK 293T (orange) and E. coli (blue).

Figure R3g. The fold change (FC) vs. intensity (upper) and m/z (lower) plots of 25 ng/10 ng samples in replicate 1 (R1). The dotted lines indicate the theoretical FC values of HEK 293T (orange) and *E. coli* (blue).

Figure R3h. The fold change (FC) vs. intensity (upper) and m/z (lower) plots of 25 ng/10 ng samples in replicate 2 (R2). The dotted lines indicate the theoretical FC values of HEK 293T (orange) and E. coli (blue).

Figure R3i. The fold change (FC) vs. intensity (upper) and m/z (lower) plots of 25 ng/10 ng samples in replicate 3 (R3). The dotted lines indicate the theoretical FC values of HEK 293T (orange) and *E. coli* (blue).

Reviewer #3 (Remarks to the Author):

I would like to acknowledge the authors for their work on this article. After this review,

all my concerns have been satisfactorily addressed, and I recommend that the work be admitted for publication.

Response: We appreciate the reviewer's constructive comments in the previous reviews, which helped us improve the algorithm and the manuscript.